# Arabidopsis TCP4 transcription factor inhibits high temperature-induced homeotic conversion of ovules

Jingqiu Lan [1,2,3], Ning Wang[1,3], Yutao Wang[1], Yidan Jiang[1], Hao Yu[1], Xiaofeng Cao [2] & Genji Qin [1] ✉

Abnormal high temperature (HT) caused by global warming threatens plant survival and food security, but the effects of HT on plant organ identity are elusive. Here, we show that Class II TEOSINTE BRANCHED 1/CYCLOIDEA/ PCF (TCP) transcription factors redundantly protect ovule identity under HT. The duodecuple *tcp2/3/4/5/10/13/17/24/1/12/18/16 (tcpDUO)* mutant displays HT-induced ovule conversion into carpelloid structures. Expression of *TCP4* in *tcpDUO* complements the ovule identity conversion. TCP4 interacts with AGAMOUS (AG), SEPALLATA3 (SEP3), and the homeodomain transcription factor BELL1 (BEL1) to strengthen the association of BEL1 with AG-SEP3. The *tcpDUO* mutant synergistically interacts with *bel1* and the ovule identity gene *SEEDSTICK* (*STK*) mutant *stk* in *tcpDUO bel1* and *tcpDUO stk*. Our findings reveal the critical roles of Class II *TCPs* in maintaining ovule identity under HT and shed light on the molecular mechanisms by which ovule identity is determined by the integration of internal factors and environmental temperature.

Global warming and the increased frequency, intensity, and duration of high temperature (HT) severely threaten the worldwide food security, as well as the diversity and distribution of plants and animals[1]. As sessile organisms, plants undergo morphological changes in a process called plant thermomorphogenesis under ambient HT. The changes associated with plant thermomorphogenesis include longer hypocotyls, leaf petioles and roots, as well as early flowering, male sterility and accelerated fruit dehiscence[2–6]. In contrast to plants, animals can move to avoid adverse living conditions such as HT, and they do not undergo morphological changes like plants under HT. However, embryo development in immobile eggs may be highly affected by HT, and this effect of HT is particularly severe in some fish and reptile species[7]. It is well established that differences in hatching temperature lead to the homeotic conversion of the sexual organs of embryos[7]. For example, the eggs of the turtle *Trachemys scripta* develop into males at 26 °C hatching temperature, but they develop into females at 32 °C, due to sexual homeotic transformation mediated by temperature-sensitive epigenetic

regulation[7]. Although plant thermomorphogenesis has been studied extensively in recent years, the effects of HT on the homeotic transformation of reproductive organs in plants remain largely unknown.

Ovules are the site of double fertilization and the precursors of seeds in angiosperms, and they are essential for plant reproduction and dispersal[8,9]. Ovules are initiated within the carpels, which are the fourth-whorl organs of flowers[10]. An angiosperm ovule typically consists of a proximal funiculus connecting to the placenta at the adaxial margin of the carpel and a distal nucellus covered by the inner and outer integuments initiated from the central chalaza[9]. The nucellus is the site at which female gametophytes are formed by megasporogenesis and megagametogenesis. In Arabidopsis, megasporogenesis begins with a subepidermal cell specified to develop into a megaspore mother cell (MMC) in the distal nucellus. The MMC undergoes meiosis to give rise to the functional megaspore (FM) which further develops into the female gametophyte[9]. The funiculus coordinately elongates, and the inner and outer integuments grow

[1]State Key Laboratory of Protein and Plant Gene Research, School of Life Sciences, Peking University, Beijing 100871, China. [2]State Key Laboratory of Plant Genomics and National Center for Plant Gene Research, CAS Center for Excellence in Molecular Plant Sciences, Institute of Genetics and Developmental Biology, Chinese Academy of Sciences, Beijing 100101, China. [3]These authors contributed equally: Jingqiu Lan, Ning Wang. ✉e-mail: qingenji@pku.edu.cn

asymmetrically to envelop the nucellus, leaving a micropyle for pollen tube entry to complete the double fertilization[9].

The specification and coordinated development of the parts of the ovule are tightly regulated and are very critical for the formation of a functional female gametophyte and thus plant reproduction. According to the floral quartet model, the ovules are specified by MADS-box quartets consisting of SEPALLATA (SEP) proteins, AGAMOUS (AG), and AG homologs such as SEEDSTICK (STK) and SHATTERPROOF1 (SHP1) or SHP2, while the carpels outside ovules are specified by quartet complexes comprising only SEPs and AG[11,12]. The co-existence of SEPs, AG, STK and SHPs suggests that the AG-SEPs-STK/SHPs quartets specifying ovule identity co-exist in the ovules with AG-SEPs quartets specifying carpel identity, suggesting that regulatory mechanisms exert tight control over the activity of AG-SEPs-STK/SHPs quartets and AG-SEPs quartets to promote ovule identity, but not carpel identity, in ovules. Genetic analysis has shown that the activity of AG-SEPs-STK/SHPs quartets and AG-SEPs quartets, and the balance between them, are essential for the specification of ovules[13–15]. It has been shown that *ag* single mutants, *sep1 sep2 sep3* triple mutants and *sep1 sep2 sep3 sep4* quadruple mutants do not form carpels or ovules because they lack both types of quartets[16–18]. Overexpression of *AG* causes the transformation of ovules into carpelloid structures, possibly due to increased activity of AG-SEPs quartets in ovules[19,20]. In the haploid-insufficient *SEP1/sep1 sep2 sep3* triple mutants, ovules are also converted into carpelloid structures, suggesting that SEPs might be inclined to form AG-SEPs quartets instead of AG-SEPs-STK/SHPs quartets when SEP proteins are deficient in ovules[21]. *STK* is specifically expressed in ovules[22,23]. Disruption of *STK* causes longer ovule funiculi[22], while *shp1 shp2* double mutants generate normal ovules[24]. However, *stk shp1 shp2* triple mutants produce carpelloid structures instead of ovules owing to a lack of AG-SEPs-STK/SHPs quartets[22]. Few factors have been identified to control the balance between the activity of AG-SEPs-STK/SHPs quartets and AG-SEPs quartets in ovules. The homeodomain transcription factor BELL1 (BEL1) has been demonstrated to regulate the identity of ovules in a temperature-dependent manner[25]. The *bel1* produced ovules that could be converted into carpelloid structures at the normal growth temperature (22 °C), but the phenotype is less severe at the low temperature (16 °C)[25]. BEL1 interferes with the function of AG-SEPs quartets by directly interacting with the AG-SEPs dimer[13,14]. However, the molecular mechanisms by which ovule identity are affected by temperature, as well as those by which BEL1 stabilizes ovule identity against the effects of temperature differences, are still elusive.

TEOSINTE BRANCHED 1/CYCLOIDEA/PCF (TCP) proteins belong to a plant-specific transcription factor family that is present in all plants[26]. These proteins contain a conserved TCP domain that is responsible for interacting with DNA or proteins[27]. The TCP family proteins in Arabidopsis are classified into the Class I and Class II subfamilies according to sequence differences in the TCP domain[26,28,29]. The Class I subfamily consists of thirteen TCP members, while the Class II subfamily contains eleven TCPs[26,28,29]. The Class II subfamily is further divided into three TB1/CYC-like TCPs (TCP1, TCP12, and TCP18) and eight CINCINNATA (CIN)-like TCPs (TCP2, TCP3, TCP4, TCP5, TCP10, TCP13, TCP17 and TCP24)[26,27]. Class II TCPs have highly redundant and dosage-dependent functions in the control of plant development[29]. They play critical roles in plant developmental plasticity and diverse processes, including leaf development[30–34], shoot branching[35], flower morphogenesis[36], trichome formation[37,38], thermomorphogenesis[39,40], and photomorphogenesis[41]. However, Class II TCPs have not yet been shown to play a role in controlling ovule identity.

We previously demonstrated that TCP5, TCP13 and TCP17 promote plant thermomorphogenesis in response to HT[39], and we showed that the activity of CIN-like TCPs is repressed by the transcriptional repressor SPOROCYTELESS/NOZZLE (SPL/NZZ) to control the specification of MMC by recruiting TOPLESS/TOPLESS-RELATED (TPL/TPR)

co-repressors during ovule development[42]. The *tcp3/4/5/10* quadruple mutant produced fertile ovules abnormally arranged in locules[42]. In this work, we generated a septuple *tcp2/3/4/5/10/13/17* (*tcpSEP*) mutant and a duodecuple *tcp2/3/4/5/10/13/17/24/1/12/18/16* (*tcpDUO*) mutant. Although *tcpSEP* generated fertile ovules, many ovules of *tcpDUO* were infertile and showed abnormal growth of funiculi, integuments, and nucelli. Interestingly, all of the ovules of *tcpDUO* were converted into carpelloid structures under HT (28 °C), indicating that Class II TCPs play central roles in preventing the homeotic transformation of ovules under HT. We further showed that BEL1 was unstable, and TCP4 interacted with BEL1 and stabilized the complexes of BEL1-AG-SEPs to maintain the repression of AG-SEPs and ovule identity under HT.

## Results

### TCP transcription factors play highly redundant and critical roles during ovule development

We previously reported that overexpression of *TCPs* disrupts MMC production during ovule development[42]. However, the *tcp3/4/5/10* quadruple mutant displayed only weak ovule developmental phenotypes; therefore, we hypothesized that the high redundancy of Class II TCP transcription factors could mask the important roles of TCPs in ovules. We first investigated ovule development in a septuple *tcp2/3/4/5/10/13/17* (*tcpSEP*) mutant generated by genetic crosses using T-DNA insertion mutants as described previously[38]. The *tcpSEP* mutants generated ovules and seeds with no significant differences from those of wild-type control plants, except that the siliques of *tcpSEP* were shorter (Fig. 1a), and the seeds were more crowded than that in wild-type control (Fig. 1b, c). This phenotype resembled that observed in *tcp3/4/5/10*[42].

The eleven Class II *TCP* genes and *TCP16*, which is classified as a Class I TCP member but contains a TCP domain more similar to that of Class II TCP proteins[43], are rather evenly distributed in all five chromosomes (Supplementary Fig. 1a). In order to finalize the possible roles of the eleven Class II *TCP* genes and *TCP16* in controlling ovule development, we decided to generate a *tcp* multiple mutant in which all these *TCPs* were mutated. We identified *SALK_023116*, *GABI_655C10* and *SALK_091920*, with a T-DNA insertion in the exon of *tcp12*, *tcp16*, and *tcp18*, respectively, and we generated *tcp1* and *tcp24* mutants carrying one-bp insertion in the appropriate exon using CRISPR/Cas9 technology (Supplementary Fig. 1b). We finally generated a duodecuple null mutant *tcp2/3/4/5/10/13/17/24/1/12/18/16* (*tcpDUO*) by genetic crosses. The *tcpDUO* generated waving leaves and displayed late flowering as that observed in *tcpSEP* when compared to the wild-type control (Supplementary Fig. 2a, b). However, the length of pistils in *tcpDUO* was significantly shorter than that in wild-type control (Supplementary Fig. 2c, d). To determine whether the pistils of *tcpDUO* could affect the growth of pollen tube, we pollinated the pistils of *tcpDUO* or wild-type control with wild-type pollen. The staining with aniline blue showed that pollen germinated on the stigma and pollen tubes grew through styles in pistils from both *tcpDUO* and wild-type control (Supplementary Fig. 2e, f). After pollination, the siliques of *tcpDUO* mutant did not fully elongated and were even shorter than that of wild-type control, and displayed low fertility (about 18.01% of 1259 ovules from 30 siliques) (Fig. 1a–d, i, j). To determine whether defects in male or female gametophytes were the cause of the observed low fertility, we first performed a reciprocal cross between *tcpDUO* and the wild-type control (Table 1). When pollen from *tcpDUO* or wild-type control plants was used to pollinate the pistils of wild-type plants, more than 98% of ovules developed into seeds, while only about 18% ovules from 10 siliques of *tcpDUO* plants formed seeds following pollination with pollen from *tcpDUO* or wild-type control plants (Table 1), suggesting that wild-type pollen did not rescue infertility in the *tcpDUO* siliques. These results indicated that the female gametophytes of *tcpDUO*, but not the male

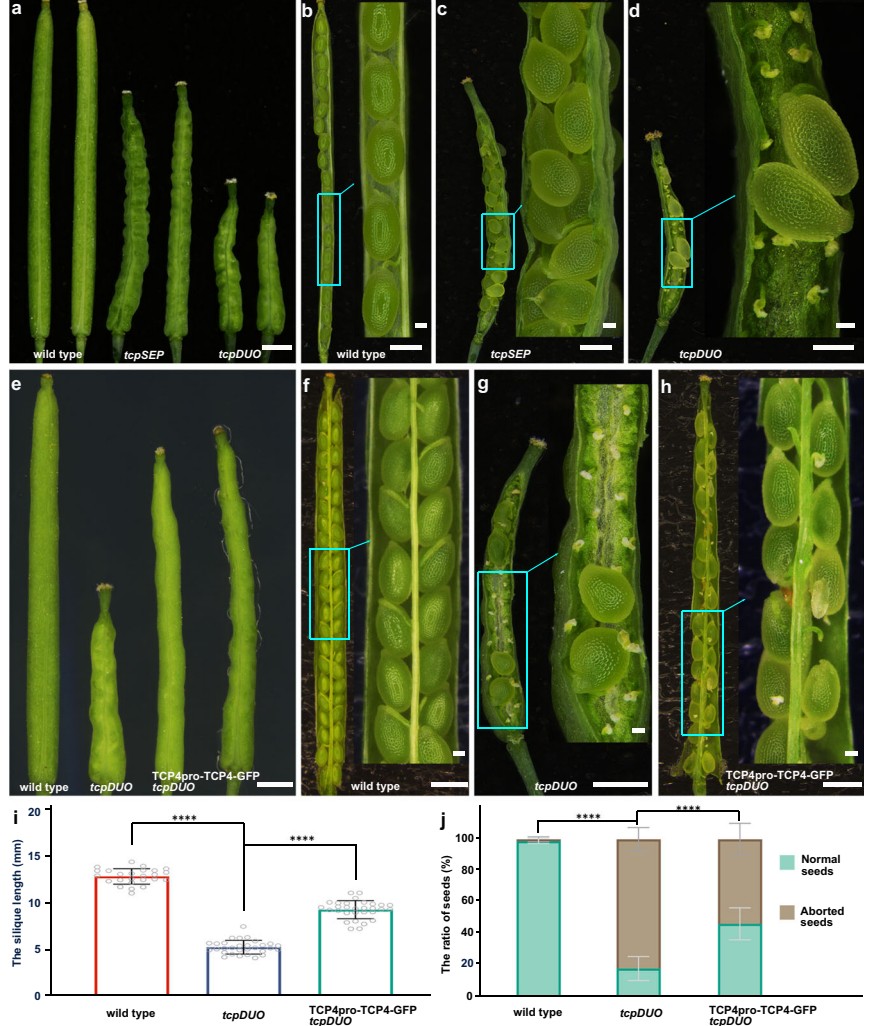

**Fig. 1 | Disruption of Class II TCP transcription factors in duodecuple *tcpDUO* null caused low fertility. a** Siliques from wild type, *tcpSEP*, and *tcpDUO*. **b–d** Dissected siliques and close-up views showing the fertility of wild type (**b**), *tcpSEP* (**c**), and the *tcpDUO* mutant (**d**). **e** Siliques of wild type, *tcpDUO*, and the complementation line TCP4pro-TCP4-GFP *tcpDUO*. **f–h** Dissected siliques and close-up views for wild type (**f**), *tcpDUO* (**g**), and TCP4pro-TCP4-GFP *tcpDUO* (**h**). The low fertility of *tcpDUO* was complemented by the transformation of TCP4pro-TCP4- GFP. **i** Statistical analysis of the silique lengths of wild type, *tcpDUO* and TCP4pro-TCP4-GFP *tcpDUO*. **j** Statistical analysis of aborted and normal seeds in wild-type, *tcpDUO*, and TCP4pro-TCP4-GFP *tcpDUO* siliques. Data are the means ± SD (*n* = 30) of three biological replicates. Student's two-sided t-test was used for statistical analysis. \*\*\*\*, *P* < 0.0001. Source data for **i** and **j** are provided as a Source Data file. Scale bars, 1 mm in **a–h**, 100 μm in the insets of **b–d** and **f–h**.

gametophytes, were defective, and maternally derived *tcp* mutations caused the aborted ovules or seeds in *tcpDUO* plants.

To confirm that TCP transcription factors play key roles in the development of ovules and maternal fertility, we selected *TCP4* as a representative gene and generated TCP4pro-TCP4-GFP, in which *TCP4* (fused with *GFP*) was driven by its own promoter for complementation.

The expression of *TCP4-GFP* in *tcpDUO* by transforming TCP4pro-TCP4-GFP significantly increased the length of siliques and decreased the ratio of aborted ovules or seeds (Fig. 1e–h, j). These results indicate that TCP4 and the other members of the Class II TCP subfamily transcription factors perform essential and redundant functions in the development of ovules and seeds.

## TCP transcription factors regulate the growth of ovule funiculi, integuments, and megagametogenesis

To reveal further details regarding ovule defects in *tcpDUO*, we first observed mature ovules using scanning electron microscopy (SEM). The SEM results showed that the funiculi from *tcpDUO* ovules contained more cells and were clearly longer than those from wild-type control ovules (Fig. 2a–d). Furthermore, while the outer integument grew over the inner integument and nucellus in the mature wild-type ovule, leaving a small, clear opening, the outer and inner integuments from the *tcpDUO* ovule were observed to grow excessively (Fig. 2a–g). We observed 54 ovules produced excessive growth in the outer and/or inner integuments from 57 ovules of *tcpDUO*. Among the 54 abnormal ovules, 46 (85.2%) ovules exhibited excessive growth in both the outer

## Table 1 | The reciprocal crosses between *tcpDUO* and the wild type

| ♀ | ♂ | The ratio of fertility per silique (%)* |
|---|---|---|
| wild type | wild type | 98.6 ± 1.83[a] |
| *tcpDUO* | *tcpDUO* | 16.2 ± 6.83[b] |
| *tcpDUO* | wild type | 17.8 ± 7.51[b] |
| wild type | *tcpDUO* | 98.5 ± 2.02[a] |

*Numbers are mean values with standard deviations in parentheses. For reciprocal crossing, *n* = 10. The different superscript letters (a, b) of the fertility percentages indicate significant differences determined using two-sided one-way ANOVA analysis, *P* < 0.0001. Source data are provided as a Source Data file.

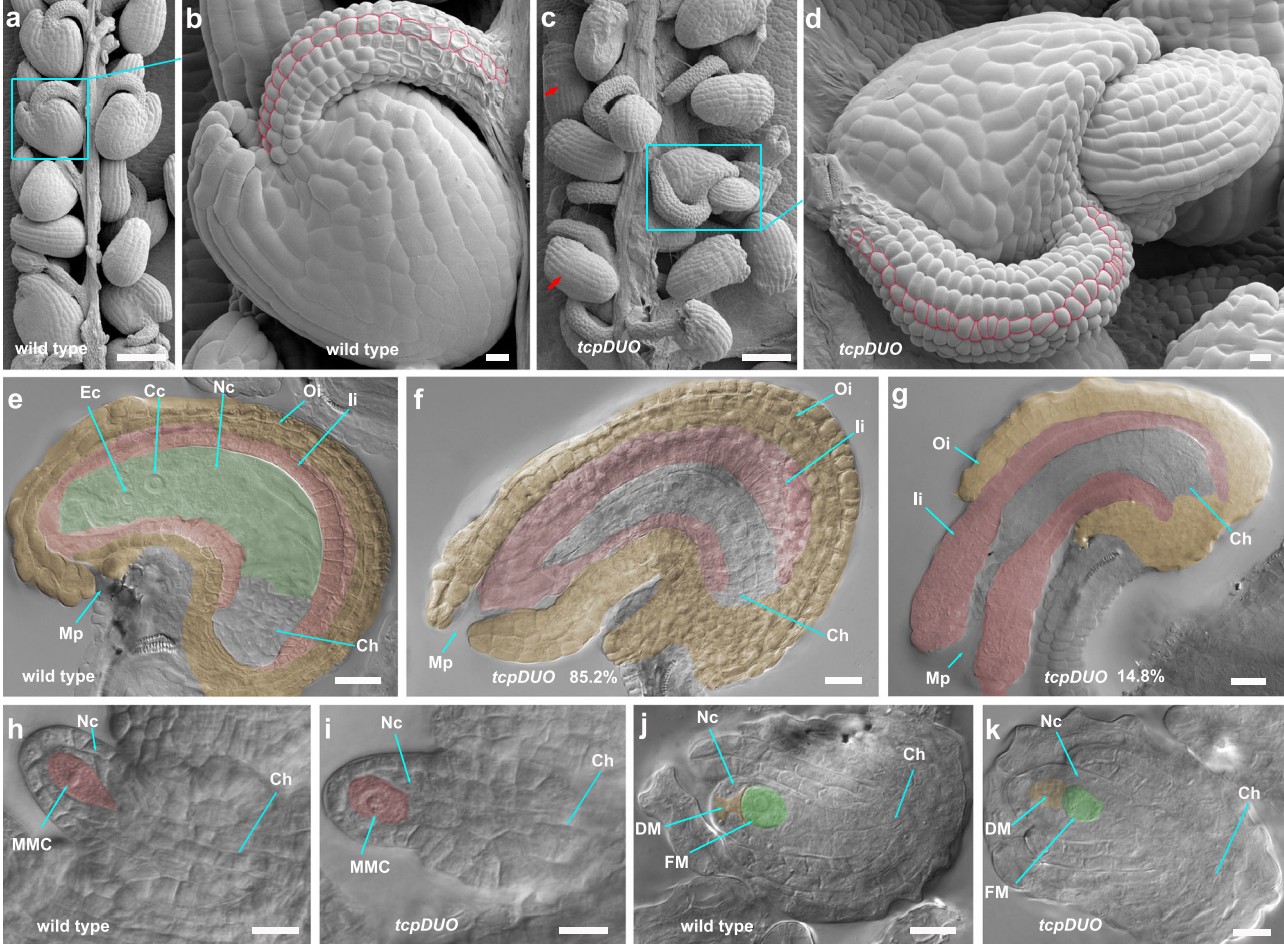

**Fig. 2 | The funiculi, integuments, and megagametogenesis were abnormal in the *tcpDUO* ovules. a–d** SEM analysis of the mature ovules from wild-type (**a, b**) and *tcpDUO* (**c, d**). **b** and **c**, Close-up view of ovules from **a** and **b**. The red arrows indicate the ovules with abnormal integuments. The cells of funiculi are marked with red lines in **b** and **d** to show the different number of cells consisting of funiculi in wild-type and *tcpDUO*. Images of **a–d** are representatives of at least five independent individuals. **e–g** Representative images of DIC micrographs of ovules from wild-type (**e**) and *tcpDUO* (**f, g**) ovules at the FG7 stage. Green-shaded areas are the nucellus in the wild-type ovule. Pink-shaded areas are inner integuments. Yellow-shaded areas are the outer integuments. The egg cell and central cell were clearly observed in the nucellus of wild-type plants (**e**). Excessive growth of outer or inner integuments was observed in the ovules of *tcpDUO* (**f, g**). **h–k** DIC micrographs of ovules from wild-type (**h, j**) and *tcpDUO* (**i, k**) during megasporogenesis. Images are representatives of at least five independent individuals. Source data for **f–g** are provided as a Source Data file. Pink-shaded areas are the megaspore mother cells. Green-shaded areas are the functional megaspore cells. MMC, megaspore mother cell. FM, functional megaspore. DM, degenerated megaspore. Mp, micropyle. Ch, chalaza. Ec, egg cell. Cc, central cell. Nc, nucellus. Ii, inner integument. Oi, outer integuments. Scale bars, 100 µm in **a** and **c**, 10 µm in **b, d, h–k**. 20 µm in **e–g**.

and inner integuments (Fig. 2f), while 8 (14.8%) ovules showed only excessive growth in inner integuments, causing defective ovules with a large micropyle and an outer integument that failed to cover portions of the inner integument (Fig. 2c, d, g).

We then observed megasporogenesis and megagametogenesis during the development of *tcpDUO* ovules. In ovules from *tcpDUO* or wild-type plants, a hypodermal cell was differentiated into a megasporocyte, which further produced an FM and three degenerated megaspores after meiosis at the tip of the nucellus, suggesting that megasporogenesis was not significantly affected in *tcpDUO* (Fig. 2h–k). During megagametogenesis, ovule autofluorescence showed that a clear FM was formed in the ovules from both *tcpDUO* and wild-type plants at the FG1 stage (Supplementary Fig. 3a, b), as demonstrated by observations using a microscope with differential interference contrast (DIC) (Fig. 2j, k). However, at the FG2 stage, two haploid nuclei were formed by the first round of mitosis (Supplementary Fig. 3c, d), but they were immediately degenerated in most *tcpDUO* ovules at the FG3 stage (Supplementary Fig. 3e, f). Four haploid nuclei were obviously observed after the second round of mitosis in wild-type ovules (Supplementary Fig. 3g), whereas more than eighty percent

(84.3%) of *tcpDUO* ovules did not undergo the second and third rounds of mitosis, leading to the presence of an abnormal nucellus without haploid nuclei (Supplementary Fig. 3h). Less than twenty percent (9.3% + 6.2%) of *tcpDUO* ovules contained a nucellus with nuclei in different arrangements (Supplementary Fig. 3i, j), which was consistent with the ratio of aborted seeds in *tcpDUO* plants (Fig. 1j). These results suggest that TCPs play pivotal roles during ovule development.

## TCP transcription factors inhibit the homeotic conversion of ovules into carpelloid structures under HT

During observation of the ovule phenotype of the *tcpDUO* mutant, we found that ovules were converted into carpelloid structures with papillae in a very low ratio (0.2%, *n* = 1259 ovules) (Fig. 3a, b), suggesting that TCP transcription factors might play roles in the determination of ovule identity. Previous report indicated that the homeotic conversion ratio of ovules into carpelloid structures in *bel1* mutants was affected by temperature[25]. Considering that TCPs participate in plant thermomorphogenesis[39,40], we treated *tcpDUO* and wild-type control plants with HT (28 °C). We found that almost all of the ovules in the *tcpDUO* pistils were converted into carpelloid structures (Fig. 3d),

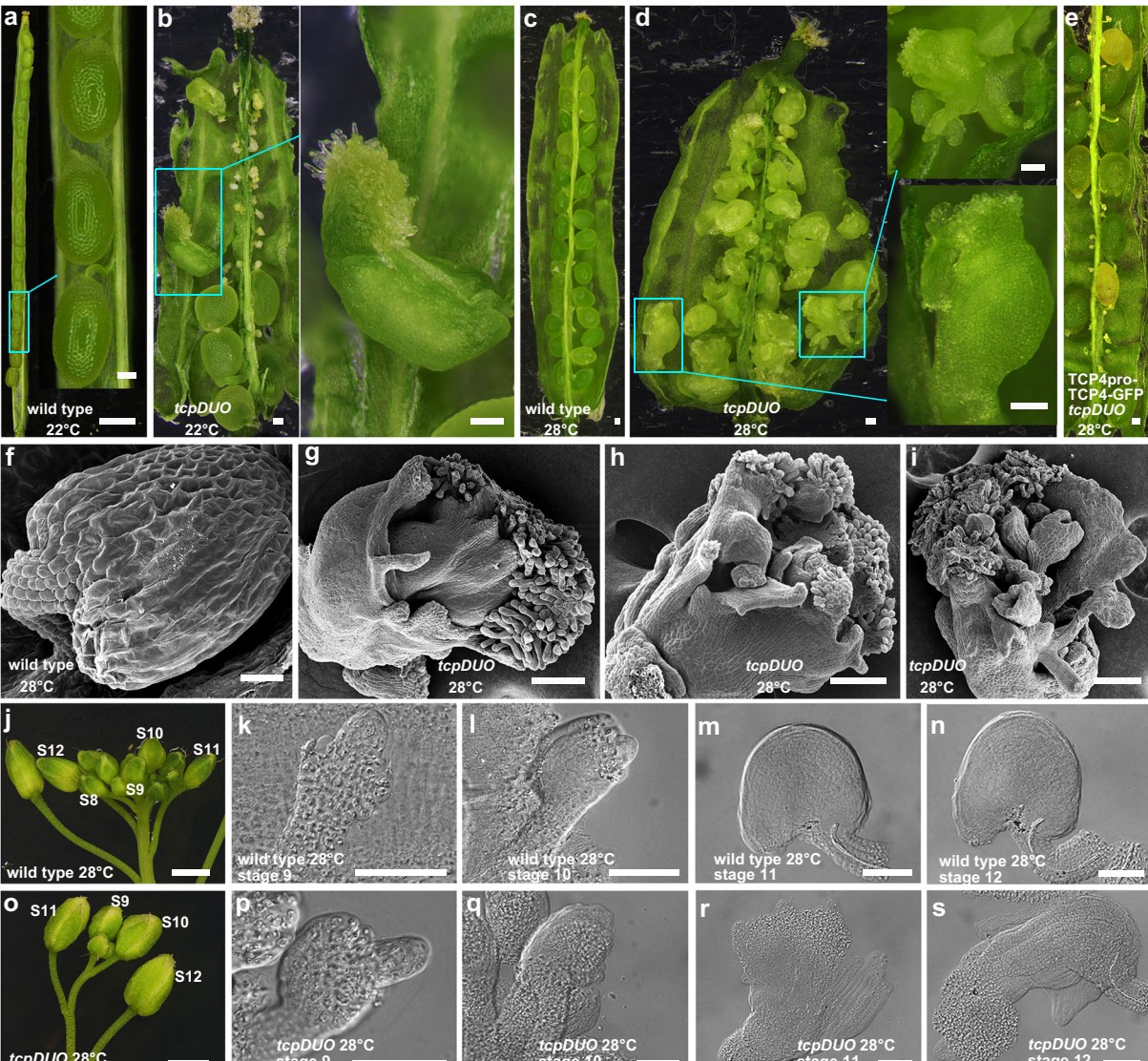

**Fig. 3 | The *tcpDUO* mutant exhibited homeotic conversion of ovules into carpels under HT. a, b** The opened wild-type (**a**) and *tcpDUO* (**b**) siliques under 22 °C. A carpelloid structure in the silique of *tcpDUO* is shown in (**b**). **c, d** Opened siliques from wild-type (**c**), *tcpDUO* (**d**) and TCP4pro-TCP4-GFP *tcpDUO* (**e**) under 28 °C. The ovules were converted into carpelloid structures in *tcpDUO* under 28 °C. The transformation of TCP4pro-TCP4-GFP rescued the homeotic conversion of ovules in *tcpDUO*. **f** SEM observation of a wild-type ovule under 28 °C. **g–i** SEM observation of *tcpDUO* carpelloid structures under 28 °C. **g** A carpelloid structure with papilla-like cells in the outer integument. **h** A carpelloid structure with papilla-like cells in the inner and outer integuments. **i** A carpelloid structure with secondary carpelloid structures. Images of **a–i** are representatives of at least ten independent individuals. **j** The inflorescences of wild type under 28 °C. The flower stages were indicated in the picture. **k–n**, The ovules from the flower at the developmental stage 9 (**k**), stage 10 (**l**), stage 11 (**m**) and stage 12 (**n**) of wild type. Images of **k–n** are representatives of at least three independent individuals. **o** The inflorescences of *tcpDUO* under 28 °C. The flower stages were indicated in the picture. **p–s** The ovules from the flower at the developmental stage 9 (**p**), stage 10 (**q**), stage 11 (**r**) and stage 12 (**s**) of *tcpDUO*. Images of **p–s** are representatives of at least three independent individuals. The pictures of **a–e**, **j**, and **o** were taken with a dissecting microscope. The pictures of **f–i** were obtained by SEM and those of **k–n** and **p–s** were taken using a DIC Microscope. Scale bars, 1 mm in **a**, 100 μm in the inset of **a**, **b–e**, 20 μm in **f–i**. 1 mm in **j** and **o**, 50 μm in **k–n** and **p–s**.

while no ovule identity conversion was observed in the wild-type control under 28 °C treatment (Fig. 3c). The expression of TCP4-GFP by transforming TCP4pro-TCP4-GFP into *tcpDUO* rescued the identity conversion in *tcpDUO* under 28 °C treatment (Fig. 3e), indicating that TCP4 plays a critical role in maintaining ovule identity under HT. The SEM results further showed that the integuments of wild type ovules were rather normal under 28 °C (Fig. 3f). However, the integuments of *tcpDUO* grew excessively, and clear papilla cells covered the top of the inner and/or outer integuments (Fig. 3g–i). In some carpelloid structures, secondary ovules or carpelloid organs were observed (Fig. 3i). To investigate the developmental stage at which the ovules of *tcpDUO* begin to be converted into carpelloid structures, we observed the ovules from the flower developmental stage 9 to stage 12 of *tcpDUO*

and wild type control under 28 °C. The results showed that all the ovules from wild type had no homeotic conversion (Fig. 3j–n). The early ovules of flowers at stage 9 and 10 were also rather normal in *tcpDUO* (Fig. 3o–q). However, the ovules at flower developmental stage 11 began to be homeotically converted into carpelloid structures (Fig. 3r). Clear carpelloid structures were observed in the flowers at stage 11 and 12 in *tcpDUO* under 28 °C (Fig. 3r, s). These results indicate that TCP transcription factors play pivotal roles in maintaining the stability of ovule identity under HT.

To determine the contribution and redundancy of *TCPs* in maintaining ovule identity under HT, we crossed the *tcp* duodecuple mutant (*tcp2/3/4/5/10/13/17/24/1/12/18/16*) with the septuple mutant (*tcp2/3/4/5/10/13/17*) to generate the different combinations of *tcp*

lower order mutants. We obtained an octuple mutant (*tcp2/3/4/5/10/13/17/24*) in which all *CIN*-like *TCP* genes were mutated, a nonuple mutant (*tcp2/3/4/5/10/13/17/1/12*), a decuple mutant (*tcp2/3/4/5/10/13/17/1/12/18*), and a undecuple mutant (*tcp2/3/4/5/10/13/17/24/1/12/18*) in which all Class II *TCPs* were mutated. We treated these *tcp* multiple mutants with 28 °C to observe the phenotype of homeotical conversion. The results showed that the disruption of all the *CIN*-like *TCPs* did not cause the conversion of ovule identity (Supplementary Fig. 4a, b). However, the compromise of all the Class II *TCPs* led to all the ovules being converted into carpelloid structures (Supplementary Fig. 4g, h). This indicated that *CYC*-like *TCPs* played critical and redundant roles with *CIN*-like *TCPs* in protecting the ovule identity under HT, and *TCP16* was dispensable in this process. No ovule conversion events were observed in the nonuple mutant and the decuple mutant (Supplementary Fig. 4c–f), indicating that *TCP24* made an important contribution in maintaining the ovule identity under HT.

## TCP transcription factors directly repressed the expression of carpel-related genes to protect ovule identity

Although the effects of HT on the growth of hypocotyls, petioles, roots, and male organs have been studied extensively[1], the mechanisms underlying the effects of HT on female organs, including the stability of ovule identity, remain largely unknown. To reveal the underlying mechanisms by which TCPs control ovule development and identity under normal temperature and HT, we emasculated the flowers of *tcpDUO* and wild-type control plants under normal temperature or HT, and pistils were collected 2 days after emasculation (DAE). These materials were used to perform RNA-seq analysis with three replicates. The principal component analysis (PCA) showed that the results from the three replicates from each sample were similar (Supplementary Fig. 5a). We first analyzed the expression level of 24 *TCP* family genes in wild-type pistils under normal temperature and HT. The RNA-seq results showed that 22 *TCPs* were expressed in wild-type pistils. *TCP16* was expressed at a very low level in wild-type pistils, and *TCP9* and *TCP1* were not found to be expressed (Supplementary Fig. 6a, b). To determine the effects of HT on the expression level of *TCPs*, we analyzed the expression fold change of *TCP* genes in wild type treated with 22 °C Vs 28 °C (Supplementary Fig. 6c). The results showed that the gene expression was not changed more than 2 fold in the *TCPs* under HT (false discovery rate < 0.01; fold-change ≥ 2.0 or ≤ −2.0).

We collected mature ovules from pistils of wild-type plants treated with 22°C or 28°C to test the expression of Class II *TCPs* and *TCP16* using reverse transcription quantitative PCR (RT-qPCR). The results showed that almost all these *TCPs* except *TCP1* and *TCP17* were expressed in mature ovules (Supplementary Fig. 7a). We further cloned a 3043-bp-bp promoter of *TCP1*, a 2466-bp promoter of *TCP12*, a 2428-bp promoter of *TCP18* and a 2571-bp promoter of *TCP24*. We generate TCP1pro-GUS, TCP12pro-GUS, TCP18pro-GUS, and TCP24pro-GUS constructs in which these promoters were used to drive *GUS* reporter gene, respectively. GUS staining showed that the expression of *TCP1* was hardly detected (Supplementary Fig. 7b, c), while the expression of *TCP12*, *TCP18* and *TCP24* was clearly observed in the early ovules (Supplementary Fig. 7d–j). Staining experiments with the previously reported TCP4pro-TCP4-GUS transgenic line confirmed that TCP4 protein was expressed during ovule development (Supplementary Fig. 7j–m).[38] In situ hybridization showed that *TCP4* was highly expressed in nucellus and integuments, consistent with the GUS staining of TCP4pro-TCP4-GUS (Supplementary Fig. 7o–r). Next, we mapped the reads of 12 *TCPs* to their corresponding genomic regions. In contrast to the *TCP* reads from wild-type plants, which mapped to the whole corresponding *TCP* genomic sequences, the 9 T-DNA insertion mutated *tcp* reads from *tcpDUO* did not, indicating that no complete transcripts for the 9 *TCPs* were generated in the *tcpDUO* null mutant (Supplementary Fig. 6b). In addition, no transcripts of *TCP1* or

*TCP16* were found in wild type and *tcpDUO*, and the transcripts of *TCP24* carrying one-bp-insertion were decreased in *tcpDUO* (Supplementary Figs. 1b, 3b).

We then compared the transcriptomes of wild-type and *tcpDUO* pistils (WT Vs *tcpDUO*) under normal temperature and HT. The results showed that 1851 differentially expressed genes (DEGs, false discovery rate <0.01; fold-change ≥ 2.0 or ≤ −2.0), including 667 up-regulated genes and 1184 down-regulated genes, were identified under 22 °C treatment (22 °C WT Vs *tcpDUO*) (Supplementary Data 1, 2, Supplementary Fig. 5b–d). Under 28 °C treatment, the total number of DEGs (3140) increased, while the number of up-regulated genes (1617) increased almost 2.5-fold (1617/667) (Supplementary Data 3, 4, Supplementary Fig. 5b–d).

Because a low proportion of ovules were converted into carpelloid structures in *tcpDUO* under a low temperature, while almost all ovules showed homeotic conversion under HT, we speculated that the expression level of DEGs related to homeotic organ transformation in *tcpDUO* might increase in response to HT. To identify the DEGs responsible for homeotic ovule conversion, we first constructed a set of overlapping DEGs that were identified in both WT in comparison with *tcpDUO* under HT or normal temperature treatment, including 289 up-regulated genes and 578 down-regulated genes (Supplementary Fig. 5e). Cluster analysis revealed that the overlapping DEGs were clustered into 6 groups according to their expression levels in wild-type and *tcpDUO* plants under 22 °C or 28 °C treatment (Supplementary Data 5, Supplementary Fig. 5f). The DEGs in Cluster 1 and Cluster 2 were up-regulated in *tcpDUO* in comparison with wild type, but had a lower expression level under 28 °C treatment in comparison with 22 °C treatment, or a similar level under the two temperatures. These characteristics suggest that cluster 1 and cluster 2 are unlikely to mediate the ovule conversion phenotype of *tcpDUO* under 28 °C treatment. However, the DEGs in Cluster 3 were expressed at higher levels in *tcpDUO* in comparison with wild type, and the up-regulation of these DEGs was greater in magnitude under HT in comparison with that observed under normal temperature treatment, consistent with the generation of more carpelloid structures in *tcpDUO* under HT. This finding suggested that Cluster 3 DEGs could contribute to the efficient ovule homeotic conversion phenotype of *tcpDUO* under HT. The DEGs in Cluster 4–6 were all down-regulated in *tcpDUO* in comparison with wild type, and the DEGs in Cluster 4 all showed lower expression levels under HT in comparison with normal temperature (Supplementary Fig. 5f). These analyses suggested that the genes in Cluster 3 and 4 might be important for the efficient ovule homeotic conversion phenotype of *tcpDUO* under HT. Gene ontology (GO) analysis of the DEGs in Cluster 3 revealed enrichment in terms such as floral organ morphogenesis, formation and development, and carpel development (Supplementary Data 6, Fig. 4a, b), while the DEGs in Cluster 4 were enriched in the sexual reproduction pathway term (Supplementary Data 7, Fig. 4c, d). Heatmap analysis showed that many genes related to carpel development, including *CRABS CLAW* (*CRC*) (encoding a YABBY transcription factor)[44], *SPATULA* (SPT)[45], *HECATE1* (*HEC1*), *HEC2*, and *HEC3* (encoding basic helix-loop-helix (bHLH) transcription factors)[46], were highly induced in *tcpDUO* in comparison with wild type at 22 °C, with even stronger induction under 28 °C treatment (Fig. 4b). In contrast, many genes related to the formation of female gametophytes, including egg cell-specific genes *EC1.1, EC1.3,* and *EC1.5*[47], were severely down-regulated in *tcpDUO* in comparison with wild type, with even stronger repression under 28 °C treatment (Fig. 4d). In addition, the expression of *INNER NO OUTER* (*INO*) key for ovule development was significantly down-regulated in *tcpDUO* under HT (Supplementary Fig. 8).

Because the most striking phenotype in *tcpDUO* under HT was the conversion of ovules into carpelloid structures, we collected ovules and carpelloid structures from wild-type and *tcpDUO* plants at 2 DAE

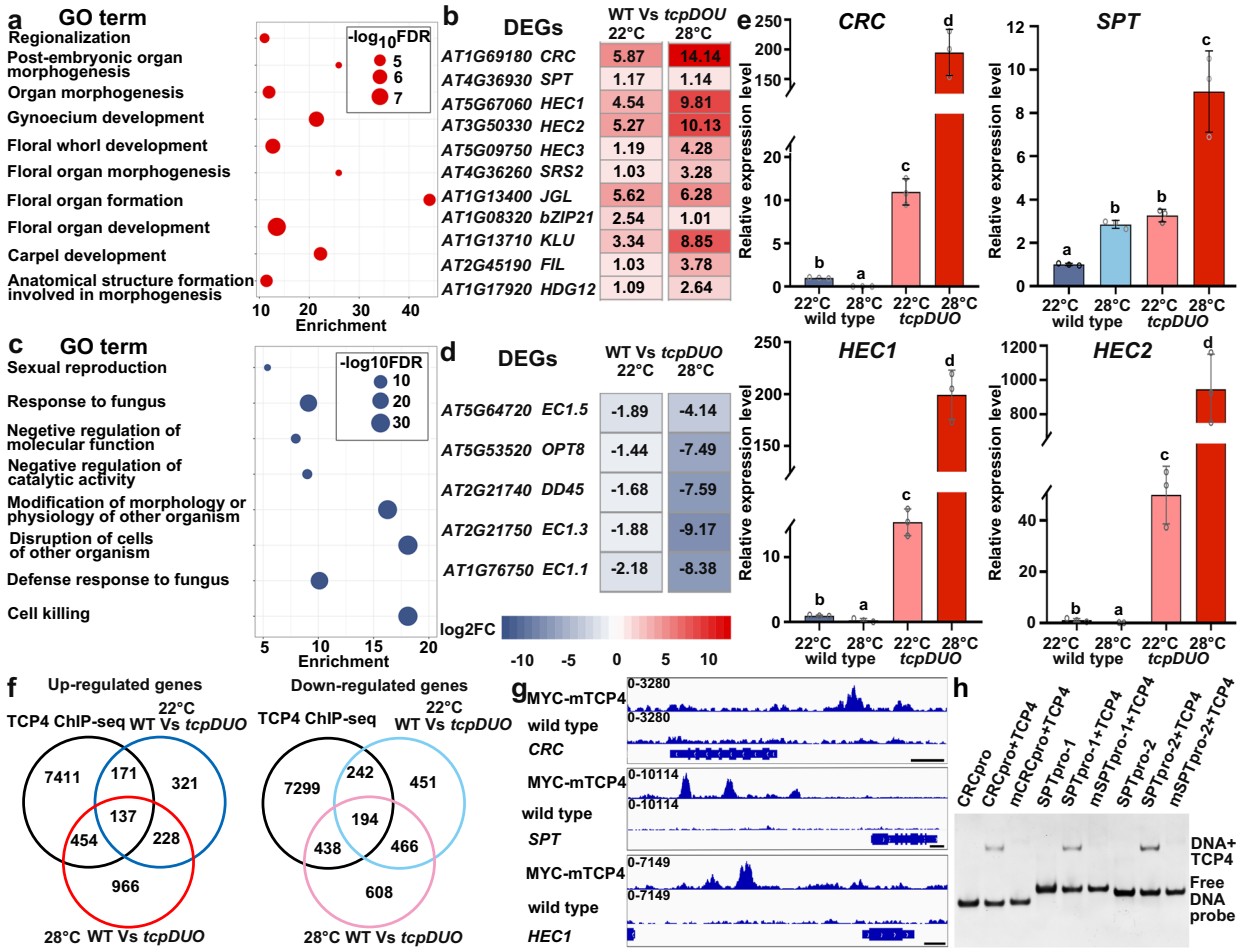

**Fig. 4 | Carpel-related genes were repressed by TCP transcription factors during ovule development. a** GO analysis of the DEGs in cluster 3. **b** Heatmap analysis of the genes related to carpel development. **c** GO analysis of the DEGs in cluster 4. **d** Heatmap analysis of the genes related to the formation of female gametophytes. **e** Relative expression levels of carpel-related genes, including *CRC*, SPT, *HEC1*, and *HEC2*, in 2 DAE ovules from wild-type plants or carpel-like structures from *tcpDUO* under 22 °C or 28 °C. The expression levels of genes in the wild-type ovules were set to 1.0. The data are the mean (±SD) of three biological replicates. One-way ANOVA analysis of variance was performed with LSD/Duncan pairwise comparison testing. Different lowercase letters indicate significant differences ($P < 0.01$). Source data for **a** and **e** are provided as a Source Data file. **f** Overlapping genes between the DEGs of WT vs. *tcpDUO* under 22 °C or 28 °C and the TCP4-binding gene loci. **g** ChIP-seq data showed that TCP4 protein was enriched in the promoter regions of *CRC*, SPT, and *HEC1* in vivo. **h** EMSA analysis indicated that TCP4 protein was directly bound to the promoter of *CRC* and SPT in vitro. The EMSA experiments were repeated at least three times. The results in a–d were from RNA-seq analysis and the results of (**e**) were from RT-qPCR analysis. Source data for **e** and **h** are provided as a Source Data file.

under normal temperature and HT. We used RT-qPCR to measure the relative expression levels of four genes known to be important for carpel development in these tissues. The RT-qPCR results showed that the expression levels of *CRC*, *SPT*, *HEC1* and *HEC2* in *tcpDUO* were much higher than those of wild-type plants under both 22 °C and 28 °C treatments (Fig. 4e). Indeed, the expression levels of *CRC*, *HEC1* and *HEC2* in the ovule-derived carpelloid structures from *tcpDUO* plants were hundreds of folds higher than those of wild-type plants under HT (Fig. 4e), consistent with the RNA-seq data and the ovule homeotic conversion phenotype of *tcpDUO* under HT.

We previously identified thousands of TCP4 binding sites using chromatin immunoprecipitation assays with sequencing (ChIP-seq) with 35S-MYC-mTCP4 in which *MYC* tag fusion with microRNA319 (miRNA319)-resistant *mTCP4* was driven by a CaMV 35 S promoter[41]. The comparison of the ChIP-seq data with the DEGs of WT Vs *tcpDUO* showed that 308 up-regulated and 436 down-regulated genes in WT Vs *tcpDUO* under normal temperature were bound by TCP4 (Fig. 4f). However, 591 up-regulated and 632 down-regulated genes in WT Vs *tcpDUO* under HT overlapped with TCP4 binding

genes (Fig. 4f), suggesting that HT may affect the expression of genes bound by TCP4. Further analysis showed that TCP4 proteins were obviously enriched in the genomic loci of *CRC*, *SPT*, and *HEC1* (Fig. 4g). Gel electrophoresis mobility shift assay (EMSA) results indicated that TCP4 bound to the promoter of *CRC* or *SPT* (Fig. 4h), suggesting that TCP4 directly interacted with the promoters of *CRC* and *SPT*.

Our RNA-seq data also showed that the master regulator *WUSCHEL* (*WUS*) was also significantly up-regulated in the pistils of *tcpDUO* (Supplementary Fig. 9a). RT-qPCR using the ovules as materials confirmed that *WUS* was highly induced in *tcpDUO*, and HT further increased the expression of *WUS* in *tcpDUO* (Supplementary Fig. 9b), suggesting that TCPs play key roles in the suppression of *WUS* under normal or high temperature. This is consistent with the indeterminate growth of carpelloid structures in *tcpDUO* under HT.

These results demonstrate that HT promotes the expression of genes essential for carpel development, including *CRC*, *SPT*, and *HEC1*, and TCP transcription factors stabilize ovule identity by repressing the induction of carpel-related genes by HT.

## TCP transcription factors repress the activity of AG-SEP3 complexes by interacting with AG and SEP3

The balance between the simultaneous activity of AG-SEPs-STK/SHPs (specifying ovule identity) and AG-SEPs (specifying carpel identity) in the ovules determines ovule identity[13]. We searched for AG-SEPs-STK/SHPs quartet genes, including AG, STK, SHP genes, and SEP genes, in our RNA-seq data. The results showed that the expression of STK was up-regulated under normal temperature, but was a little down-regulated under HT in pistils (Supplementary Fig. 10a). SHP1 was significantly induced in the pistils of tcpDUO. The expression of SHP2 and SEP genes were not significantly altered in pistils (Supplementary Fig. 10a). We next tested the expression change of STK in ovules by RT-qPCR. The results showed that the treatment of HT did not change the expression of STK in wild type. However, HT significantly repressed STK in tcpDUO (Supplementary Fig. 10b). In situ hybridization assays confirmed that the expression of STK was decreased and was mainly observed in the funiculi in tcpDUO, when compared to that in wild-type control under HT (Supplementary Fig. 10d–f). These data suggest that TCPs are important for maintaining the expression level of STK in ovules under HT, and the decreased expression of STK is consistent with the homeotic conversion in tcpDUO under HT.

Considering that both AG-SEPs quartets and TCP4 directly regulate carpel development genes, including CRC and SPT[48–50], we hypothesized that TCPs could interact with AG-SEPs at the protein level. We performed yeast two-hybrid assays to examine the interaction between TCPs and AG or SEP3. The results showed that both AG and SEP3 each interacted with TCP4 and TCP18 in yeast two-hybrid assays (Fig. 5a, d). Firefly luciferase complementation imaging (LCI) and co-immunoprecipitation (Co-IP) experiments confirmed that TCP4 interacted with AG and SEP3 in vivo (Fig. 5b, e, c, f). To identify the domains in AG and SEP3 responsible for their interactions with TCP4 and TCP18, we separated both AG and SEP3 into three fragments: AG-MI or SEP3-MI (containing the MADS-box and Intervening domain), AG-IK or SEP3-IK (containing the Intervening and Keratin domains), and AG-C or SEP3-C (containing the C-terminus) (Supplementary Fig. 11a, c). Yeast two-hybrid assays indicated that these truncated proteins all interacted with TCP4 and TCP18 (Supplementary Fig. 11b, d), while no interactions were detected in the negative controls. That is, no interactions between AG and TCP16, SEP3 and TCP16, STK and TCP4, and STK and TCP18 were found (Supplementary Fig. 11b, d). These data suggested that the various domains of AG and SEP3 play roles in their interactions with TCPs.

To further determine the effect of interactions between TCP4 and AG-SEP3 complexes, we cloned the promoters of AG-SEP3 downstream genes CRC and SPT[45–47]. We generated the reporters 35Spro-REN-CRCpro-LUC and 35Spro-REN-SPTpro-LUC, in which the firefly Luciferase (LUC) gene was driven by the CRC or SPT promoter, respectively, and the renilla Luciferase (REN) gene was driven by a CaMV 35S promoter as an internal expression control (Fig. 5g). We co-transformed the effectors 35Spro-AG and 35Spro-SEP3, in which AG and SEP3 were driven by a CaMV 35S promoter, and a reporter (35Spro-REN-CRCpro-LUC or 35Spro-REN-SPTpro-LUC) into Arabidopsis protoplasts. The results showed that the AG-SEP3 complex activated the LUC reporter as expected, while 35Spro-mTCP4, in which miRNA319-resistant mTCP4 was driven by a CaMV 35S promoter, repressed LUC expression (Fig. 5h, i). Co-expression of 35Spro-mTCP4 with the AG-SEP3 complex significantly repressed the LUC reporter expression level, suggesting that TCP4 inhibited the activity of AG-SEP3 (Fig. 5h, i).

These results suggest that TCP proteins prevent AG-SEP3 from activating the transcription of carpel-related genes in ovules.

## TCP4 promotes the association of BEL1 with AG-SEP3 by directly interacting with BEL1

BEL1 has been reported to repress the temperature-dependent homeotic conversion of ovules into carpelloid structures by interacting with the AG-SEP3 complex[13,14]. Our results showed that TCP4 also interacted with AG-SEP3 to repress its activity (Fig. 5), and the ovules of tcpDUO were converted into carpelloid structures under HT (Fig. 3), resembling those observed in a previous study of bel1[13]. However, the expression of BEL1 was not obviously altered in pistils of tcpDUO under normal temperature or HT (Supplementary Fig. 10a, c), suggesting that TCPs may not regulate BEL1 at the transcriptional level. To test whether TCPs regulate BEL1 at the protein level, we first assessed the interaction between BEL1 and TCP4 using yeast two hybrid assays, revealing that BEL1 clearly interacted with TCP4 (Fig. 6a). We confirmed the association of TCP4 with BEL1 in vivo using LCI and Co-IP experiments (Fig. 6b, c). Since TCP4 interacted with both BEL1 and AG-SEP3, we hypothesized that TCP4 could strengthen the interaction between BEL1 and AG-SEP3. To test this hypothesis, we fused the N-terminus of LUC (nLUC) with AG, while the C-terminus of LUC (cLUC) was fused with BEL1. Strong fluorescence was observed following co-expression of AG-nLUC, 35S-SEP3 and BEL1-cLUC under normal temperature, whereas no fluorescence was produced by the control combinations (Fig. 6d), consistent with previous results showing that BEL1 interacts with the AG-SEP3 complex[13]. When we co-expressed the miRNA319-resistant mTCP4-MYC with AG-nLUC, 35S-SEP3 and BEL1-cLUC, the fluorescence was significantly increased in three replications (Fig. 6d, f and Supplementary Fig. 12a, b), suggesting that TCP4 may reinforce the association of BEL1 and AG-SEP3. Our results indicated that HT is an important environmental factor promoting the homeotic conversion of ovules into carpelloid structures in tcpDUO (Fig. 3). Thus, we speculated that the interaction between BEL1 and AG-SEP3 could be compromised to release the activity of AG-SEP3 under HT. Indeed, the HT treatment significantly weakened the interaction between BEL1 and AG-SEP3 (Fig. 6e, f and Supplementary Fig. 12c, d).

To further reveal the molecular mechanism by which HT weakened the interaction between BEL1 and AG-SEP3, we investigated the stability of BEL1 under HT, because the transcript level of BEL1 was not significantly altered by HT treatment (Supplementary Fig. 10a, c). We generated 35S-BEL1-FLAG-P2A-GFP, in which expression of BEL1-FLAG and GFP (connected by a sequence encoding P2A) was driven by a CaMV 35 S promoter. P2A is a sequence from porcine teschovirus-1. The 2A peptide mediates self-cleavage, and thus results in one mRNA containing a P2A sequence to be translated into two separate proteins[51,52]. Because P2A self-cleavage caused the expressed BEL1-FLAG-P2A-GFP to be cleaved into equal amounts of BEL1-FLAG and GFP, the abundance of GFP served as the internal control for the evaluation of BEL1-FLAG stability. We transiently expressed 35S-BEL1-FLAG-P2A-GFP in tobacco leaves and treated the leaves with HT. The results showed that the abundance of both the uncleaved BEL1-FLAG-P2A-GFP product and BEL1-FLAG cleaved from it decreased after HT treatment, while the abundance of GFP cleaved from the fusion protein was not altered, indicating that BEL1 was unstable under HT (Fig. 6g). To determine whether the stability of BEL1 under HT was dependent on the 26S proteasome, we treated the samples with MG132 (a proteasome inhibitor) and HT. The degradation of BEL1 under HT was highly inhibited by MG132, suggesting that the 26S proteasome was required for the degradation of BEL1 (Fig. 6h). To provide more evidences to determine the stability of BEL1 under HT, we generated 35Spro-BEL1-FLAG in which BEL1 fusion with the sequence encoding FLAG tag was driven by a CaMV 35S promoter. We treated the flowers of 35Spro-BEL1-FLAG transgenic plants with a protein synthesis inhibitor cycloheximide (CHX). The results showed that HT treatment led to the obvious decrease of BEL1, while the proteasome inhibitor MG132 inhibited the degradation (Fig. 6i), further confirming that BEL1 was regulated by the degradation via 26 S proteasome under HT. We further co-expressed miR319-resistant mTCP4 with 35S-BEL1-FLAG-P2A-GFP and found that TCP4 might increase the stability of BEL1 under HT

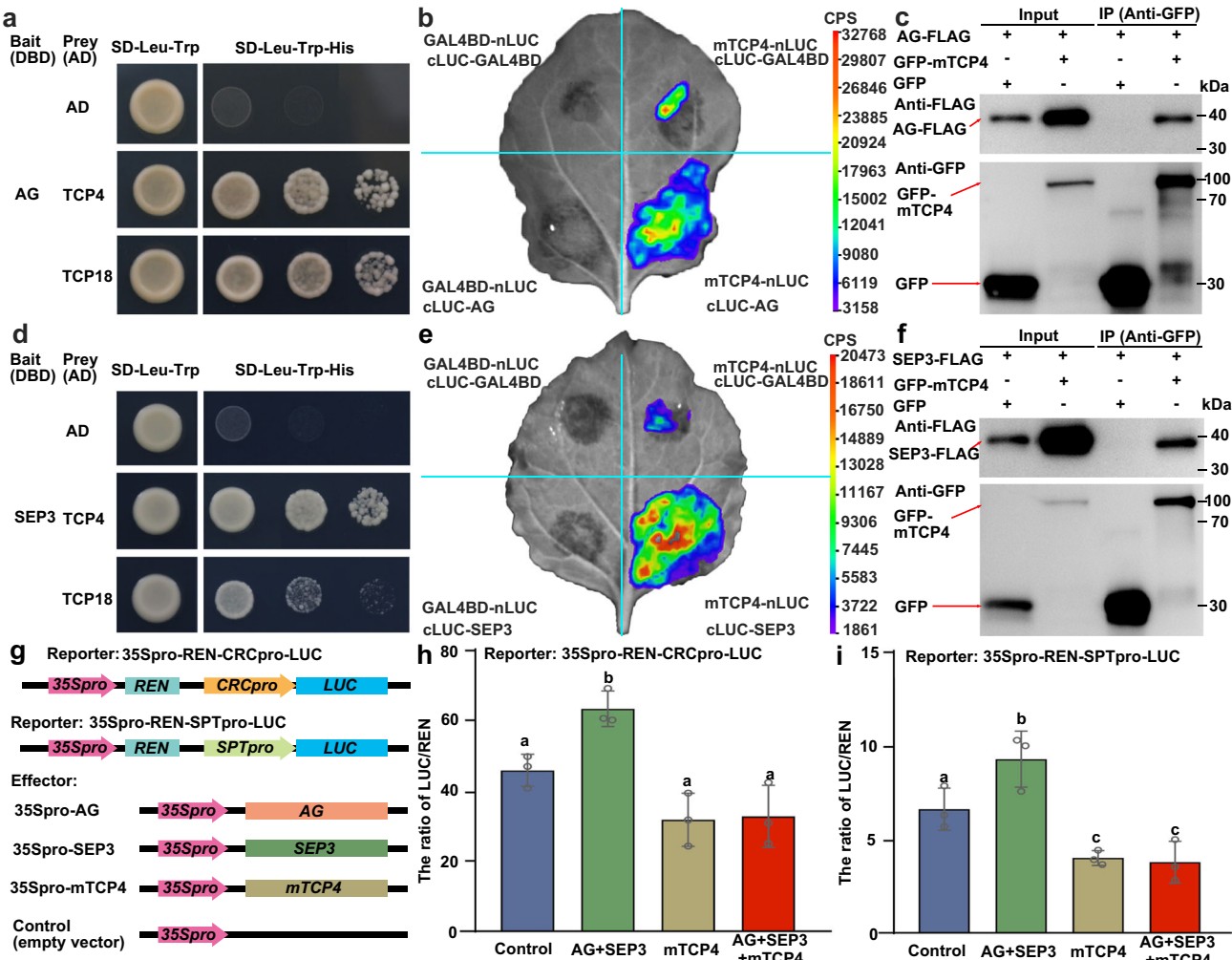

**Fig. 5 | TCPs directly interacted with both AG and SEP3 to repress the trans-activation activity of the AG-SEP3 complex. a** The CIN-like TCP4 and the CYC-like TCP18 interacted with AG using yeast two-hybrid assays. The transformed yeast were spotted on the selection SD-Leu-Trp-His medium with 10-, 100-, and 1,000-fold dilutions. 2.5 mM 3-amino-1,2,4-trizole was supplemented to inhibit the self-activation activity of AG. Full-length AG was used as bait. AD, activation domain; DBD, DNA-binding domain. **b** TCP4 interacted with AG in vivo as shown by LCI in tobacco leaves. The miR319-resistant TCP4 protein was fused with nLUC at the C-terminus, and AG was fused with cLUC at the N-terminus. nLUC and cLUC fused with GAL4BD were used as negative controls. **c** Co-IP confirmed that TCP4 interacted with AG. 35Spro-GFP-mTCP4 or the control 35Spro-GFP was co-transformed into Arabidopsis protoplasts with 35Spro-AG-FLAG. The extracted proteins were immunoprecipitated with GFP-trap Agarose beads and detected by western blots with antibodies against GFP or FLAG. **d** TCP4 and TCP18 interacted with SEP3 using yeast two hybrid assays. The full-length SEP3 was used as the bait. The selection conditions were as described above. **e** LCI analysis showed that TCP4 interacted with SEP3. SEP3 was fused with cLUC at the N-terminus. **f** Co-IP confirmed that TCP4 interacted with SEP3. 35Spro-GFP-mTCP4 or the control 35Spro-GFP was co-transformed into Arabidopsis protoplasts with 35Spro-SEP3-FLAG. **g** The schematic diagram of the constructs used for the transient expression assays. **h, i** The trans-activation activity of the AG-SEP3 complex was significantly suppressed by co-expressing TCP4 using the reporter 35Spro-REN-CRCpro-LUC (**h**) or 35Spro-REN-SPTpro-LUC (**i**) using a transient expression system in Arabidopsis protoplasts. Source data for **c, f, h,** and **i** are provided as a Source Data file. The experiments of **c** and **f** were repeated at least three times, with at least three independent biological replicates. Significant differences are indicated by different letters (*P* < 0.05). Three biologically independent experiments were performed. Data are presented as mean values ± SEM. One-way two-sided ANOVA with post hoc Tukey HSD tests was used.

(Fig. 6j), consistent with the results showing that TCP4 promoted the interaction between BEL1 and AG-SEP3.

These results suggest that TCP transcription factors may promote ovule identity by directly interacting with BEL1 to prevent degradation of BEL1 by the 26 S proteasome under HT.

### TCP transcription factors synergistically interact with BEL1 and STK

Our biochemical analysis indicated that TCP transcription factors and BEL1 directly interacted with each other. In addition, a previous report and our data showed that TCPs and BEL1 both repressed the activity of the AG-SEP3 complex by directly interacting with AG-SEP3[13]. Disruption of TCPs or BEL1 led to temperature-dependent aberrations in the

establishment of ovule identity[25], suggesting that TCPs and BEL1 may play synergistic roles in maintaining ovule identity. To assess this possibility, we used CRISPR/Cas9 technology to mutate *BEL1* in the wild-type and *tcpDUO* backgrounds. We obtained the *bel1-1*, *bel1-8*, *bel1-11,* and *bel1-15* lines, with one-bp-insertion, a 455 bp, 454 bp, or 460 bp deletion, respectively, in the first exon of *BEL1* in the wild-type and *tcpDUO* backgrounds (Fig. 7a). The ovule phenotypes of the mutant lines were observed under normal temperature using SEM. The results showed that the wild-type control generated normal ovules (Fig. 7c), while *bel1* mutant produced bell-shaped ovules absent of inner integuments as previously reported (Fig. 7e, f)[25]. Both the single mutant and *tcpDUO* rarely displayed homeotic ovule conversion into carpelloid structures under normal temperature (Fig. 7d–f), whereas

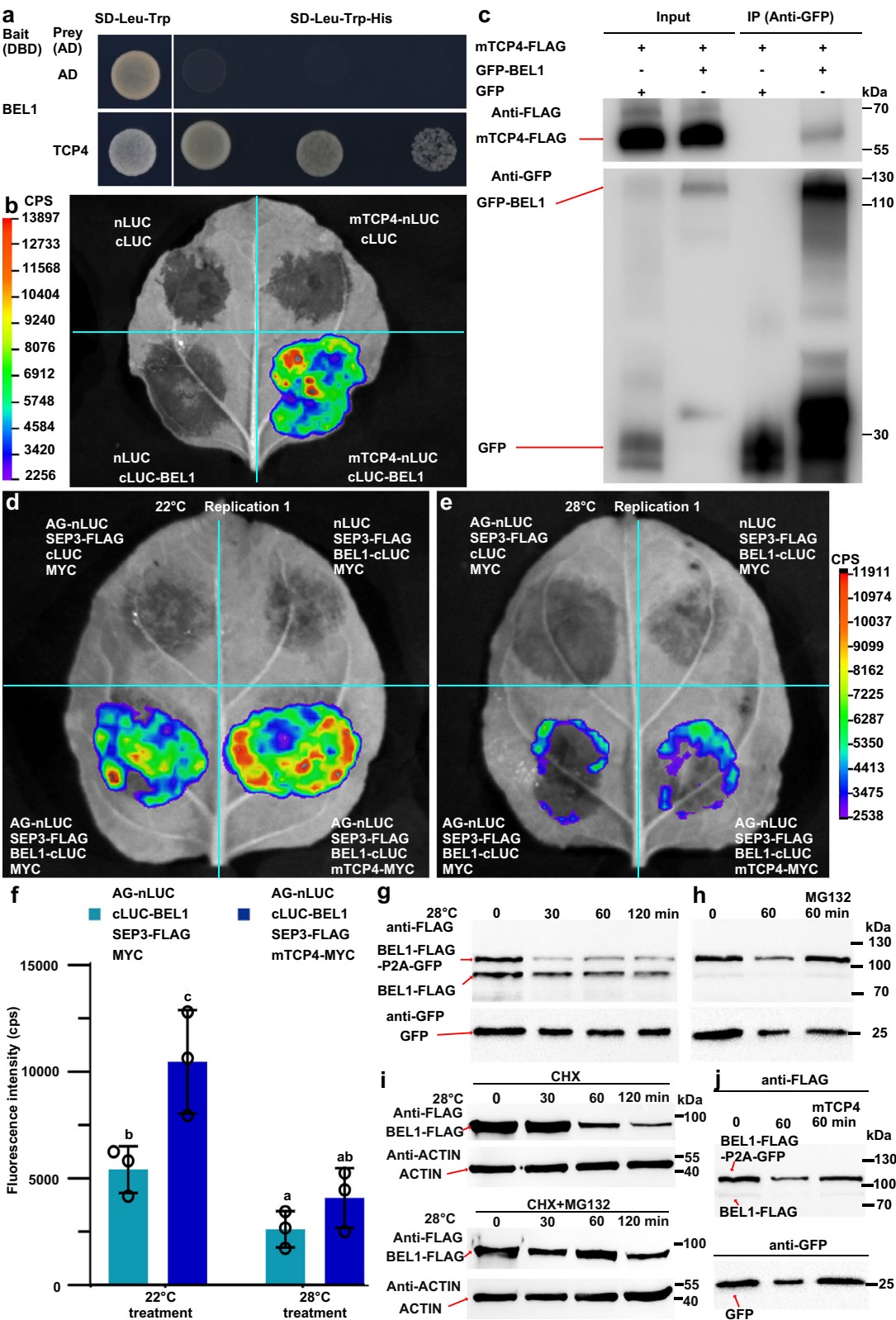

all of the ovules of the tredecuple mutant *tcpDUO bel1* turned into carpelloid structures with papilla cells and long funiculi under normal temperature (Fig. 7g–j). In the carpelloid structures of *tcpDUO bel1*, many ovule-like structures were initiated from the chalaza position (Fig. 7h–j). The integuments of the ovule-like structures continued to grow and were then converted into carpelloid structures with papilla cells (Fig. 7h, i). The carpelloid structures showed an indeterminate growth, because the new carpel-like structures continued to be produced in the inner of the old ones (Fig. 7h–j). The striking phenotypes of *tcpDUO bel1* demonstrated that TCP transcription factors play essential roles in determining ovule identity in cooperation with BEL1.

The results described above indicated that TCP transcription factors repress the activity of AG-SEPs complexes in ovules, tilting the balance toward the activity of AG-SEPs-STK/SHPs, thus promoting

**Fig. 6 | TCP4 interacted with BEL1 to inhibit the degradation of BEL1 and enhance the association of BEL1 with the AG-SEP3 complex. a** TCP4 interacted with BEL1 using yeast two-hybrid assays. The transformed yeasts were grown on selection SD-Leu-Trp-His medium with 10-, 100-, and 1,000-fold dilutions. The full-length BEL1 was used as the bait. 2.5 mM 3-AT was added to inhibit the self-activation activity of BEL1. AD, activation domain; DBD, DNA-binding domain. **b** LCI assays indicated that TCP4 interacted with BEL1 in vivo. **c** Co-IP experiments confirmed that TCP4 interacted with BEL1 in vivo. 35Spro-GFP-BEL1 or the control 35Spro-GFP was co-transformed into tobacco leaves with 35Spro-mTCP4-FLAG. The extracted proteins were immunoprecipitated with GFP-trap Agarose beads and detected by western blotting with antibodies against GFP or FLAG. The experiments were repeated at least three times, with at least three independent biological replicates. **d** The association of BEL1 with AG-SEP3 was enhanced by the co-expression of miR319-resistant *mTCP4* under 22 °C using a transient expression system in tobacco leaves. **e** The association of BEL1 with AG-SEP3 was obviously attenuated under HT. **f** The quantitative analysis of fluorescence in (**e**, **f**) and the replications in Supplementary Fig. 12. Three biologically independent experiments were performed. Data are presented as mean values ± SEM. Statistical significance was determined by one-way ANOVA. Significant differences at $P < 0.05$ are indicated by different lowercase letters. **g** The expression of BEL1-FLAG-P2A-GFP in tobacco leaves showed that BEL1-FLAG was unstable under HT. **h** The treatment with MG132 inhibited the degradation of BEL1-FLAG-P2A-GFP and BEL1-FLAG expressed in tobacco leaves, indicating that the degradation of BEL1 was dependent on 26 S proteasome was required for the degradation of BEL1 under HT. **i** The CHX or CHX + MG132 treatment of inflorescence tissues from 35Spro-BEL1-FLAG transgenic plants showed that BEL1 was regulated by degradation via 26 S proteasome. **j** The degradation of BEL1 might be inhibited by the co-expression of miR319-resistant *mTCP4* with BEL1-FLAG-P2A-GFP in tobacco leaves under HT. Source data are provided as a Source Data file. The experiments of **g**–**j** were repeated at least three times, with at least three independent biological replicates.

ovule identity. Accordingly, the decreased activity of AG-SEPs-STK/SHPs in *tcpDUO* could synergistically interact with the increased activity of AG-SEPs. To test this hypothesis, we generated the tredecuple mutant *tcpDUO stk*, with 1651 bp and 871 bp deletions between the second and third exons of *STK*, in the *tcpDUO* and wild-type backgrounds (Fig. 7b). The SEM observations indicated that the *stk* mutant formed ovules with longer funiculi as previously reported[20] (Fig. 7k). However, the carpels of the tredecuple mutant *tcpDUO stk* were full of carpelloid structures instead of ovules (Fig. 7l–n). The carpel-like structures were connected by even longer funiculi (Fig. 7l, n). In addition, stigmatic papilla cells were clearly observed at the tip of the carpelloid structures (Fig. 7l–n). Nevertheless, no ovule-like structures or secondary carpel-like structures were observed in *tcpDUO stk* (Fig. 7l–n).

The ovule defective phenotypes of *tcpDUO bel1* and *tcpDUO stk* unequivocally demonstrate that TCP transcription factors play a critical role in maintaining the identity of ovules.

## Discussion

In this study, we demonstrated that plant-specific TCP transcription factors play critical and highly redundant roles in the control of ovule development and identity. The generation of a high-order *tcpDUO* mutant, in which both *CIN*- and *CYC*-like *TCPs* were disrupted, revealed that the two clades of TCPs were redundantly pivotal for ovule development. Our observations of the conversion of ovules into carpelloid structures in *tcpDUO* under HT indicated that TCPs have highly redundant functions in protecting the stability of ovule identity under HT. We propose a seesaw working model for the function of TCPs in determining ovule development and identity. In wild-type ovules, AG, SEPs, STK and SHPs co-exist in ovules. The ovules are specified by AG-SEPs-STK/SHPs, while the carpels are specified by AG-SEPs. The balance between the two kinds of complexes is important for the specification of ovule identity[13–15]. Under normal temperature (22 °C), TCPs interacts with BEL1 and AG-SEPs to inhibit the activity of AG-SEPs in wild type. The inhibition of AG-SEPs activity leads to the relatively higher activity of AG-SEPs-STK/SHPs which promotes the expression of ovule-related genes to determine ovule identity (Fig. 8a). In wild-type ovules under HT (28°C), BEL1 degradation induced by HT releases the activity of AG-SEPs, but TCPs and the remaining BEL1 (protected by TCPs) are sufficient to inhibit the activity of AG-SEPs. TCPs also promote the expression of *STK* to prevent the conversion of ovules into carpelloid structures under HT (Fig. 8b). In *tcpDUO* ovules under 22 °C, the loss of TCP function increases the activity of AG-SEPs, but the stable BEL1 under 22 °C inhibits the activity of AG-SEPs; therefore, ovule development is affected, but only a very small proportion of ovules are converted into carpelloid structures (Fig. 8c). However, in the *tcpDUO* ovules under HT, the loss of TCP function and HT-induced degradation of BEL1 release the activity of AG-SEPs, the expression of *STK* is decreased without TCPs and the predominant activity of AG-

SEPs over AG-SEPs-STK/SHPs promotes the expression of carpel-related genes and the generation of carpelloid structures (Fig. 8d). In the tredecuple mutant *tcpDUO bel1*, because the inhibitory influences of both TCPs and BEL1 are lost, the released activity of AG-SEPs causes all of the ovules of *tcpDUO bel1* to be converted into carpelloid structures, even under normal temperature (Fig. 8e). In the tredecuple mutant *tcpDUO stk*, the loss of *STK* function reduces the activity of AG-SEPs-SHPs, while the loss of function of *TCP* released the activity of AG-SEPs; therefore, the relative higher activity of AG-SEPs promotes the formation of carpelloid structures instead of ovules in the carpels of *tcpDUO stk* under normal temperature (Fig. 8f). Although all the ovules are converted into carpelloid structures in the carpels of *tcpDUO bel1* and *tcpDUO stk*, there are differences between them. *tcpDUO bel1* produced ovule-like structures in the carpelloid structures which displayed indeterminate growth, while *tcpDUO stk* produced even longer funiculi. These differences are consistent with the distinct functions between BEL1 and STK. It has been reported that BEL1 plays a critical role in repressing the formation of ectopic nucelli and the homeotic conversions of ovule integuments into carpelloid structures by regulating the proper expression of *PIN1* and *WUS*[53]. In this paper, we found that the expression of *WUS* was highly up-regulated in *tcpDUO*, indicating TCPs are also important for the regulation of *WUS*. However, *STK* was reported to suppress the elongation of funiculi and to maintain the identity of ovules[22]. Our data demonstrate that CIN- and CYC-like TCPs redundantly act as critical factors in determining ovule identity, while important environmental factors such as HT can reduce the stability of ovule identity, and TCPs act with BEL1 to stabilize ovule identity and antagonize the effects of HT.

The emergence of ovules was the most important morphological innovation for the fertility, dispersal, and survival of seed plants[54,55]. The determination of ovule identity is a fundamental step in ovule development. During the past few decades, factors that play a role in determining ovule identity have been identified. Genetic analysis demonstrated the roles of D-function MADS box proteins in specifying ovule identity, including STK, SHP1, and SHP2, in the widely accepted ABCDE model of floral organ specification[22]. In addition, BEL1 homeodomain protein was also shown to play an essential role in specifying ovule identity[56–58]. The double mutant *shp1 shp2* produces rather normal ovules, and the single mutant *stk* generates ovules with longer funiculi and normal identity[22,24]. However, the triple mutant *stk shp1 shp2* displays severe defects in the production of ovules[22]. At the early developmental stage, the ovules of *stk shp1 shp2* have no obvious differences from those of wild-type plants; megasporogenesis proceeds normally and the two integuments are initiated in the ovule of the triple mutant[13]. Nevertheless, the integuments of *stk shp1 shp2* grow aberrantly and are converted into a carpelloid structure during the late developmental stage[13,22]. A similar progression was observed in the *bel1* mutant line[13,25]. The process of megasporogenesis by *bel1* is rather normal, and the megaspore is formed in the ovules[23,56].

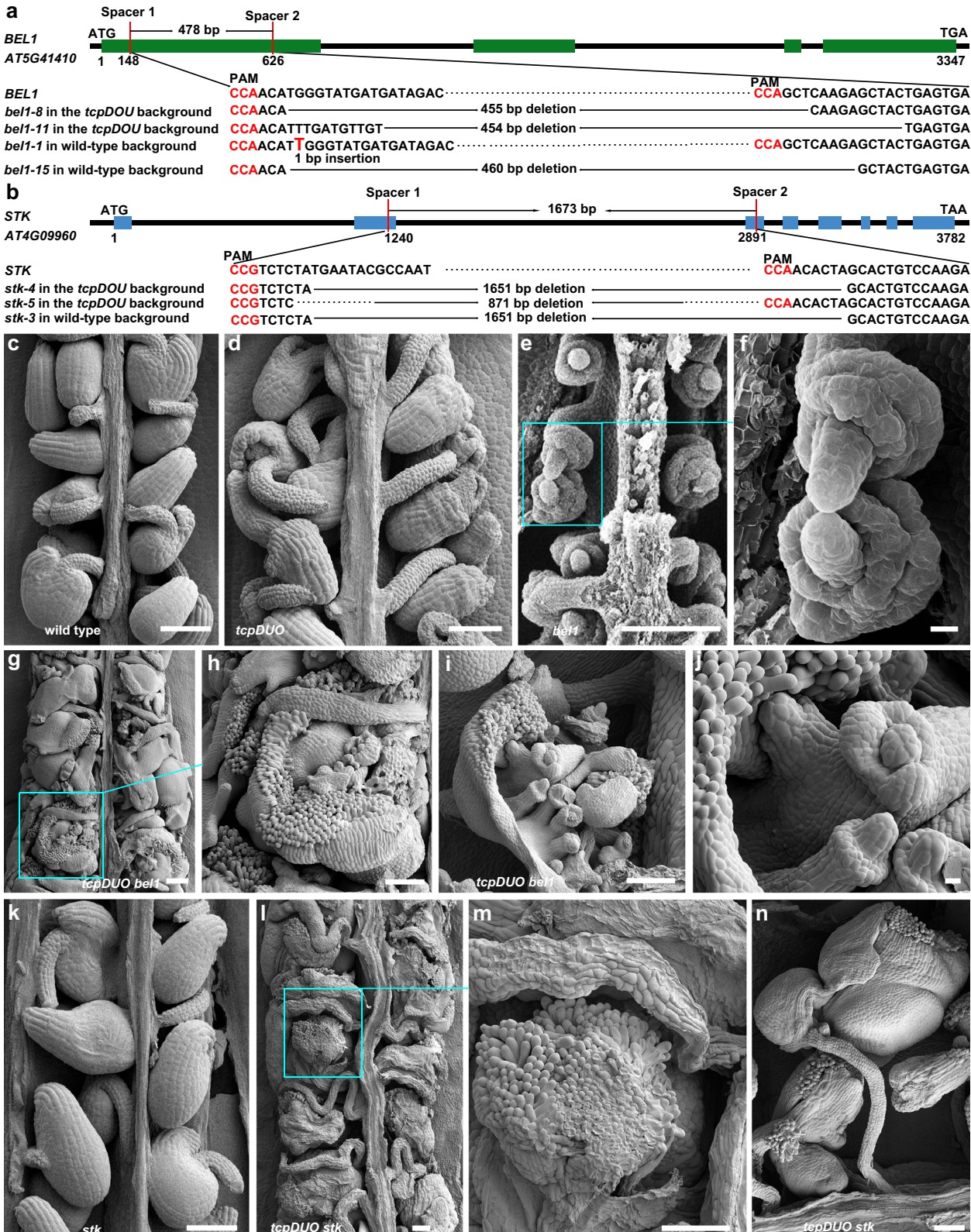

**Fig. 7 | The genetic analysis between *tcpDUO* and *bel1* or *stk*. a** Schematic diagram showing the deletions and one-bp insertion created by CRISPR/Cas9 in *bel1* mutants in the wild-type and *tcpDUO* backgrounds. **b** Schematic diagram showing the deletions created by CRISPR/Cas9 in *stk* mutants in the wild-type and *tcpDUO* backgrounds. The green (**a**) and blue (**b**) squares indicate exons, and the black lines represent introns. **c–n** SEM analysis of ovules or carpelloid structures from wild-type and mutants under normal temperature (22 °C). **c–f** Mature ovules of wild-type (**c**), *tcpDUO* (**d**) and *bel1* (**e**, **f**). **g** All ovules were converted into carpelloid structures in the tredecuple mutant *tcpDUO bel1*. **h–j** Close-up views of the carpelloid structures in *tcpDUO bel1*. The papilla cells and secondary ovules were clearly observed in the carpelloid structures. **k** Mature ovules of *stk*. **l, m** Carpelloid structures in the tredecuple mutant *tcpDUO stk*. Even longer funiculi and clear papilla cells were observed. Images of **c–n** are representatives of at least ten independent individuals. Scale bars, 100 μm in **c–e, g–i**, and **k–m**, 10 μm in **f** and **k**.

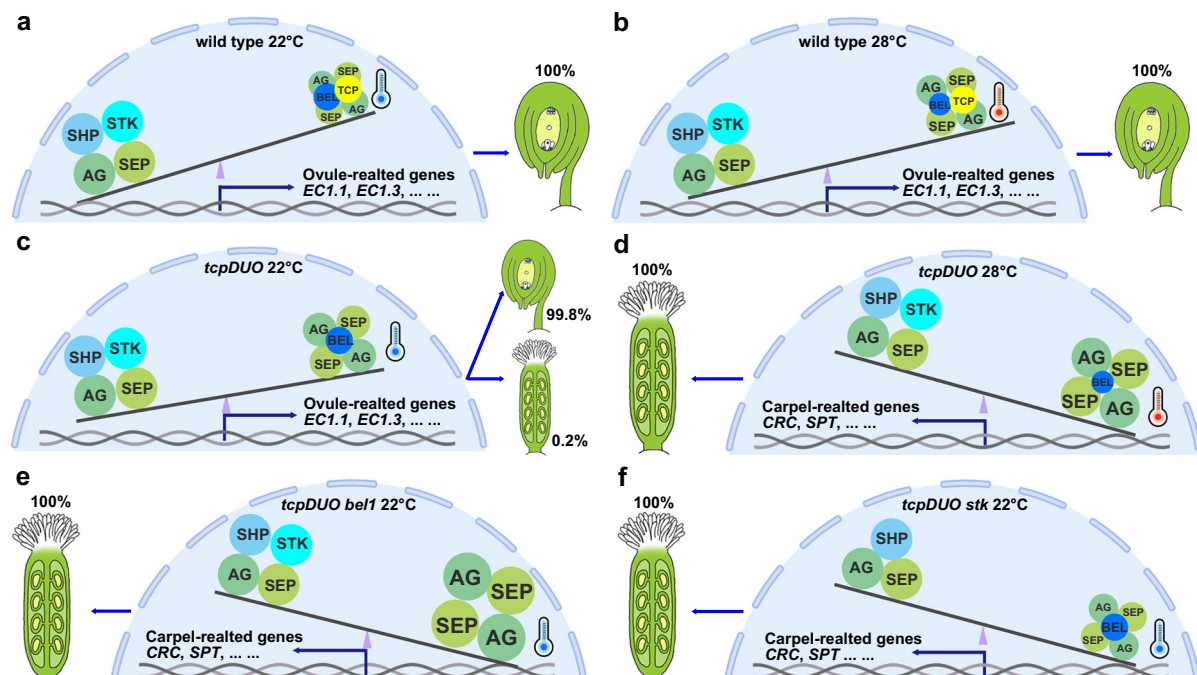

**Fig. 8 | Seesaw working model of TCP transcription factors in the control of ovule identity. a** Ovule development in wild-type under 22 °C. The activity of AG-SEPs-STK/SHPs was predominant due to the repressive effects of TCPs and BEL1 on the activity of AG-SEPs. **b** Ovule development in wild-type under 28 °C. Although the degradation of BEL1 was induced by HT, TCPs and the remaining BEL1 were sufficient to repress the activity of AG-SEP quartets. The activity of AG-SEPs-STK/SHPs quartets was still predominant. **c** Ovule development in *tcpDUO* under 22 °C. Without the repressive effects of TCPs on AG-SEP quartets, the relatively stable BEL1 protein only slightly inhibited the activity of AG-SEP quartets, causing a low proportion of carpelloid structures to be derived from ovule conversion in *tcpDUO*. **d** Ovule development in *tcpDUO* under 28 °C. Without the repressive effects of

TCPs on AG-SEP quartets, degradation of BEL1 was not sufficient to inhibit the activity of AG-SEP quartets, causing the conversion of all ovules into carpelloid structures in *tcpDUO*. **e** Ovule development in the *tcpDUO bel1* tredecuple mutant. Without the repressive effects of both TCPs and BEL1 on AG-SEP quartets, all ovules in *tcpDUO bel1* were converted into carpelloid structures under 22 °C. **f** Ovule development in the *tcpDUO stk* tredecuple mutant. Without the repressive effects of TCPs on AG-SEP quartets, the relatively stable BEL1 protein under 22 °C was barely sufficient to inhibit the activity of AG-SEP quartets. However, the activity of AG-SEPs-SHPs was low without *STK* function, causing the relative predominance of AG-SEP quartets and carpelloid structures in the tredecuple mutant *tcpDUO stk*.

However, the conversion of ovules into carpelloid structures is also observed in *bel1* at the late developmental stage of ovules[25]. In this study, we showed that CIN- and CYC-like TCP transcription factors have highly redundant functions in the control of ovule identity. The *tcpDUO* mutant line displays several phenotypes similar to those of *stk, stk shp1 shp2,* and *bel1*. First, *tcpDUO* produces longer funiculi, similar to *stk* and *stk shp1 shp2*. Second, megasporogenesis proceeds rather normally in *tcpDUO*, whereas megagametogenesis proceeds abnormally, as in *stk shp1 shp2* and *bel1*. Third, *tcpDUO*, *stk shp1 shp2* and *bel1* all produce carpelloid structures instead of ovules. These similar phenotypes suggest that TCP transcription factors function in the same pathway with STK/SHPs and BEL1 in the control of ovule development and identity. Indeed, our biochemical evidence demonstrates that TCP4 interacts with AG-SEP3 and BEL1. The interaction of AG-SEP3 with TCP4 represses the transactivation activity of the AG-SEP3 complex, while the interaction of TCP4 and BEL1 increases the stability of BEL1 and thus stabilizes the interaction between BEL1 and AG-SEP3 to enhance the suppressive effect of BEL1 on the function of AG-SEP complexes. In addition, because TCP4 directly binds to the promoter region of carpel-related genes such as *CRC*, *SPT* and *HEC1*, TCP4 may also act as transcription factor to inhibit these genes and to prevent the carpelloid conversion during ovule development by recruiting the transcriptional repressor TCP INTERACTOR CONTAINING EAR MOTIF PROTEIN 1 (TIE1) and the corepressor TPL/TPRs[59].

Temperature is one of the most important environmental conditions because it affects the development, reproduction, geological distribution and survival of organisms[1,60]. The elevated temperature caused by global warming has brought severe challenges to the distribution and survival of many plants and animals[1]. HT affects not only

the progress of development but also the program of development, in organisms. In plants, thermomorphogenesis has been coined to describe a series of morphological changes, including elongated petioles and hypocotyls, which occur under mildly elevated temperatures[1]. Using elongated hypocotyls under HT as an output trait, many factors have been shown to play roles in the control of plant thermomorphogenesis[1,39,61,62]. However, the mechanisms by which the program of plant organ identity are affected by HT remain largely unknown. It is well established that homeotic conversion between male and female organ identity results in the determination of sexual organ identity under low or high temperatures for some animals, including turtles[7]. For example, the eggs of the turtle *T. scripta* all develop into female individuals under a high hatching temperature (32 °C), whereas they develop into male individuals under a relatively low hatching temperature (26 °C)[7]. The deposition of trimethylated histone 3 at lysine residue 27 (H3K27me3) in the promoter of *Doublesex And Mab-3 Related Transcription Factor 1* (*DMRT1*), which is crucial for male sex determination, is affected by temperature, leading to different expression levels of *DMRT1* under low or high temperatures[7]. In plants, the homeotic conversion of ovules into carpelloid structures is also affected by temperature in the *bel1* mutant line[25]. The relative high temperatures favor the conversion of ovules into carpelloid structures[25]. However, the molecular mechanisms by which HT reduces the stability of ovule identity remain unknown. In this study, we found that HT strongly favors the conversion of ovules into carpelloid structures in *tcpDUO*. At normal temperature (22 °C), only 0.2% of ovules are transformed into carpelloid structures in *tcpDUO*, while HT (28 °C) treatment increases the rate of carpelloid structure conversion to 100%, indicating that ovule identity was less

stable under HT, as observed in *bel1*. Our biochemical data showed that TCP4 interacted with BEL1, and both TCP4 and BEL1 repressed the expression of carpel identity genes. Furthermore, we show that HT promotes the degradation of BEL1 via the 26 S proteasome, and TCP4 may stabilize BEL1 protein to facilitate the BEL1 inhibition of AG-SEPs in repressing the specification of carpel identity. This finding suggests that plants protect ovule identity by stabilizing a key regulator of organ identity at the protein level under HT, in contrast to the epigenetic modification of a gene locus crucial for reproductive organ identity under low or high temperatures in animals[7].

TCP transcription factors are grouped into Class I and II subfamilies based on the TCP domain, and Class II TCPs are further classified in two clades: CIN-like TCPs and CYC-like TCPs[28]. The *CIN* gene was first identified in snapdragon (*Antirrhinum majus*) by analyzing the *cin* mutant line, which produces wavy leaves with aberrant flatness[63]; thus, CIN-like TCPs are mainly characterized as central regulators in the control of leaf morphology in diverse plant species[31,32]. The *CYC* gene is a foundational member of the TCP family and was also first identified in snapdragon[64]. Disruption of *CYC* and its close homolog *DICHOTOMA* (*DICH*) converts snapdragon flower symmetry from bilateral to radial symmetry. The other foundational member of the TCP family is *TEOSINTE BRANCHED1* (*TB1*), which also belongs to the CYC-like TCP clade and is an important domesticated gene controlling shoot branching in maize (*Zea mays*)[65,66]. CYC-like TCPs have been identified mainly as regulators in the control of shoot branching and flower development. Both CIN-like TCPs and CYC-like TCPs have redundant functions in regulating plant development[26,38]. We previously reported that CIN-like TCPs play redundant roles in the repression of green petals and trichome formation[38,67]. The potential redundancy of the functions of CIN-like and CYC-like TCPs has not been investigated due to the difficulty associated with generating high-order multiple *tcp* mutants. In this study, we succeeded in generating the duodecuple mutant *tcpDUO* line by combining twelve mutations that disrupt both CIN-like and CYC-like TCPs. We revealed that CIN- and CYC-like TCPs redundantly control ovule identity. The functions of Class II TCPs in determining ovule identity have been masked by high cross-protein redundancy and thus have not been identified in previous studies. In addition to the homeotic transformation of ovules into carpelloid structures in *tcpDUO*, this mutant line also displays some other interesting phenotypes, which could be valuable subjects for experiments aimed at identifying additional functions of Class II TCPs. Our findings reveal an important area of function for Class II TCP transcription factors and expand our understanding of the molecular mechanisms underlying the determination of ovule identity.

## Methods

### Plant materials and growth conditions

The Arabidopsis Columbia-0 (Col-0) ecotype was used for wild-type and mutant lines in this study. The *tcpDUO* duodecuple mutant was generated on the basis of the *tcpSEP* septuple mutant by genetic crosses. The *tcp12/brc2* (SALK_023116), *tcp16* (GABI_655C10), and *tcp18/brc1* (SALK_091920) mutants are T-DNA insertion mutants from public databases. The *tcp1* and *tcp24* mutants were generated by CRISPR/Cas9 technology. Arabidopsis seeds were sterilized with 75% ethanol for 10 min and were then grown on half-strength Murashige and Skoog (1/2MS) solid medium containing 10 g/L sucrose. Seed stratification was carried out at 4 °C for 48 h. The plates were then transferred to 22 °C chambers with a long-day condition (16 h light/8 h dark, 170 mmol/m2/s light intensity). The seedlings were transferred into soil six days after germination and grown in a greenhouse under the conditions described above.

For observation of the HT-induced ovule homeotic conversion phenotype of *tcpDUO*, bolting plants were transferred to 28 °C

chambers with a long-day condition for 6 days. Then the flowers were emasculated at developmental stage 9. After two days, the ovules, the carpelloid structures or pistils were isolated for observation.

Tobacco (*Nicotiana benthamiana*) seeds were sown in soil, and tobacco plants were grown for the transient expression experiments in the same greenhouse.

### Genotyping and gene expression analysis

For genotype analysis of the T-DNA insertion *tcp* mutants, including *tcp2*, *tcp3*, *tcp4*, *tcp5*, *tcp10*, *tcp12*, *tcp13*, *tcp16*, *tcp17*, and *tcp18*, the fragments from wild-type loci were amplified with the TCPx-G-F and TCPx-G-R primer pairs, while the fragments from the T-DNA insertion sites of mutants were amplified with the TCPx-T and TCPx-G-R or TCPx-G-F primer pairs.

For genotyping of the mutants generated by CRISPR/Cas9, including *tcp1*, *tcp24*, *bel1*, and *stk*, the genomic fragments were amplified by the corresponding Genotyping-F and Genotyping-R primer pairs. The mutations were identified by sequencing.

For analysis of gene expression, the tissue samples, including pistils, ovules or carpelloid structures, were dissected from the flowers of wild-type or *tcpDUO* mutants. The tissues were immediately put in liquid nitrogen after dissection. Total RNA was isolated with a Plant Total RNA Purification Kit (GeneMark, TR02-150), and was then reverse-transcribed with an M-MLV kit (Promega, M170A). The products were diluted and then used as the templates for RT-qPCR with three repeats. UltraSYBR Mixture (CWBIO, CW2601M) and a Thermo-Fisher QuantStudio 5 real-time system were used for RT-qPCR. The cycling conditions were 94 °C for 30 s, 60 °C for 30 s and 72 °C for 30 s. The relative expression levels of genes were calculated based on the $2^{-\triangle\triangle CT}$ (cycle threshold) method[68]. The gene expression levels were normalized based on the expression level of *ACT7*.

The primers used in this study are listed in Supplementary Table 1.

### Generation of binary constructs and transformation

For complementation analysis, the 2818 bp genomic sequence containing the *TCP4* promoter and *TCP4* coding region without the stop codon was amplified and cloned into the pENTR/D TOPO vector (Invitrogen) to generate pENTR-TCP4pg as previously described[38]. The TCP4pro-TCP4-GFP construct was generated by LR reaction between pENTR-TCP4pg and pK7FWG0 (Invitrogen).

For transient expression in tobacco leaves or yeast two hybrid assays, the coding regions of *TCP4*, *AG*, *SEP3*, and *BEL1* were amplified from Arabidopsis inflorescence cDNA and cloned into the pENTR/D TOPO vector to generate pENTR-TCP4, pENTR-AG, pENTR-SEP3, and pENTR-BEL1. The miR319-resistant *TCP4* (*mTCP4*) was amplified by PCR-based mutagenesis using the primer pair TCP4-mut-F/TCP4-mut-R as described previously[59], and then cloned into pENTR/D TOPO to generate pENTR-mTCP4. 35Spro-GFP-mTCP4 and 35Spro-GFP-BEL1 were generated by LR reaction between pENTR-mTCP4 or pENTR-BEL1, respectively, and pK7WGF2 (Ghent University). The 35Spro-AG-FLAG and 35Spro-SEP3-FLAG constructs were generated by LR reaction between pENTR-AG or pENTR-SEP3, respectively, and pB7FLAGWG2.

The constructs were introduced into *Agrobacterium tumefaciens* strain GV3101 to allow the generation of Arabidopsis transgenic lines using the floral dip method or for transient expression in tobacco leaves[38]. All primers are listed in Supplementary table 1.

### Microscopy analysis

For DIC observation of cleared ovules, the pistils were dissected and the ovules were separated in ovule clearing solution (8 g chloral hydrate, 1 mL 100% glycerol and 3 mL ddH$_2$O) for several seconds, after which they were observed under a Zeiss AX10 Imager M2 microscope.

Ovules were observed using a confocal laser scanning microscope (CLSM). Briefly, the inflorescences were fixed in 4% glutaraldehyde (in 12.5 mM cacodylate, pH 6.9) and vacuumed twice for 1 h. The tissues

were fixed overnight at 4 °C. Samples were then dehydrated through a conventional ethanol series (15%, 30%, 45%, 50%, 60%, 75%, 85%, and 95%) for 30 min per step and cleared in 2:1 (v/v) benzyl benzoate:benzyl alcohol overnight. Pistils were dissected and mounted with immersion oil. A confocal laser scanning microscope (LSM710, Zeiss) was used for the observation of ovules.

For SEM observation, the samples were prepared as described in the user's manual of the FEI Helios Nanolab G3 UC. The pictures were captured by an EasyLift EX NaonoManipulator and Auto Slice and View G3 software.

### Yeast two-hybrid assays

For testing the interaction between TCP proteins with MADS-box proteins and BEL1 in yeasts, the bait constructs were generated by LR reactions between pENTR-AG/SEP3/STK/BEL1 and pDEST32 (Invitrogen). The prey constructs were generated by LR reactions between pENTR-TCP4/TCP16/TCP18 and pDEST22 (Invitrogen). For the truncation analysis of SEP3 and AG, the proteins were truncated according to the conserved sequences of the MADS-box domain, intervening domain, keratin domain, and C terminus. The primers used for truncation analysis are listed in Supplementary Table 1.

The bait and prey constructs were co-transformed into *Saccharomyces cerevisiae* strain AH109. The blank pDEST22 was co-transformed with each prey as a negative control. Medium supplemented with SD-Leu-Trp-His with or without 2.5 mM 3-amino-1,2,4-triazole (3-AT) was used for yeast selection.

### Firefly luciferase complementation assays

To perform the firefly luciferase complementation assays in tobacco transient expression systems, the pCAMBIA1300-cLUC-GW and pCAMBIA1300-nLUC-GW vectors were used. To test the interaction between TCP4 and AG or SEP3 *in planta*, 35Spro-mTCP4-nLUC, 35Spro-cLUC-AG, and 35Spro-cLUC-SEP3 were generated by LR reactions between pCAMBIA1300-nLUC-GW or pCAMBIA1300-cLUC-GW and pENTR-mTCP4, pENTR-AG or pENTR-SEP3. To test the interaction between BEL1 and TCP4 or AG-SEP3, the 35Spro-cLUC-mTCP4, 35Spro-cLUC-AG and 35Spro-BEL1-nLUC constructs were generated by LR reactions between pCAMBIA1300-cLUC-GW or pCAMBIA1300-nLUC-GW and pENTR-mTCP4 or pENTR-AG.

Tobacco leaves were co-infiltrated with different combinations of vectors. The tobacco plants were grown under a weak light for 12 hr and transferred into a greenhouse with a long-day condition (22 °C) for 32 hr. To test the interactions between proteins under 22 °C and 28 °C treatments, the tobacco plants were further subjected to an ambient temperature of 22 °C or 28 °C for 4 hr. The infiltrated leaves were then dissected and injected with 200 μM luciferin. The leaves were kept in darkness for 10 mins. The illumination images were captured with a low-light cooled CCD imaging apparatus (NightOWL II LB983) and indiGO software. An exposure time of 3 min was used to take images. The relative luminescence unit (RLU) was calculated with indiGO software.

### Co-IP and western blotting

To test the interaction between TCP4 and AG or SEP3 by Co-IP in Arabidopsis protoplasts, the 35Spro-GFP-mTCP4, 35Spro-AG-FLAG, and 35Spro-SEP3-FLAG plasmids were purified from *Escherichia coli* strain DH5α for transformation into Arabidopsis protoplasts. To prepare protoplasts, Arabidopsis rosette leaves were digested with enzyme solution (1.5% cellulose R10, 0.4% macerozyme R10, 0.4 M mannitol, 20 mM KCl, 20 mM MES, 10 mM CaCl₂, and 0.1% BSA) for 5 h. The protoplasts were collected and suspended with MMG buffer (0.4 M mannitol, 15 mM MgCl₂, and 4 mM MES) to a final concentration of 2 ~ 5 × 10$^5$ cells/mL. For each Co-IP experiment, 2 mL protoplast solution was used. Transfection was performed with 100 μg purified plasmids for each expression construct combination in PEG solution

(40% PEG4000, 0.2 M mannitol, and 0.1 M CaCl₂). The combined volume of the protoplast solution and the plasmids was kept constant among the construct combinations. After incubation at room temperature for 10 min, 10 mL W5 buffer (154 mM NaCl, 125 mM CaCl₂, 5 mM KCl, 2 mM MES, and 5 mM glucose) was added to terminate the transfection. The transfected protoplasts were incubated under a weak light for 14 ~ 16 hr at room temperature, collected, and treated with IP buffer (50 mM pH 8.0 Tris-HCl, 100 mM NaCl, 1 mM MgCl₂, 10% glycerol, 1% NP-40, 1 mM PMSF, 1 mM DTT, 20 mM MG132, and 1× protease inhibitor cocktail). The lysis mix was incubated on ice for 10 min and then centrifuged at 12000 × *g* for 10 min. The supernatant was mixed with the prewashed anti-GFP agarose beads (KTSM1301, KT HEALTH), and rotated at 4 °C for 4 hr. The beads were washed five times with the IP buffer. The bound proteins were eluted from the affinity beads by heating the beads to 100 °C for 10 min in 100 μL 1× SDS loading buffer, after which the proteins were detected by western blotting.

To test the interactions between TCP4 and BEL1, 35Spro-GFP-BEL1 and 35Spro-mTCP4-FLAG constructs were co-transformed into tobacco leaves. The tobacco plants were put under a weak light for 12 h, and then transferred into a greenhouse with a long-day condition (22 °C) for 36 h. The transformed leaves were dissected, homogenized with liquid nitrogen, and suspended with IP buffer. The procedures used for protein extraction, immunoprecipitation, and immunoblotting were described above.

To evaluate the stability of BEL1 under HT, 35S-BEL1-FLAG-P2A-GFP was generated by LR reaction between pENTR-BEL1-FLAG-P2A-GFP and pK2GW7 (Ghent University). The fragments BEL1-FLAG(N) and (C)FLAG-P2A-GFP were cloned into the pENTR-TOPO vector by Gibson Assembly (SC612, Genesand). BEL1-FLAG(N) was amplified from the pENTR-BEL1 plasmid using Gib-BEL1-FLAG-F/R primers. (C)FLAG-P2A-GFP was amplified from the pUBQ10-Cas9-P2A-GFP-rbcS-E9t plasmid[52] using Gib-P2A-GFP-F/R primers.

The constructed plasmid was transferred into *A. tumefaciens* strain GV3101, and 35S-BEL1-FLAG-P2A-GFP was transformed into *N. benthamiana* leaves. After 2 days of culture, the tobacco plants were treated at 28 °C for the specified period. Proteins were extracted, and the BEL1-FLAG protein was detected with an anti-FLAG antibody (A8592, Sigma, 1:1000 dilution). GFP was detected with an anti-GFP antibody (AE011, ABclonal, 1:1000 dilution) as a control.

### In vivo degradation assay

To perform in vivo BEL1 degradation assay, the inflorescences from 35S-BEL1-FLAG transgenic plants were treated by solutions containing 50 mM CHX with or without 100 mM MG132 for a time course. The treated tissues were ground into powder in liquid nitrogen. Total proteins were extracted with extraction buffer (0.1 M HEPES at pH7.5, 5 mM EGTA, 5 mM EDTA, 5 mM NaF, 0.05 M phosphoglycerol, 1% Triton X-100, 10% glycerin, and protease inhibitor cocktail, PMSF) for 1 h and then centrifuged for 10 min. The proteins were detected by western blotting using the anti-FLAG antibody (A8592, Sigma) and an anti-Actin antibody (AC009, ABclonal).

### Transient dual-luciferase reporter assay

To investigate the transactivation activity of the AG-SEP3 complex, the dual-luciferase reporter assay was used with an Arabidopsis protoplast transient expression system. The 4067-bp and 4907-bp promoters of *CRC* and SPT, which are direct targets of the AG-SEP3 complex, were amplified with the CRCpro-F/R and SPTpro-F/R primer pairs. The promoters were each cloned into the pENTR/D TOPO vector to generate pENTR-CRCpro and SPTpro. The reporter constructs 35Spro-REN-CRCpro-LUC and 35Spro-REN-SPTpro-LUC were generated by LR reactions between pENTR-CRCpro or pENTR-SPTpro and pGreenII0800-GW-LUC. pGreenII0800-GW-LUC was generated by ligating attR1-ccdB-attR2 amplified from pK2GW7 using the primers

attR1-Xho I-F and attR2-Spe I-R, with pGreenII0800-LUC digested by *Xho* I and *Spe* I. To generate effectors, the coding regions of *mTCP4*, *AG,* and *SEP3* were cloned into pGreenII62SK using the FastClone method to generate 35Spro-mTCP4, 35Spro-AG and 35Spro-SEP3. The primers used for amplifying the backbones of pGreenII62SK were 62SK-FC-F and 62SK-FC-R.

Arabidopsis protoplasts were prepared as described above. For dual-luciferase reporter assays, 100 μL protoplasts were used for each transfection experiment. For plasmid transfection, each plasmid was adjusted to 10 μg. The transfected protoplasts were centrifuged and suspended with 1 mL W5 solution and incubated under a weak light for 12 - 16 hr. The LUC/REN intensity was detected with a dual-luciferase reporter system (Promega, GLO-MAX 20/20 luminometer).

## CRISPR/Cas9-induced mutants
To generate *tcp1* and *tcp24*, a CRISPR/Cas9 system containing an egg cell-specific promoter was used as previously described[69]. To generate the tredecuple mutant *tcpDUO bel1* or *tcpDUO stk*, a CRISPR/Cas9 system containing a *UBIQUITIN10* (*UBQ10*) promoter was used[52]. The spacer oligonucleotides were designed on the sequences of target genes, ligated into pAtU6 vectors, and cloned into pCAMBIA1300-UBQ10-Cas9-P2A-GFP. The constructs were transformed into wild-type or *tcpDUO*. The mutated fragments were first amplified using PCR and were then determined by sequencing. The oligonucleotides and primers are listed in Supplementary Table 1.

## GUS staining
TCP4pro-TCP4-GUS transgenic plant generation and β-glucuronidase (GUS) staining were carried out[36]. Briefly, the pistils of TCP4pro-TCP4-GUS plants were isolated and immersed in GUS staining solution at 37°C overnight. The samples were then transferred into 75% ethanol. The ovule staining pattern was observed using a Zeiss AX10 Imager M2 microscope.

## RNA-seq analysis
To perform RNA-seq analysis, the bolting wild-type or *tcpDUO* plants were treated at 28 °C for 6 d. Then the flowers at developmental stage 9 were emasculated, and the pistils were isolated after 2 DAE from the wild-type or *tcpDUO* plants treated with 22 °C or 28 °C for total RNA extraction. RNA-seq library preparation and high-throughput sequencing were performed by Novogene Inc. (Tianjin, China). The libraries were sequenced on an Illumina NovaSeq 6000 platform by generating 150 bp paired-end reads. Quality control of the raw reads was performed by fastp (0.20.1). The clean reads were mapped to the Arabidopsis reference genome (TAIR10) by STAR. Expression counts were generated by featureCounts of Subread 2.0.1with default parameters. The R package DEseq2 was used for gene differential expression analysis (1.32.0). Volcano plots were generated by R package ggpubr (0.4.0). Genes with Bonferroni-adjusted $p$-value (padj) ≤ 0.05 and a $\log_2$ fold change ≤ −1 or ≥ 1 were considered to be significantly differentially expressed.

## Electrophoretic mobility shift assay
The DNA sequence encoding the TCP4 DNA binding domain was cloned into pET-21b with a C-terminal His-tag. The vector was transformed into *E. coli* strain BL21. The bacteria were cultured overnight at 18°C, and proteins were induced by adding IPTG to a final concentration of 0.5 mM. The cells were collected and resuspended in buffer A (25 mM Tris and 1 M NaCl [pH 8.0]). The target protein was eluted with 200−500 mM imidazole using a Ni chelating column (GE Healthcare, USA). The crude protein was then subjected to size-exclusion chromatography (Superdex 75, GE Healthcare) for the final purification step.

20−30 bp DNA fragments containing a potential TCP4 binding motif were synthesized. The binding motif was mutated to AAAAAA in the mutated DNA fragments. Double-stranded DNA was obtained by annealing equimolar concentrations in an annealing buffer (25 mM Tris and 100 mM NaCl [pH 8.0]).

To carry out EMSA, the purified TCP4 DNA binding domain proteins (about 180 ng) were mixed with double-stranded DNA (about 200 ng). The mixture was incubated at 4°C for 20 min and then loaded onto a 7% nondenaturing polyacrylamide gel and subjected to electrophoresis using 0.5×TBE buffer. Electrophoresis was performed at 120 V for 40 min. Ethidium bromide staining was used to detect DNA under Tanon2500.

## In situ hybridization analyses
The in situ hybridization experiments were performed as previously described[70]. Briefly, the transcripts of *TCP4* or *STK* were amplified with primer pairs listed in Supplementary Table 1 for probe preparation. Probes were transcribed with the templates in vitro with the DIG RNA labeling kit (Roche, Switzerland) following the user manual. 1 mL hybridization buffer was prepared by adding 87.5 μL DEPC-H₂O, 125 μL in situ hybridization salts, 500 μL deionized formamide, 250 μL 50% dextran sulfate, 25 μL 50× Denhardt's solution, and 12.5 μL 100 mg/mL tRNA for hybridization.

For sample fixation and hybridization, the fresh pistils of wild type and *tcpDUO* growing at 22 °C or 28 °C were dissected and immediately fixed in FAA solution. The tissues were then embedded in paraffin (Sigma-Aldrich, USA) and sectioned to 7 μm-thick slides. The samples were tiled on poly-L-lysine-treated microscope slides (Sigma-Aldrich, USA), then de-paraffinized using HistoClear (Amersco). The samples were re-hydrated by ethanol series of 100%, 100%, 95%, 90%, 80%, 60%, and 30% (vol/vol) at room temperature. The slides attached with samples were transferred to DEPC-H₂O, then treated with freshly prepared 2×SSC for 10 min. Samples were digested with 1 μg/mL proteinase K for 20 min at 37 °C. The digestion was stopped by being treated with 0.2% glycine for 2 min. After washing with PBS twice and dehydrating with ethanol series, samples were hybridized with probes overnight at 55 °C. The samples were washed twice with 0.2× SSC at 55 °C, then twice with NTE buffer (0.01 mol/L Tris-HCl with pH8.0, 0.001 mol/L EDTA, 0.5 mol/L NaCl) at 37 °C. The samples were digested with 5 μg/mL RNase A in NTE buffer for 30 min, followed with twice washing in NTE buffer at 37 °C and once in 0.2× SSC at 55 °C. The samples were blocked in a blocking buffer (Roche) at room temperature for 45 min. Then the hybridized transcripts were detected with anti-digoxigenin antisera conjugated to alkaline phosphatase (Roche). The signals were detected with a microscope after incubation overnight in the dark at room temperature.

## Accession numbers
Accession numbers of proteins in this article are TCP1 (AT1G67260), TCP2 (AT4G18390), TCP3 (AT1G53230), TCP4 (AT3G15030), TCP5 (AT5G60970), TCP10 (AT2G31070), TCP12 (AT1G68800), TCP13 (AT3G02150), TCP16 (AT3G45150), TCP17 (AT5G08070), TCP18 (AT3G18550), TCP24 (AT1G30210), BEL1 (AT5G41410), CRC (AT1G69180), HEC1 (AT5G67060), HEC2 (AT3G50330), HEC3 (AT5G09750), INO (AT1G23420), SEP3 (AT1G24260), SHP1 (AT3G58780), SHP2 (AT2G42830), SPT (AT4G36930), STK (AT4G09960) and WUS (AT2G17950). The amino acid, CDS, and genomic sequences used in this article are available at TAIR (https://www.arabidopsis.org/).

## Reporting summary
Further information on research design is available in the Nature Portfolio Reporting Summary linked to this article.

# Data availability
The raw RNA sequencing data generated in this study have been deposited and available in the Sequence Read Archive (SRA) under

BioProject ID PRJNA891376. TCP4 ChIP-seq data are available under BioProject ID PRJNA474688. Source data are provided with this paper.

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

## Acknowledgements

We thank Li-Jia Qu (Peking University) and Hongya Gu (Peking University) for their discussions and valuable suggestions. We thank Yuval Eshed (Weizmann Institute of Science, Israel) for kindly providing the *tcp5 tcp13 tcp17* triple mutant seeds. We thank Tomotsugu Koyama (Kyoto University) for kindly providing the seeds of *tcp3 tcp4 tcp10* and *tcp3 tcp4 tcp5 tcp10 tcp13*. We also thank the Core Facilities of Life Sciences, Peking University, and the National Center for Protein Sciences at Peking University for assistance with the SEM assay. We are grateful to Dr. Yiqun Liu, Miss Yingying Guo, and Ms. Li Zhang for technical support for the SEM analysis. We also thank Dr. Shuaibin Zhang and Dr. Xiu-Li Hou for helping with the in situ hybridization assay. This research was supported by the National Natural Science Foundation of China (Grant No. 31970194), the National Science Fund for Distinguished Young Scholars of China (Grant No. 31725005), and the Science Fund for the Creative Research Groups of the National Natural Science Foundation of China (Grant No. 31621001).

## Author contributions

G.Q. conceived and designed the project. J.L., N.W., Y.W., and Y.J. conducted the experiments. G.Q., J.L., N.W., Y.W., H.Y., and X.C. analyzed the data. G.Q., J.L., and N.W. wrote the manuscript.

## Competing interests

The authors declare no competing interests.
