## [Peer Review File · Nature Communications]

Arabidopsis TCP4 transcription factor inhibits high temperature-induced homeotic conversion of ovulesREVIEWER COMMENTS

Reviewer #1 (Remarks to the Author):

Lan et al., have reported the role of Class II TEOSINTE BRANCHED 1/CYCLOIDEA/ PCF (TCP) in the determination of ovule identity characterizing the duodecuple tcp2/3/4/5/10/13/17/24/1/12/18/16 (tcpDUO) mutant. The tcpDUO mutant developed sterile ovules.

The authors reported that tcpDUO emasculated mutants exposed to a temperature of 28°C developed carpelloid structures instead of ovules, whereas only a small number of ovules showed carpelloid structures at 22°C. The authors have performed morphological analysis of tcpDUO mutant showing ovule defects.

They report that the tcpDUO mutant phenotype could be partially restored by introducing the pTCP4::TCP4-GFP construct in the mutant background.

To investigate the molecular mechanism controlling the observed homeotic conversion, the authors have performed RNAseq of wt and tcpDUO mature pistils at 22°C and 28°C. By comparing the RNAseq datasets the authors did not detect significant changes in ovule/carpel identity gene expression, such as STK, SHP1, SHP2 and AGAMOUS, however they reported that carpel expressed genes as HECATEs, and CRC were up regulated in tcpDUO respect to wt at 28°C.

To verify whether the observed homeotic transformations were due to changes in MADS-domain floral organ identity protein complex compositions, the authors have performed protein-protein interaction assays. They showed that TCP4 could interact with AG, SEP3 and BEL1 and they concluded that this interaction might facilitate a negative regulation of the AG-SEP3 function in ovules.

To study the genetic interactions between the TCPs, STK and BEL the authors have introduced by CRISPR-Cas9 technology in the tcpDUO mutant, mutations in the STK and BEL genes, creating tredecuple mutants. Based on the mutants morphological analysis and the protein-protein interaction assay, they conclude that TCPs are required for the stabilization of the protein complex involved in the determination of ovule identity in Arabidopsis.

Although the data included in the manuscript are of interest and show a role of TCPs in ovule identity determination, I have concerns about the results presented, lack of rationale behind some of the experimental design and the conclusion.

Major Concern

1. The authors describe the analysis of a duodecuple tcp2/3/4/5/10/13/17/24/1/12/18/16 (tcpDUO) mutant without describing what happened when a smaller number of TCP genes are mutated. Are all 12 TCPs fully redundant?

I have some doubts about this since some of the TCP genes that were mutated in the tcpDUO mutant were not expressed in pistils according to the presented RNAseq data (at least in pistils at a mature stage)

The authors mentioned that they have analyzed a tcp2/3/4/5/10/13/17 (tcpSEP) septuple mutant, and that this tcpSEP mutant did not show a significant phenotype at 22°C. The authors do not provide any evidence that the seven TCPs are expressed during ovule development therefore it is not clear why they assign a putative function in ovule development to these genes.

Similarly, the authors have to explain the rationale on which they have decided to mutagenize TCP 24/1/12/18/16 and no other TCPs without first analyzing the expression profile of the selected genes.

2. Are all the 12 TCPs expressed in the same ovule tissues at the same time? From the

RNAseq analysis (done on mature pistils) it seems that not all the included TCPs were expressed in the pistils. Are they expressed in ovules? The authors should provide RNA in situ hybridization studies for the TCPs included in their study.

TCP expression levels must also be discussed in the manuscript in more detail. Some TCPs that have been included in the analysis doesn't seem to be expressed (TCP1) or very low expressed (TCP16, 18, 4, 24). Is it possible that those TCPs are expressed or more expressed in early stages of ovules development? How sure the authors are that these TCP genes are expressed at all in ovules?

3. The pTCP4::TCP4-GFP construct is able to rescue only partially the phenotype. Is it possible that the TCPs that were analyzed are not fully redundant? Furthermore, it is unclear why the authors used TCP4 for this complementation study.

Moreover authors have used pTCP4::TCP4-GFP for their complementation study but show the expression of pTCP4::TCP4-GUS in Supplemental fig. 3 c-e. Why don't they show the GFP expression of the pTCP4::TCP4-GFP construct in wild type and in mutant background? As mentioned also above the authors must include TCP4 in situ hybridization analysis during early stage of ovule development to show overlapping expression with BELL1, AGAMOUS and SEP3 since the authors suggest that TCP4, BELL1, AG and SEP3 interact during ovule development. The RNAseq data which they included in the manuscript showed a very low TCP4 expression in the pistil and the TCP4::GUS pictures do not show sufficient detail and stages to show that TCP4 endogenous expression is in the same domain of expression as BELL1, AG and SEP3. Again, confocal analysis using the GFP construct at different stages of development will for sure give a much better resolution.

It will be of interest analyses the expression of the TCPs genes in *stkshp1shp2*, *bel* and *ag* backgrounds to verify whether the TCPs are downstream the ovule/carpel identity complexes.

4. The characterization of the *tcpDUO* mutants must be described in more detail.

i) Has the *tcpDUO* pistil a wild-type phenotype? Is the transmitting tract correctly formed allowing the pollen tubes to grow like in wild-type pistils? The *tcpDUO* sterile phenotype could be of interest however it is not described in proper detail.

ii) The authors showed two different phenotypes in Fig. 2 f and g. However, they do not mention the frequency of these phenotypes: "frequently observed" is not a scientifically sound description. The authors should specify the % of ovules of the specific phenotypes and the total number of ovules observed.

iii) The Figures from 2q to 2u are too small and it is not possible to observe the female gametophyte phenotype. None of the ovules shown in Figures 2s, t and u have the integuments defects that are shown in Fig 2g. This is in contrast with the sentence in line 207 "..... and an outer integument that frequently failed to cover portions of the inner integument."

iv) I believe that it is important to understand at which stage the homeotic conversion is taking place and which ovule tissues were involved in the homeotic conversion.

Consequently, the authors must provide ovule morphological analysis of *tcpDUO* vs wt, at 22°C and 28°C from stage 8/9 to 12 (according to Smyth et al., 1990).

v) I think that the aborted seeds reported in Figure 1 are mainly unfertilized and/or not well-developed ovules rather than aborted seeds.

5. The authors wrote "we treated the emasculated flowers of *tcpDUO* and wild-type control with HT (28°C). We found that almost all of the ovules in the *tcpDUO* pistils were converted into carpelloid structures (Fig. 3d)". According to this sentence the authors have treated the flowers containing mature ovules (stage in which anthers can be removed) with HT (28°C).

This description is different in respect to what they wrote in the materials and methods section where they wrote that they treat plants after bolting for 6 days and that they emasculated the plants after this treatment.

I suggest that the authors clearly describe at what flower stages they have applied the HT. They can refer to Smyth et al., 1990.

Furthermore, the description of the treatment and time of pistil analysis (morphological analysis and RNAseq) have to be consistent in the different sections of the results and in the material and methods.

6. The tcpDUO mutant pistil has ovules converted into carpelloid structures. They authors reported that the ovule identity gene STK, does not change its expression in tcpDUO respect to wt referring to the RNAseq data. What about other genes expressed in ovules such as BEL, INO, NZZ, WUS ect?

I believe that to conclude that STK (expression in the tcpDUO ovules do not change, it is absolutely necessary include in situ hybridization showing evidence for this assumption.

7. The authors performed an EMSA shift assay to study TCP4 interaction to regulatory regions of CRC and SPT. I believe that a CHIP assay is necessary to confirm the data and to assess whether HT affects TCP4 binding to the DNA in vivo. Since the authors have a pTCP4::TCP4-GFP construct available that complements the mutant phenotype it is not clear why they use an in vitro EMSA assay instead of a in vivo ChIP analysis.

8. It is quite peculiar that all the SEP3 and AG protein domains are able to interact with TCP4/TCP18 since these protein domains have very different structures. A negative control, like a not interacting TCP protein will be important to show the specificity and to exclude that the observed interactions are unspecific.

The conclusion that all these proteins can interact in ovules is also not supported.

Experiments showing TCP4/18 co-expression in the ovule together with the other proteins tested is mandatory.

9. The tcpDUO bel and tcpDUO stk seem to have different phenotypes. According to YAMADA et al. (2019), BEL is important to repress the formation of an ectopic nucella from the chalaza as well as repressed in later stages homeotic conversions of ovule integuments into carpelloid structures. Furthermore, it has been reported that BEL is involved together with HD-ZIP factors in controlling WUS expression in the chalaza (Yamada et al., 2016). I think that the phenotype of the tcpDUO bel and tcpDUO stk must be described in more details and discussed considering what it is already know about STK and BEL function in chalaza and integument development.

Minor

The tcpDUO phenotype analysis have to be described in more details : "about 16.1 %", (line 179); "about 18% (line 184)"... ect the authors have to include how many pistil/siliques they have analyzed.

EC1.1, EC1.3 and EC1.5 are expressed in the egg cell. Therefore, it is expected that they are not expressed in the tcpDUO mutant since in this mutant the mature female gametophytes are not formed even at 22°C. It will be better to report the expression of other ovule specific genes. The observation that CRC, HEC1, 2 and 3 are upregulated in the tcpDUO mutant might be a consequence that many ovules are converted into carpelloid structures.

Line 224: replace "differet" with "different"

Line 267: please, explain what you meant for "seed arrangement" because it is not clear.

I suggest including the position of the TDNA in supplemental Fig.3b to make the Figure clearer.

Reviewer #2 (Remarks to the Author):

In this manuscript the role of the Arabidopsis type II TCP transcription factors in ovule development is studied under high ambient temperature conditions. The authors conclude that the functioning of class II TCP proteins is essential to guarantee ovule development instead of the formation of carpeloid structures at high ambient temperatures. The authors generated an impressive duodecuple mutant to come to this conclusion. Subsequently, various molecular and biochemical assays were performed to shed light on the underlying molecular-mode-of-action. The majority of these experiments have been well performed, but as further explained below, in some cases the set-up or interpretation of data needs attention.

1. The authors generated a duodecuple mutant, in which all class II TCPs and the class I TCP16 gene are modified. For the phenotyping, the focus is directly on, and limited to ovule development. Only in the last sentence of the discussion, the authors indicate that this mutant has additional phenotypes, as expected. It is good to focus; however, to judge whether the observed phenotypes are not just pleiotropic effects due to overall strong phenotypic alterations, the authors should at least show the effect of these multiple mutations at whole plant level.
2. Line 263/264: '.....and their transcript levels were not obviously affected by HT, with the exception of TCP14.'. I believe that at least also TCP7 is different. 'Not obviously affected' is a vague term. Please check for each TCP gene whether there is a statistical difference and change the text accordingly.
3. Line 266-268. In contrast to a very detailed and well-performed phenotypic analysis of the ovules in the duodecuple mutant, expression analysis for the TCPs under study is very limited. Only for TCP4 some low resolution GUS staining pictures are provided in the supplements. It is not needed to show the detailed spatial and temporal expression pattern of each individual TCP gene during ovule development, but a more elaborate investigation of TCP4 expression linked to the proposed functions would be essential. For example: What about GFP signal in the complementing TCP4pro-TCP4-GFP line? Linked to this: I do not see GUS signal in supp Fig. 3C!?
4. Line 280/281: 'These data suggest that TCPs may be important mediators of gene repression in pistils under HT.'. The authors are already careful, but whereas we deal here with the final steady state expression levels, observed effects can be completely indirect and opposite of eventual initial direct effects. I would remove this suggestion.
5. Line 365-368, Yeast two hybrid results. The fact that TCP4 and TCP18 bind to each individual domain of SEP3 and AG could also mean that the observed interactions are not specific. I miss here some controls. Could other MADS proteins be included? And what about STK and SHP1/2? Are these also interacting? If so, what does that mean for the proposed model!!!??
6. Line 400/401: '....suggesting that TCPs and BEL1 formed a complex with AG-SEP3 to inhibit the activity of AG-SEP3 quartets.'. This could be, but the authors do not provide any evidence that the effect is specifically on quartet formation. Note that the performed experiments do not allow to say anything about the exact stoichiometry of the formed complexes! Please modify.
7. Line 409: '.....the fluorescence was significantly increased'. I am not convinced. Please quantify, perform a proper statistical test, and draw the conclusion based on sufficient repetitions.
8. The model presented in figure 8 and also the concluding sentence in the discussion: 'We

further showed that TCP4 directly interacted with BEL1 and AG-SEP3 to stabilize the BEL1-AG-SEP3 complex and negatively regulated the function of AG-SEP quartets.', suggest that the TCP4 mode of action is to interfere with AG-SEP3 complex formation. This could indeed be the case, but how to link this to the observed differences in expression of supposed downstream genes and showing that TCP4 can bind to the promoters of these genes??? This suggests that the observed phenotypic effects depend on TCP4-mediated DNA binding! I do not expect the authors to figure out what is exactly happening and causal, but this discrepancy should at least be dealt with in the discussion.

9. Line 586/587: '....and both TCP4 and BEL1 repressed the expression of carpel identity genes by interacting with the AG-SEP3 quartet.....'. See my comment above. No evidence is provided that it acts directly on quartet formation! Further more, only circumventational evidence is provided that the proteins are present in one large complex.

Minor points

- Line 32/33: 'Disruption of all Class II TCPs in a duodecuple tcp2/3/4/5/10/13/17/24/1/12/18/16 (tcpDUO) mutant....'. Please make clear that TCP16 is not a class II TCP. For example by changing this sentence into: 'Disruption of all Class II TCPs and the Class I TCP16 gene in a duodecuple tcp2/3/4/5/10/13/17/24/1/12/18/16 (tcpDUO) mutant.....'.

- Line 170: 'member but contains a TCP domain more similar to that of Class II TCP proteins....'. Please provide the correct reference for this statement:
<https://doi.org/10.1074/jbc.M111.256271>

- Line 271/272: 'indicating that no complete transcripts for the 12 TCPs were generated in the tcpDUO null mutant'. This is true for the majority of the TCPs, but cannot be seen for the CRISPR mutagenised TCP24 gene in the current representation. Also for the TCPs for which no expression was found it cannot be concluded. Please be more precise and rephrase accordingly.

- Label figure 3H. tcpUO should be tcpDUO.

- Label Supp fig 4B: tcpDOU should be tcpDUO.

Reviewer #3 (Remarks to the Author):

This complex manuscript uses a range of methods to examine the roles of TCP genes in ovule development. Much of the work is well-described and supports conclusions of the manuscript, other parts of the work are more difficult to follow, and some appear insufficient to support associated conclusions. I will not separate out the positive and less positive evaluation of the work, but rather will follow the order of the manuscript in my evaluation of the presented work.

They show that combined mutation in 12 TCP genes (tcpDUO) has significant effects on pistil development and the fertility of ovules. The fruits are of reduced length, but this appears to result from a reduction in seed set. TCP4 is able to complement these effects. The tcp4DUO mutant line is shown to produce carpeloid ovules at high frequency in elevated temperatures, and this effect is also rescued by the TCP4 gene.

RNAseq experiments show that carpeloid ovules express carpel developmental regulators. This is to be expected and so is not a particularly interesting result. They show by ChIP that TCP4 appears to bind to regions flanking carpel development genes CRC, SPT, and HEC1. This is interesting as it suggests possible direct effects on expression of these genes. This binding is further supported by gel shift assays in Fig. 4k. Given that expression of these

genes is elevated in tcpDUO mutants it is a good hypothesis that the TCP proteins repress expression of these genes, but this is not directly demonstrated. The genes could be indirect targets of such repression. For Fig. 4, only panels e, g, h, j and k seem relevant to the text and perhaps the other panels should be moved to supplementary figure.

They show by three methods that TCP4 interacts with AG and SEP2 proteins, strongly supporting these interactions. They do the same to show interaction between TCP4 and BEL1.

They use transient expression studies to show that TCP4 can repress expression from the CRC and SPT promoters, and that this repression is dominant over expression enhancement induced by a combination of AG and SEP3 proteins. That this is not just an interaction with AG-SEP3 complex is shown by the repression of expression occurring even in the absence of AG and SEP3 (this conflicts with what they say on lines 383 – 385). They also use gel shift assays to show direct interaction of TCP4 with these promoter regions. Using split luciferase assays they provide evidence that TCP4 can increase the affinity of AG/SEP3 for BEL1. However, these qualitative assays are not completely convincing as they can vary from infiltration to infiltration and that could explain the variability. So, while this is evidence, it does not allow a firm conclusion.

In Line 420 – 437 they claim instability of BEL1 is shown by Fig. 6 f – h, but this is not convincing. The control band appears to change in concert with the BEL1 containing bands. The 0 time controls in the three panels vary significantly and they do not provide an explanation for this. Based on the description of the experiments it would have been expected that these zero time points would be similar. Also, they appear to measure steady-state protein levels without inhibition of protein synthesis, so protein stability is not actually measured. This experiment does not appear to provide evidence of differential stability of BEL1 in high temperature. Perhaps the experiment is misunderstood, but then it needs to be more clearly explained.

They examine combinations of tcpDUO mutations with newly generated alleles of bel1 and stk and compare the effects to the single mutants. They do not say what allele of bel1 is shown in Fig. 7 e and f. The ovules in these images don't look like any bel1 ovules I have seen (I have seen a lot). The outer layers have surface features like normal outer integuments, and there appear to be separate inner and outer integument structures, which is never the case in strong bel1 mutants. In contrast, the secondary ovules in Fig. 7 j look exactly like bel1 ovules from previous publications. Also, past experience with bel1 indicates that waiting longer (developmentally) leads to more ovules becoming carpelloid (as these authors later note on lines 631 – 632). The ovules in some of the figures showing carpelloid ovules in Fig. 7 appear to be in older pistils than those in panels showing less carpelloid conversion. The time after floral opening should be described for the structures imaged in these figures to allow evaluation of the effects independent of carpel aging. Still, the formation of carpelloid structures appears significantly enhanced in the combination of tcpDUO with bel1 over either gene mutant alone. Similarly, they show that tcpDUO in combination with stk makes carpelloid ovules, which are not seen in the stk single mutant. Thus they show that TCP proteins help maintain ovule structure and prevent carpelloid conversion of ovules, especially at elevated temperatures.

Their model is reasonable and most of it is supported by their data, but it does not incorporate all of their data. It is consistent with TCP proteins binding to BEL1, SEP and AG, but does not incorporate the apparent ability of TCP to bind to SEP and AG when BEL1 is not present. It also does not incorporate the ability of TCP to directly bind the CRC and SPT promoters in the absence of other proteins. The observations lead to the question: is TCP4 a transcription factor or a protein interaction factor? It is possible it is both and perhaps both activities should be discussed. The model incorporates differential protein instability of BEL1, but unless I misunderstand their assay, this has not been demonstrated in the manuscript.

This is not essential to the model, but is part of their explanation of the apparent temperature-dependence of the carpelloid ovule formation.

I provide below more detailed comments for suggestions for improvement or correction of the manuscript.

Lines 79 – 89 There is more detail provided than needed on embryo sac development. Development of the embryo sac is not a real subject of this manuscript.

Line 105-106 For the statement “Overexpression of AG causes the transformation of ovules into carpelloid structures, possibly due to increased activity of AG-SEPs quartets in ovules.” They cite ref. 16, Bowman, Drews and Meyerowitz (1994). But that paper doesn’t address this topic. That ectopic AG expression produces carpelloid ovules in Arabidopsis was shown by Ray et al. (1994) PNAS 91:5761-5766, and in tobacco by Mandel, et al. (1992) Cell 71:133-143. The reference should be corrected. Notably, Ray et al. also hypothesized that BEL1 inhibited AG carpel promoting activity in ovules, a point of the current manuscript.

Line 116 This isn’t really true. The cited paper showed that *bel1* ovules are aberrant at the normal growth temperature of 22° but that the effects are less severe at the low temperature of 16°. It may be that particular *bel1* allele is temperature sensitive. Also, there are many changes for a plant grown at 16°. 22° is not an elevated temperature for Arabidopsis.

Line 126 The TCP family is not “highly conserved”, in fact members are quite divergent. Perhaps they meant to say the TCP family is “present in all plants”.

Line 206 – 208. It actually looks like the main defect is a failure in the outer integument (OI) to grow sufficiently. Look at the cells at the edge of the OI, they are long in WT and very small in *tcpDUO*. The failure of the OI to grow could allow the II to grow because of the absence of a confining OI. This is the case in other ovule mutants where OI growth is deficient. Still, it does result in what they describe.

Line 227 the failure in female gametophyte (FG) formation could be a secondary effect of the aberrant sporophytic parts of the ovules. This is the case for mutants like *bel1* and *ant* where all FG form normally in heterozygous plants, where half of FG would be haploid mutants. So the effect on the mutant is on the ovule, not the FG directly.

Line 408 – 412 The signal with TCP4 present does appear stronger, but how much variation is there between individual leaf infiltrations of the with the same input. It is hard to conclude that this is real enhancement without more duplication or quantitation.

Line 420 – 421 by “expression level” I think they mean “steady state level of mRNA”. Expression could refer to the amount of protein. They should make this clear.

Line 422 What is “P2A”? I can find no reference for this. Two prolines and an alanine? Or is it some peptide sequence? Why is it cleaved? Is it an intein? Clearly it is not cleaved efficiently as the whole protein is visible in every lane of Fig. 6 f – h. More information is needed here.

Line 420 – 437. And Fig. 5 f – h. Not convinced of the conclusions here at all. Why is the 0 time point so different between f and g and f and h? Almost no cleaved BEL1-FLAG even though GFP levels are nearly constant. In g the control GFP band looks to change as much as the whole protein and the free BEL1-FLAG so that an effect of MG132 is not at all clear. Same is true in h for proposed effect of TCP4. These results are not at all convincing.

Line 932 – 933 Year is incorrect for reference 40.

Line 1282 – 1285 Figs. f, g, h are mislabeled e, f, g in the legend.

They often fail to say what was done in an experiment. For example. Supp. Fig. 4. Is this RNA seq data? Is this QRT-PCR? There is no information on what experiment was performed to get the reported numbers. I am guessing RNA seq, but there is no way to tell

from the information provided. The method could be added to the last sentence of the legend "...of three biological replicates of [whatever this experiment was]. They need to always describe what the experiment is for all of the figures. They should check over the whole manuscript for this.

SUMMARY

This is a very interesting manuscript and most of the conclusions are supported by the data. In particular, they show an important function for TCP genes in maintenance of ovule identity and show that this results in part from interactions with known ovule identity genes. This importance is more apparent at elevated temperature. If the poorly supported conclusions were removed, it would still be an interesting manuscript that makes a significant contribution to the field.

Response to Comments of Reviewer #1

Lan et al., have reported the role of Class II TEOSINTE BRANCHED 1/CYCLOIDEA/PCF (TCP) in the determination of ovule identity characterizing the duodecuple *tcp2/3/4/5/10/13/17/24/1/12/18/16* (*tcpDUO*) mutant. The *tcpDUO* mutant developed sterile ovules.

The authors reported that *tcpDUO* emasculated mutants exposed to a temperature of 28°C developed carpelloid structures instead of ovules, whereas only a small number of ovules showed carpelloid structures at 22°C. The authors have performed morphological analysis of *tcpDUO* mutant showing ovule defects.

They report that the *tcpDUO* mutant phenotype could be partially restored by introducing the pTCP4::TCP4-GFP construct in the mutant background.

To investigate the molecular mechanism controlling the observed homeotic conversion, the authors have performed RNAseq of wt and *tcpDUO* mature pistils at 22°C and 28°C. By comparing the RNAseq datasets the authors did not detect significant changes in ovule/carpel identity gene expression, such as STK, SHP1, SHP2 and AGAMOUS, however they reported that carpel expressed genes as HECATEs, and CRC were up regulated in *tcpDUO* respect to wt at 28°C.

To verify whether the observed homeotic transformations were due to changes in MADS-domain floral organ identity protein complex compositions, the authors have performed protein-protein interaction assays. They showed that TCP4 could interact with AG, SEP3 and BEL1 and they concluded that this interaction might facilitate a negative regulation of the AG-SEP3 function in ovules.

To study the genetic interactions between the TCPs, STK and BEL the authors have introduced by CRISPR-Cas9 technology in the *tcpDUO* mutant, mutations in the STK and BEL genes, creating tredecuple mutants. Based on the mutants morphological analysis and the protein-protein interaction assay, they conclude that TCPs are required for the stabilization of the protein complex involved in the determination of ovule identity in Arabidopsis.

Although the data included in the manuscript are of interest and show a role of TCPs in ovule identity determination, I have concerns about the results presented, lack of rationale behind some of the experimental design and the conclusion.

Major Concern

1. The authors describe the analysis of a duodecuple *tcp2/3/4/5/10/13/17/24/1/12/18/16* (*tcpDUO*) mutant without describing what happened when a smaller number of TCP genes are mutated. Are all 12 TCPs fully redundant?

I have some doubts about this since some of the TCP genes that were mutated in the *tcpDUO* mutant were not expressed in pistils according to the presented RNAseq data (at least in pistils at a mature stage)

The authors mentioned that they have analyzed a *tcp2/3/4/5/10/13/17* (*tcpSEP*) septuple mutant, and that this *tcpSEP* mutant did not show a significant phenotype at 22°C. The authors do not provide any evidence that the seven TCPs are expressed during ovule development therefore it is not clear why they assign a putative function in ovule development to these genes.

Similarly, the authors have to explain the rational on which they have decided to mutagenize *TCP24/1/12/18/16* and no other *TCPs* without first analyzing the expression profile of the selected genes.

Answer: Thank Reviewer #1 for the comments on the rational on why we generated a duodecuple *tcp* mutant at beginning, and why we did not generate lower order multiple *tcp* mutants to evaluate the redundancy and roles of *CYC*-like *TCPs* and other *TCPs* according to the expression of *TCPs* in pistils at first. Actually, we were not sure whether *CIN*-like *TCPs* had functional redundancy with *CYC*-like *TCPs* during ovule development. It takes painstaking and tedious work of genotyping to generate different *tcp* combinations of high order mutants. To save time and reduce the work, we need to know whether the disruption of all the Class II *TCPs* could cause some phenotypes at first. Because *TCP16* was reported to contain a *TCP* domain that is more similar to that of Class II *TCP* proteins, we decided to generate a duodecuple *tcp* mutant disrupting eleven Class II *TCPs* and one *TCP16*. If the *tcp* duodecuple mutant has no defective phenotypes in ovules, we could conclude that *CIN*-like *TCPs* may have no functional redundancy with *CYC*-like *TCPs* in this process. If we first generate a lower order mutant without ovule phenotype, we still do not know whether Class II *TCPs* redundantly control ovule development. The generation and analysis of different combinations of *tcp* mutant need a lot of time and tedious work. Fortunately, we succeeded obtaining the *tcp* duodecuple mutant and observed the exciting ovule defective phenotypes. After that, we began to investigate the expression of these *TCPs* in ovules and molecular mechanisms by which *TCPs* control ovule identity.

In this paper, we focused on the roles of TCP4 in determining the identity of ovule under high temperature, because the expression of *TCP4* completely complemented the conversion of ovules into carpelloid structures under high temperature.

According to the comments of Reviewer #1, we crossed the *tcp* duodecuple mutant *tcp2/3/4/5/10/13/17/24/1/12/18/16* with the septuple mutant *tcp2/3/4/5/10/13/17* to generate the different combinations of *tcp* lower order mutants for evaluating the redundancy of *TCP* functions. It was not easy for us to obtain all the different combinations of *tcp* lower order mutants, because tedious genotyping work was also need to separate the already combined mutants. However, we succeeded obtaining a octuple mutant *tcp2/3/4/5/10/13/17/24* in which all *CIN*-like *TCP* genes were mutated, a nonuple mutant *tcp2/3/4/5/10/13/17/1/12*, a decuple mutant *tcp2/3/4/5/10/13/17/1/12/18*, and a undecuple mutant *tcp2/3/4/5/10/13/17/24/1/12/18* in which all Class II *TCPs* were mutated. The results showed that disruption of all the *CIN*-like *TCPs* did not cause the conversion of ovule identity under HT. However, the compromise of all the Class II *TCPs* led to all the ovules converted into carpelloid structures under HT. This indicated that *CYC*-like *TCPs* played critical and redundant roles with *CIN*-like *TCPs* in protecting the ovule identity under HT, and TCP16 was dispensable in this process. No ovule conversion events were observed in the nonuple mutant and the decuple mutant, indicating that TCP24 made an important contribution in maintaining the ovule identity under HT.

We briefly explained the rational on the generation of the *tcp* duodecuple mutant in the Results and included the new observations of different multiple *tcp* mutants with different *tcp* combinations in new Supplementary Fig. 4 and in the Results of the revised manuscript as follows (Line 165-168, 264-279).

“In order to finalize the possible roles of the eleven Class II *TCP* genes and *TCP16* in controlling ovule development, we decided to generate a *tcp* multiple mutant in which all these *TCPs* were mutated.”

“To determine the contribution and redundancy of *TCPs* in maintaining ovule identity under HT, we crossed the *tcp* duodecuple mutant (*tcp2/3/4/5/10/13/17/24/1/12/18/16*) with the septuple mutant (*tcp2/3/4/5/10/13/17*) to generate the different combinations of *tcp* lower order mutants. We obtained a octuple mutant (*tcp2/3/4/5/10/13/17/24*) in which all *CIN*-like *TCP* genes were mutated, a nonuple mutant (*tcp2/3/4/5/10/13/17/1/12*), a decuple mutant (*tcp2/3/4/5/10/13/17/1/12/18*), and a undecuple mutant

(*tcp2/3/4/5/10/13/17/24/1/12/18*) in which all Class II *TCPs* were mutated. We treated these *tcp* multiple mutants with 28°C to observe the phenotype of homeotical conversion. The results showed that the disruption of all the *CIN*-like *TCPs* did not cause the conversion of ovule identity (Supplementary Fig. 4a, b). However, the compromise of all the Class II *TCPs* led to all the ovules converted into carpelloid structures (Supplementary Fig. 4g, h). This indicated that *CYC*-like *TCPs* played critical and redundant roles with *CIN*-like *TCPs* in protecting the ovule identity under HT, and *TCP16* was dispensable in this process. No ovule conversion events were observed in the nonuple mutant and the decuple mutant (Supplementary Fig. 4c-f), indicating that *TCP24* made an important contribution in maintaining the ovule identity under HT.”

2. Are all the 12 *TCPs* expressed in the same ovule tissues at the same time? From the RNAseq analysis (done on mature pistils) it seems that not all the included *TCPs* were expressed in the pistils. Are they expressed in ovules? The authors should provide RNA in situ hybridization studies for the *TCPs* included in their study.

TCP expression levels must also be discussed in the manuscript in more detail. Some *TCPs* that have been included in the analysis doesn't seem to be expressed (*TCP1*) or very low expressed (*TCP16, 18, 4, 24*). Is it possible that those *TCPs* are expressed or more expressed in early stages of ovules development? How sure the authors are that these *TCP* genes are expressed at all in ovules?

Answer: Thank Reviewer #1 for the comments and questions. In this manuscript, we focused on our study on the roles of *TCP4* in inhibiting high temperature-induced homeotic conversion of ovules, because it is really a tough and tedious work to elucidate the individual roles of all the twelve *TCPs*. The reasons are as follows. 1. The expression of *TCPs* may be dynamical and/or specific. That is, some *TCPs* might be only expressed in a specific stage in a short time during ovule development, but the role of them may be enough and important. If we do not detect the expression of some *TCPs* in our experiments, we cannot conclude that they do not play roles during ovule development, because we could not make sure whether we have not missed some developmental stage at which the *TCPs* were transiently expressed. We think that the high or low expression level of *TCPs* could not decide the importance of *TCPs* in ovule development. Because some *TCPs* may be expressed specifically in only several ovule cells. Although the detected expression level of them may be low, the function of them may be indispensable. Our genetic complementation analysis unequivocally

demonstrated the very important roles of *TCP4* in maintaining the ovule identity under high temperature. Our genetic analysis data showed that the decuple mutant *tcp2/3/4/5/10/13/17/1/12/18* did not display ovule identity conversion, while the undecuple mutant *tcp2/3/4/5/10/13/17/24/1/12/18* in which *TCP24* was disrupted in the background of the above decuple mutant *tcp2/3/4/5/10/13/17/1/12/18* produced carpelloid structures instead of ovules. This indicated the crucial functions of *TCP24* in suppressing the ovule identity conversion.

To finally determine each *TCP* roles during ovule development, one should generate eleven *tcp* decuple mutants which lacks each *tcp* mutant in the undecuple mutant *tcp2/3/4/5/10/13/17/24/1/12/18*. That is, for example, the phenotypical comparison between *tcp2/3/4/5/10/13/17/24/1/12/18* and *tcp3/4/5/10/13/17/24/1/12/18* could find the contribution of *TCP2* in specifying ovule, etc. To obtain the eleven *tcp* decuple mutants, one must cross *tcp2/3/4/5/10/13/17/24/1/12/18* with wild-type Arabidopsis, and genotyping thousands of plants from offspring of several generations. So, we are sorry that we cannot complete the work in short time. In our manuscript, we do not show all the individual *TCP* roles in ovule development. However, our data clearly show that Class II *TCPs* have redundant roles in protecting the identity of ovules under high temperature, and *TCP4* plays critical roles in this process by genetic complementation analysis.

Following the comments of Reviewer #1, we collected mature ovules from pistils of wild-type plants treated with 22°C or 28°C to test the expression of Class II *TCPs* and *TCP16* using RT-qPCR. The results showed that almost all these *TCPs* was expressed in ovules. Compared to the expression level of *TCP4*, *TCP2*, *TCP3* and *TCP24* had a higher expression level. The expression level of *TCP5*, *TCP10*, *TCP13*, *TCP12*, *TCP18* and *TCP16* was relatively lower, and *TCP17* and *TCP1* had the lowest expression level. We further cloned the 3043-bp promoter of *TCP1*, the 2466-bp promoter of *TCP12*, the 2428-bp promoter of *TCP18* and the 2571-bp promoter of *TCP24*. We generate *TCP1*pro-GUS, *TCP12*pro-GUS, *TCP18*pro-GUS and *TCP24*pro-GUS constructs in which these promoters were used to drive GUS reporter, respectively. GUS staining analysis showed that the expression of *TCP1* was hardly detected, while the expression of *TCP12*, *TCP18* and *TCP24* was clearly observed in the early ovules.

We also provide more new data to elucidate the expression pattern of *TCP4*. We performed in situ hybridization. The expression of *TCP4* shown by in situ hybridization was consistent with the GUS staining pattern of pTCP4::TCP4-GUS.

We included these data in new Supplementary Figure 7 and in the Results of the revised manuscript as follows (Line 300-321).

“We collected mature ovules from pistils of wild type plants treated with 22°C or 28°C to test the expression of Class II *TCPs* and *TCP16* using reverse transcription quantitative PCR (RT-qPCR). The results showed that almost all these *TCPs* except *TCP1* and *TCP17* were expressed in mature ovules (Supplementary Fig. 7a). We further cloned a 3043-bp promoter of *TCP1*, a 2466-bp promoter of *TCP12*, a 2428-bp promoter of *TCP18* and a 2571-bp promoter of *TCP24*. We generate TCP1pro-GUS, TCP12pro-GUS, TCP18pro-GUS and TCP24pro-GUS constructs in which these promoters were used to drive *GUS* reporter gene, respectively. GUS staining showed that the expression of *TCP1* was hardly detected (Supplementary Fig. 7b, c), while the expression of *TCP12*, *TCP18* and *TCP24* was clearly observed in the early ovules (Supplementary Fig. 7d-i). Staining experiments with the previously reported TCP4pro-TCP4-GUS transgenic line confirmed that TCP4 protein was expressed during ovule development (Supplementary Fig. 7j-m)³⁶. In situ hybridization showed that *TCP4* was highly expressed in nucellus and integuments, consistent with the GUS staining of TCP4pro-TCP4-GUS (Supplementary Fig. 7o-r).”

3. The pTCP4::TCP4-GFP construct is able to rescue only partially the phenotype. Is it possible that the *TCPs* that were analyzed are not fully redundant? Furthermore, it is unclear why the authors used TCP4 for this complementation study.

Moreover authors have used pTCP4::TCP4-GFP for their complementation study but show the expression of pTCP4::TCP4-GUS in Supplemental fig. 3 c-e. Why don't they show the GFP expression of the pTCP4::TCP4-GFP construct in wild type and in mutant background?

Answer: Thank Reviewer #1 for these questions. Although TCP4pro-TCP4-GFP could not completely rescued the infertility of ovules under HT, it completely complemented the conversion of ovules into carpelloid structures under HT (Fig.3E), indicating TCP4 played critical roles in protecting ovule identity under HT. In addition, TCP4 is one of the most important members in the Class II TCP family transcription factors. TCP4 has been used a typical representative to elucidate the important role of *TCPs* in leaf

development, trichome formation, petal color, and etc. (Crawford et al., 2004; Koyama et al., 2007; Koyama et al., 2010; Efroni et al., 2008; Efroni et al., 2013; Vadde et al., 2019; Lan et al., 2021; Zheng et al., 2022, please see the following References below this part). So, we selected TCP4 to identify its roles in maintaining ovule identity.

We used TCP4pro-TCP4-GUS in which a 2818-bp-long *TCP4* promoter was used to drive *GUS* reporter gene fusion with *TCP4* genomic DNA to analyze the expression pattern of TCP4 protein, because the GUS staining of petals from TCP4pro-TCP4-GUS transgenic plants perfectly matches the green-to-white color shift and the final green-white color pattern of petals in our previous work and the roles of TCP4 in the suppression of petal greening (Zheng et al., 2022, please see Figure 1 for response). We think that the GUS staining of TCP4pro-TCP4-GUS can represent the real expression pattern of TCP4 protein. The other reason is that the expression of TCP4-GFP was too low to be observed in our observation condition in the TCP4pro-TCP4-GFP transgenic plants, although TCP4pro-TCP4-GFP complemented the ovule identity conversion of *tcpDUO*. The possible reason was that TCP4-GFP could be post-transcriptionally down-regulated by microRNA319.

Figure 1 for response. The GUS staining of petals from TCP4pro-TCP4-GUS transgenic plants perfectly matches the green-to-white color shift and the final green-white color pattern of petals in our previous work (Zheng et al., 2022), in perfect consistence with the function of TCP4 repression of petal greening by inhibiting chlorophyll biosynthesis.

References:

- Crawford, B. C. W., Nath, U., Carpenter, R. & Coen, E. S.** CINCINNATA controls both cell differentiation and growth in petal lobes and leaves of *Antirrhinum*. *Plant Physiol.* 135, 244-253, (2004).
- Koyama, T., Furutani, M., Tasaka, M. & Ohme-Takagi, M.** TCP transcription factors control the morphology of shoot lateral organs via negative regulation of the expression of boundary-specific genes in *Arabidopsis*. *Plant Cell* 19, 473-484, (2007).
- Koyama, T., Mitsuda, N., Seki, M., Shinozaki, K. & Ohme-Takagi, M.** TCP transcription factors regulate the activities of ASYMMETRIC LEAVES1 and miR164, as well as the auxin response, during differentiation of leaves in *Arabidopsis*. *Plant Cell* 22, 3574-3588, (2010).
- Efroni, I., Blum, E., Goldshmidt, A. & Eshed, Y.** A protracted and dynamic maturation schedule underlies *Arabidopsis* leaf development. *Plant Cell* 20, 2293-2306, (2008).
- Efroni, I. et al.** Regulation of leaf maturation by chromatin-mediated modulation of cytokinin responses. *Dev. Cell* 24, 438-445, (2013).
- Vadde, B. V. L., Challa, K. R., Sunkara, P., Hegde, A. S. & Nath, U.** The TCP4 transcription factor directly activates TRICHOMELESS1 and 2 and suppresses trichome initiation. *Plant Physiol.* 181, 1587-1599, (2019).
- Lan, J. et al.** TCP transcription factors suppress cotyledon trichomes by impeding a cell differentiation-regulating complex. *Plant Physiol.* 186, 434-451, (2021).
- Zheng, X. et al.** *Arabidopsis* transcription factor TCP4 represses chlorophyll biosynthesis to prevent petal greening. *Plant Commun.* 3, 100309, (2022).

As mentioned also above the authors must include TCP4 in situ hybridization analysis during early stage of ovule development to show overlapping expression with BELL1, AGAMOUS and SEP3 since the authors suggest that TCP4, BELL1, AG and SEP3 interact during ovule development. The RNAseq data which they included in the manuscript showed a very low TCP4 expression in the pistil and the TCP4::GUS pictures do not show sufficient detail and stages to show that TCP4 endogenous expression is in the same domain of expression as BELL1, AG and SEP3. Again, confocal analysis using the GFP construct at different stages of development will for sure give a much better resolution.

Answer: Thank Reviewer #1 for the suggestion. We performed RT-qPCR analysis using the ovule tissues, the results showed that *TCP4* was expressed in ovules. GUS staining of TCP4pro-TCP4-GUS showed that TCP4 protein was clearly expressed in ovules. We further performed in situ hybridization of *TCP4* as suggested by Reviewer #1. The results showed that *TCP4* was highly expressed in both nucellus and integuments during megagametogenesis, consistent with the expression patterns shown by GUS staining of TCP4pro-TCP4-GUS (Supplementary Fig 7). The previous published data showed that *BEL1* was expressed in integuments (Reiser et al., 1995). *AG* was highly expressed in the integuments of the developing ovules (Reiser et al., 1995; Bowman et al., 1991). *SEP3* was expressed throughout ovules, including the nucleus, inner and outer integuments and funiculus (Mandel and Yanofsky, 1998) (Please see the following References and Figure 2 for response). Our data and the published data indicated that the expression pattern of *TCP4* was overlapped with *BEL1*, *AG* and *SEP3*.

We included these data in new Supplementary Figure 7 and in the Results of the revised manuscript (Line 300-314).

“We collected mature ovules from pistils of wild type plants treated with 22°C or 28°C to test the expression of Class II *TCPs* and *TCP16* using reverse transcription quantitative PCR (RT-qPCR). The results showed that almost all these *TCPs* except *TCP1* and *TCP17* were expressed in mature ovules (Supplementary Fig. 7a). We further cloned a 3043-bp promoter of *TCP1*, a 2466-bp promoter of *TCP12*, a 2428-bp promoter of *TCP18* and a 2571-bp promoter of *TCP24*. We generate TCP1pro-GUS, TCP12pro-GUS, TCP18pro-GUS and TCP24pro-GUS constructs in which these promoters were used to drive *GUS* reporter gene, respectively. GUS staining showed that the expression of *TCP1* was hardly detected (Supplementary Fig. 7b, c), while the expression of *TCP12*, *TCP18* and *TCP24* was clearly observed in the early ovules (Supplementary Fig. 7d-i). Staining experiments with the previously reported TCP4pro-TCP4-GUS transgenic line confirmed that TCP4 protein was expressed during ovule development (Supplementary Fig. 7j-m)³⁶. In situ hybridization showed that *TCP4* was highly expressed in nucellus and integuments, consistent with the GUS staining of TCP4pro-TCP4-GUS (Supplementary Fig. 7o-r).”

References:

Reiser, L. et al. The BELL1 gene encodes a homeodomain protein involved in pattern formation in the Arabidopsis ovule primordium. Cell 83, 735-742, (1995).

Bowman, J. L., Drews, G. N. & Meyerowitz, E. M. Expression of the Arabidopsis floral homeotic gene AGAMOUS is restricted to specific cell-types late in flower development. Plant Cell 3, 749-758, (1991).

Mandel, M. A. and Yanofsky, M. F. The Arabidopsis AGL9 MADS-box gene is expressed in young flower primordia. Sex. Plant Reprod. 11, 22-28, (1998).

BEL1 expression patterns showed in Reiser et al., 1995

Figure 4. In Situ Localization of *BEL1* RNA in Wild-Type and *Bell-3* Ovules
All tissues were hybridized with a *BEL1* antisense riboprobe. Morphological features of flowers, ovary, and ovule are indicated as follows: stamen (st); carpel (c); ovule (o); ovary wall (ow); ovule primordium (op); megasporocyte (ms); funiculus (fu); nucellus (nu); inner integument (i); outer integument (oi); endothelium (en); female gametophyte (asterisk); microphyte (mp); chazala (ch); and integument-like structure (ils). Scale bar represents 10 μ m, except for (A) where it represents 50 μ m. Wild-type inflorescence (A); stage 8 wild-type ovule (B); stage 9 wild-type ovule (C); stage 10 wild-type ovule (D); stage 11 wild-type ovule (E); stage 13 wild-type ovule (F); stage 9 *Bell-3* ovule (G); stage 11 *Bell-3* ovule (H).

AG expression patterns showed in Reiser et al., 1995

Figure 5. In Situ Localization of *AG* RNA in Wild-Type and *Bell-3* Mutant Ovules
All tissues were hybridized with the *AG* antisense riboprobe. Morphological features are indicated as in Figure 4. Scale bar represents 10 μ m, except (I) where it represents 50 μ m. Stage 9 wild-type ovule (A); stage 10 wild-type ovule (B); stage 11 wild-type ovule (C); stage 12 wild-type ovule (D); stage 13 wild-type ovule (E); stage 9 *Bell-3* ovule (F); stage 11 *Bell-3* ovule (G); stage 13 *Bell-3* ovule (H); stage 17 *Bell-3* ovule that has formed a carpel-like structure (cls) (I).

AG expression patterns showed in Bowman et al., 1991

Figure 2. Distribution of *AG* RNA in Developing Carpels.
(A) and (B) In situ hybridization of an *AG* anti-mRNA probe with a stage 9 gynoeceum. (A) Bright-field micrograph. (B) Dark-field micrograph.
(C) and (D) In situ hybridization of an *AG* anti-mRNA probe with a stage 12 gynoeceum. (C) Bright-field micrograph. (D) Dark-field micrograph.
(E) and (F) In situ hybridization of ³H-poly(U) probe with a stage 12 gynoeceum. (E) Bright-field micrograph. (F) Dark-field micrograph.
(G) and (H) In situ hybridization of an *AG* anti-mRNA probe with a differentiated ovule. (G) Bright-field micrograph. (H) Dark-field micrograph.
Flower structures are given the following abbreviations: ES, embryo sac; I, integuments; E, endothelium; O, ovary; Ov, ovule; P, placenta; St, stigma; Sty, style.
Bars = 50 μ m.

SEP3 (AGL9) expression patterns showed in Mandel and Yanofsky, 1998

Fig. 3A-P. *AGL9* Expression in *Arabidopsis* wild-type tissues. In situ hybridizations of wild-type plant tissues with an *AGL9*-specific antisense mRNA probe is shown. A, C, E, G, I, K, M, and O are bright-field photographs of tissue sections. B, D, F, H, J, L, N, and P are bright-field/field double exposures. All sections are longitudinal, except I and J, that are horizontal. A and B show a stage 2 and a stage 3 floral primordia. C and D show a wild-type inflorescence with a stage 7 flower. The arrows indicate stage 2 floral primordia. E and F show a stage 8 flower. G and H show a stage 10 flower. I and J show a carpel section of a stage 11 flower. K and L show a carpel section of a stage 10 flower. M and N show a carpel section of an early stage 12 flower. O and P show a carpel section of a stage 14 flower. In: Inflorescence meristem, *sc* sepal, *pc* petal, *st* stigma, *g* gynoeceum, *en* endothelium, *cy* style, *op* ovule primordia, *cw* carpel wall, *zy* septum, *ov* ovule.

Figure 2 for response. The expression pattern of *BEL1*, *AG* and *SEP3* in ovules shown in the published papers.

It will be of interest analyses the expression of the TCPs genes in *stkshp1shp2*, *bel* and *ag* backgrounds to verify whether the TCPs are downstream the ovule/carpel identity complexes.

Answer: We generated *stk shp1 shp2*, and *bell* mutants using CRISPR/Cas9 technology. The *stk shp1 shp2*, and *bell* mutants displayed similar phenotypes as reported. We performed RT-qPCR to test the expression of Class II TCPs and *TCP16* using mature pistils of the wild type, *stk shp1 shp2*, and *bell* mutants. The results showed that the expression level of these TCPs was not significantly changed in *bell*. However, the expression level of *TCP10*, *TCP13* and *TCP16* was significantly increased in the triple mutant *stk shp1shp2* (multiple *t* test, *: $P < 0.05$; **: $P < 0.01$; ***: $P < 0.001$). (Please see Figure 3 for response)

We think that the results were little related to our conclusion of this work. We did not include the results in the revised manuscript. We hope that Reviewer #1 could agree with us.

Figure 3 for response. The relative expression level of TCPs in pistils of the *stk shp1 shp2* and *bell* mutants.

4. The characterization of the *tcpDUO* mutants must be described in more detail.

i) Has the *tcpDUO* pistil a wild-type phenotype? Is the transmitting tract correctly formed allowing the pollen tubes to grow like in wild-type pistils? The *tcpDUO* sterile phenotype could be of interest however it is not described in proper detail.

Answer: Thank Reviewer #1 for the good suggestion. According to the suggestion, we analyzed the phenotype of the *tcpDUO* mutant in more detail. The results showed that

the length of pistils in the *tcpDUO* mutant was significantly shorter than that in wild-type control, corresponding to the shorter siliques in *tcpDUO*. To determine whether the pistils of *tcpDUO* could allow the growth of pollen, we pollinated the pistils of *tcpDUO* or wild-type control with wild-type pollen. The staining with aniline blue showed that pollen germinated on the stigma and pollen tubes grew through styles in pistils from both *tcpDUO* and wild-type control, indicating the transmitting tract of *tcpDUO* was rather normal.

We included these data in Supplementary Figure 2 and in the Results of the revised manuscript as follows (Line 173-181).

“The *tcpDUO* generated waving leaves and displayed late flowering as that observed in *tcpSEP* when compared to the wild-type control (Supplementary Fig. 2a, b). However, the length of pistils in *tcpDUO* was significantly shorter than that in wild-type control (Supplementary Fig. 2c, d). To determine whether the pistils of *tcpDUO* could affect the growth of pollen tube, we pollinated the pistils of *tcpDUO* or wild-type control with wild-type pollen. The staining with aniline blue showed that pollen germinated on the stigma and pollen tubes grew through styles in pistils from both *tcpDUO* and wild-type control (Supplementary Fig. 2e, f), indicating that the transmitting tract of *tcpDUO* was rather normal.”

ii) The authors showed two different phenotypes in Fig. 2 f and g. However, they do not mention the frequency of these phenotypes: “frequently observed” is not a scientifically sound description. The authors should specify the % of ovules of the specific phenotypes and the total number of ovules observed.

Answer: Thank Reviewer #1 for the good suggestion. We observed 57 ovules from *tcpDUO*, among them, 54 ovules displayed abnormal morphology. Among the 54 abnormal ovules, 46 (85.2%) ovules exhibited excessive growth in both the outer and inner integuments, while 8 (14.8%) ovules had only excessive growth in inner integuments.

We added these data in the revised Figure 2 and in the Results in the revised manuscript as follows (Line 210-216).

“We observed 54 ovules produced excessive growth in the outer and/or inner integuments from 57 ovules of *tcpDUO*. Among the 54 abnormal ovules, 46 (85.2%) ovules exhibited excessive growth in both the outer and inner integuments (Fig. 2f), while 8 (14.8%) ovules showed only excessive growth in inner integuments, causing

defective ovules with a large micropyle and an outer integument that failed to cover portions of the inner integument (Fig. 2c, d, g).”

iii) The Figures from 2q to 2u are too small and it is not possible to observe the female gametophyte phenotype. None of the ovules shown in Figures 2s, t and u have the integuments defects that are shown in Fig 2g. This is in contrast with the sentence in line 207 “..... and an outer integument that frequently failed to cover portions of the inner integument.”

Answer: Figure 2l to 2u mainly exhibit the defective female gametophytes. We did not consider the length of the outer and/or inner integuments, because the length of the outer and/or inner integuments was not always correlated with the sterility of the female gametophytes. According to the comments of Reviewer #1, we reorganized the images and provided the indication and explanation of the abnormal length of the outer and inner integuments in the images. Because there is not enough space to magnify the images in Figure 2, and Reviewer #3 suggested that the female gametophyte phenotype should be moved to the Supplemental Figure, we moved the 2l and 2m to Supplementary Figure 3. Thank you.

iv) I believe that it is important to understand at which stage the homeotic conversion is taking place and which ovule tissues were involved in the homeotic conversion. Consequently, the authors must provide ovule morphological analysis of *tcpDUO* vs wt, at 22°C and 28°C from stage 8/9 to 12 (according to Smyth et al., 1990).

Answer: Thank Reviewer #1 for the good suggestion. Because *tcpDUO* and wild type had no ovule homeotic conversion at 22°C, we did not observe the ovules of *tcpDUO* and wild type under 22°C. Following the suggestion of Reviewer #1, we observed the ovules from the flower developmental stage 9 to stage 12 of *tcpDUO* and wild type under 28°C. The results showed that the ovule homeotic conversion took place at flower developmental stage 11 in *tcpDUO* under 28°C.

We added these data in revised Figure 3 and in the Results in the revised manuscript as follows (Line 254-261).

“To investigate the developmental stage at which the ovules of *tcpDUO* begin to be converted into carpelloid structures, we observed the ovules from the flower developmental stage 9 to stage 12 of *tcpDUO* and wild type control under 28°C. The results showed that all the ovules from wild type had no homeotic conversion (Fig. 3j-

n). The early ovules of flowers at stage 9 and 10 were also rather normal in *tcpDUO* (Fig. 3o-q). However, the ovules at flower developmental stage 11 began to be homeotically converted into carpelloid structures (Fig. 3r). Clear carpelloid structures were observed in the flowers at stage 11 and 12 in *tcpDUO* under 28°C (Fig. 3r, s).”

v) I think that the aborted seeds reported in Figure 1 are mainly unfertilized and/or not well-developed ovules rather than aborted seeds.

Answer: According to the comments, we replaced “the aborted seeds” with “the aborted ovules or seeds.” in the revised manuscript. Thank you.

5. The authors wrote “we treated the emasculated flowers of *tcpDUO* and wild-type control with HT (28°C). We found that almost all of the ovules in the *tcpDUO* pistils were converted into carpelloid structures (Fig. 3d)”. According to this sentence the authors have treated the flowers containing mature ovules (stage in which anthers can be removed) with HT (28°C). This description is different in respect to what they wrote in the materials and methods section where they wrote that they treat plants after bolting for 6 days and that they emasculated the plants after this treatment.

I suggest that the authors clearly describe at what flower stages they have applied the HT. They can refer to Smyth et al., 1990.

Furthermore, the description of the treatment and time of pistil analysis (morphological analysis and RNAseq) have to be consistent in the different sections of the results and in the material and methods.

Answer: We are sorry for the misleading sentence. In our study, the *tcpDUO* or wild-type control plants were first placed in the 28°C incubator for 6 days. Then the flowers were emasculated at developmental stage 9. After two days, the ovules, the carpelloid structures or pistils were observed or isolated for experiments. We rewrote the sentence and also revised the description of material treatment with high temperature in Methods as follows (Line 243-244, 719-720, 916-919). Thank you.

“we treated *tcpDUO* and wild-type control plants with HT (28°C).”

“For observation of the HT-induced ovule homeotic conversion phenotype of *tcpDUO*, bolting plants were transferred to 28°C chambers with a long-day condition for 6 days. Then the flowers were emasculated at developmental stage 9. After two days, the ovules, the carpelloid structures or pistils were isolated for observation.”

“To perform RNA-seq analysis, the bolting wild-type or *tcpDUO* plants were treated at 28°C for 6 d. Then the flowers at developmental stage 9 were emasculated, and the pistils were isolated after 2 DAE from the wild-type or *tcpDUO* plants treated with 22°C or 28°C for total RNA extraction.”

6. The *tcpDUO* mutant pistil has ovules converted into carpelloid structures. They authors reported that the ovule identity gene *STK*, does not change its expression in *tcpDUO* respect to wt referring to the RNAseq data. What about other genes expressed in ovules such as *BEL*, *INO*, *NZZ*, *WUS* ect?

I believe that to conclude that *STK* (expression in the *tcpDUO* ovules do not change, it is absolutely necessary include in situ hybridization showing evidence for this assumption.

Answer: Thank Reviewer #1 for the suggestion. To promote the accuracy, we performed RT-qPCR to test the expression change of *STK*, *BEL1* and *WUS* using cDNA from the ovules instead of pistils. These three genes were most related to the ovule identity phenotypes observed in *tcpDUO*. The results showed that the treatment of HT did not change the expression of *STK* in wild type. However, HT significantly repressed the expression of *STK* in *tcpDUO*, suggesting TCPs were important for maintaining the expression level of *STK* in ovules under HT. The expression of *BEL1* was a little down-regulated in the ovules of wild type or *tcpDUO* under HT. The expression of *WUS* was highly up-regulated by the loss of TCP functions in *tcpDUO*, and HT further increased the expression of *WUS* in *tcpDUO*, suggesting that TCPs play key roles in the suppression of *WUS* under normal or high temperature. This is consistent with the indeterminate growth of carpelloid structures in *tcpDUO* under HT.

We further performed in situ hybridization as suggested by Reviewer #1. The results showed that the expression of *STK* was decreased and was mainly observed in the funiculi in *tcpDUO*, when compared to that in wild-type control under HT.

We included the results in new Supplemental Fig. 8, 9 and 10. and in the revised manuscript as follows (Line 366-368, 392-398, 410-417, 448-450).

“In addition, the expression of *INNER NO OUTER (INO)* key for ovule development was significantly down-regulated in *tcpDUO* under HT (Supplementary Fig. 8).”

“Our RNA-seq data also showed that the master regulator *WUSCHEL (WUS)* was also significantly up-regulated in the pistils of *tcpDUO* (Supplementary Fig. 9a). RT-

qPCR using the ovules as materials confirmed that *WUS* was highly induced in *tcpDUO*, and HT further increased the expression of *WUS* in *tcpDUO* (Supplementary Fig. 9b), suggesting that TCPs play key roles in the suppression of *WUS* under normal or high temperature. This is consistent with the indeterminate growth of carpelloid structures in *tcpDUO* under HT.”

“We next tested the expression change of *STK* in ovules by RT-qPCR. The results showed that the treatment of HT did not change the expression of *STK* in wild type. However, HT significantly repressed *STK* in *tcpDUO* (Supplementary Fig. 10b). In situ hybridization assays confirmed that the expression of *STK* was decreased and was mainly observed in the funiculi in *tcpDUO*, when compared to that in wild-type control under HT (Supplementary Fig. 10c-e). These data suggest that TCPs are important for maintaining the expression level of *STK* in ovules under HT.”

“However, the expression of *BEL1* was not obviously altered in pistils of *tcpDUO* under normal temperature or HT (Supplementary Fig. 10a, c)”

7. The authors performed an EMSA shift assay to study TCP4 interaction to regulatory regions of *CRC* and *SPT*. I believe that a ChIP assay is necessary to confirm the data and to assess whether HT affects TCP4 binding to the DNA in vivo. Since the authors have a pTCP4::TCP4-GFP construct available that complements the mutant phenotype it is not clear why they use an in vitro EMSA assay instead of a in vivo ChIP analysis.

Answer: We used EMSA to confirm that TCP4 bound to the promoter regions of *CRC* and *SPT* *in vitro*, because our ChIP-seq data using 35S-MYC-mTCP4 in which *MYC* tag fusion with microRNA319 (miRNA319)-resistant *mTCP4* was driven by a CaMV 35S promoter had shown that TCP4 bound to the promoter regions of *CRC* and *SPT* *in vivo*. As we mentioned above, although TCP4pro-TCP4-GFP completely complemented the ovule identity conversion of *tcpDUO* under HT, the expression of TCP4-GFP was very low possibly because of the post-transcriptional down-regulation by miRNA319. So, it is hard to perform ChIP-PCR or ChIP-seq using TCP4pro-TCP4-GFP transgenic plants. Indeed, we tried ChIP-PCR using TCP4pro-TCP4-GFP, but the experiments were failed. We rewrote the sentence to explain the ChIP-seq data more clearly as follows (Line 381-383). Thank you.

“We previously identified thousands of TCP4 binding sites using chromatin immunoprecipitation assays with sequencing (ChIP-seq) with 35S-MYC-mTCP4 in which MYC tag fusion with microRNA319 (miRNA319)-resistant mTCP4 was driven by a CaMV 35S promoter³⁹”

8. It is quite peculiar that all the SEP3 and AG protein domains are able to interact with TCP4/TCP18 since these protein domains have very different structures. A negative control, like a not interacting TCP protein will be important to show the specificity and to exclude that the observed interactions are unspecific.

The conclusion that all these proteins can interact in ovules is also not supported. Experiments showing TCP4/18 co-expression in the ovule together with the other proteins tested is mandatory.

Answer: We agree with the comments proposed by Reviewer #1. As described above, we performed RT-qPCR analysis using the ovule tissues, the GUS staining of TCP4pro-TCP4-GUS and in situ hybridization of *TCP4* to demonstrate that the expression pattern of *TCP4* was co-expressed with *BEL1*, *AG* and *SEP3* in ovules.

We also think that it was unusual that all the truncated SEP3 and AG proteins interacted with TCP4/TCP18 in yeast two hybrid assays. We do not know the reason currently. Because the interactions between the truncated SEP3/AG proteins and TCP4/TCP18 was not necessary for the conclusion of our work, we decided to remove the data as suggested by Reviewer #3 in the revised manuscript. We hope that Reviewer #1 could agree with us. Thank you.

9. The tcpDUO bel and tcpDUO stk seem to have different phenotypes. According to YAMADA et al. (2019), BEL is important to repress the formation of an ectopic nucella from the chalaza as well as repressed in later stages homeotic conversions of ovule integuments into carpelloid structures. Furthermore, it has been reported that BEL is involved together with HD-ZIP factors in controlling WUS expression in the chalaza (Yamada et al., 2016). I think that the phenotype of the tcpDUO bel and tcpDUO stk must be described in more details and discussed considering what it is already know about STK and BEL function in chalaza and integument development.

Answer: Thank Reviewer #1 for the suggestion. According to the suggestion, we described the phenotypes of *tcpDUO bel* and *tcpDUO stk* in more details as follows (Line 521-526, 537-542, 576-586). We also discussed the differences in Discussion as suggested by Reviewer #1 as follows.

“In the carpelloid structures of *tcpDUO bell*, many ovule-like structures were initiated from the chalaza position (Fig. 7h-j). The integuments of the ovule-like structures continued to grow and was then converted into carpelloid structures with papilla cells (Fig. 7h, i). The carpelloid structures showed an indeterminate growth, because the new carpel-like structures continued to be produced in the inner of the old ones (Fig. 7h-j).”

“However, the carpels of the tredecuple mutant *tcpDUO stk* were full of carpelloid structures instead of ovules (Fig. 7l-n). The carpel-like structures were connected by even longer funiculi (Fig. 7l, n). In addition, stigmatic papilla cells were clearly observed at the tip of the carpelloid structures (Fig. 7l-n). Nevertheless, no ovule-like structures or secondary carpel-like structures were observed in *tcpDUO stk* (Fig. 7l-n).”

“Although all the ovules are converted into carpelloid structures in the carpels of *tcpDUO bell* and *tcpDUO stk*, there are differences between them. *tcpDUO bell* produced ovule-like structures in the carpelloid structures which displayed indeterminate growth, while *tcpDUO stk* produced even longer funiculi. These differences are consistent with the distinct functions between BEL1 and STK. It has been reported that BEL1 plays critical roles in repressing the formation of ectopic nucelli and the homeotic conversions of ovule integuments into carpelloid structures by regulating the proper expression of *PINI* and *WUS*⁵³. In this paper, we found that the expression of *WUS* was highly up-regulated in *tcpDUO*, indicating TCPs are also important for the regulation of *WUS*. However, *STK* was reported to suppress the elongation of funiculi and to maintain the identity of ovules⁵⁴.”

Minor

The *tcpDUO* phenotype analysis have to be described in more details : “about 18.01 %”, (line 179); “about 18% (line 184)”... ect the authors have to include how many pistil/siliques they have analyzed.

Answer: We added the detailed data in the description of *tcpDUO* as follows (Line 181-183, 187-188). Thank you.

“After pollination, the *tcpDUO* mutant produced even shorter siliques and displayed low fertility (about 18.01% of 1259 ovules from 30 siliques) (Fig. 1a-d, i, j).”

“while only about 18% ovules from 10 siliques of *tcpDUO* plants formed seeds following pollination with pollen from *tcpDUO* or wild-type control plants”

EC1.1, EC1.3 and EC1.5 are expressed in the egg cell. Therefore, it is expected that they are not expressed in the *tcpDUO* mutant since in this mutant the mature female gametophytes are not formed even at 22°C. It will be better to report the expression of other ovule specific genes. The observation that *CRC*, *HEC1*, 2 and 3 are upregulated in the *tcpDUO* mutant might be a consequence that many ovules are converted into carpelloid structures.

Answer: It was hard to say the up-regulation of *CRC*, *SPT* and *HECs* was the reason or the result of the conversion of ovules into carpelloid structures. Since *TCP4* directly regulates *CRC* and *SPT*, we incline to think that the up-regulation of *CRC*, *SPT* and *HECs* may cause the conversion of ovules into carpelloid structures. As suggested by Reviewer #1, we added the expression change of *INO* essential for ovule development in *tcpDUO* in the revised manuscript as follows (Line 366-368). Thank you.

“In addition, the expression of *INNER NO OUTER (INO)* key for ovule development was significantly down-regulated in *tcpDUO* under HT (Supplementary Fig. 8).”

Line 224: replace “differet” with “different”

Answer: We fixed it. Thanks.

Line 267: please, explain what you meant for “seed arrangement” because it is not clear.

Answer: We explained “seed arrangement” as suggested by Reviewer #1 as follows (Line 158-161). Thank you.

“The *tcpSEP* mutants generated ovules and seeds with no significant differences from those of wild-type control plants, except that the siliques of *tcpSEP* were shorter (Fig. 1a). In addition, the seeds are normally interdigitated in a silique of wild-type, while the seeds frequently arose consecutively from the same side of the locule in that of *tcpSEP* (Fig. 1b, c).”

I suggest including the position of the TDNA in supplemental Fig.3b to make the Figure clearer.

Answer: We added the T-DNA site in the revised Supplemental Fig. 6b (the previous Supplemental Fig. 3b). Thank Reviewer #1 for the good suggestion.

Reviewer #2 (Remarks to the Author):

In this manuscript the role of the Arabidopsis type II TCP transcription factors in ovule development is studied under high ambient temperature conditions. The authors conclude that the functioning of class II TCP proteins is essential to guarantee ovule development instead of the formation of carpelloid structures at high ambient temperatures. The authors generated an impressive duodecuple mutant to come to this conclusion. Subsequently, various molecular and biochemical assays were performed to shed light on the underlying molecular-mode-of-action. The majority of these experiments have been well performed, but as further explained below, in some cases the set-up or interpretation of data needs attention.

1. The authors generated a duodecuple mutant, in which all class II TCPs and the class I TCP16 gene are modified. For the phenotyping, the focus is directly on, and limited to ovule development. Only in the last sentence of the discussion, the authors indicate that this mutant has additional phenotypes, as expected. It is good to focus; however, to judge whether the observed phenotypes are not just pleiotropic effects due to overall strong phenotypic alterations, the authors should at least show the effect of these multiple mutations at whole plant level.

Answer: We added the mature *tcpSEP*, *tcpDUO* mutants and the wild-type control in the revised Supplemental Fig.2 and in the results as follows (Line 173-175). Thank Reviewer #2 for the suggestion.

“The *tcpDUO* generated waving leaves and displayed late flowering as that observed in *tcpSEP* when compared to the wild-type control (Supplementary Fig. 2a, b).”

2. Line 263/264: ‘.....and their transcript levels were not obviously affected by HT, with the exception of TCP14.’. I believe that at least also TCP7 is different. 'Not obviously affected' is a vague term. Please check for each TCP gene whether there is a statistical difference and change the text accordingly.

Answer: Thank Reviewer #2 for the suggestion. According to the suggestion, we re-analyzed the expression fold change of *TCP* genes in wild type treated with 22°C Vs 28°C (Supplemental Fig. 3C). The results showed that no *TCPs* were significantly down-regulated or up-regulated by high temperature in pistils. We included the results in the revised manuscript and rewrote the results as follows (Line 296-299).

“To determine the effects of HT on the expression level of *TCPs*, we analyzed the expression fold change of *TCP* genes in wild type treated with 22°C Vs 28°C

(Supplementary Fig. 6c). The results showed that no *TCPs* were significantly regulated by HT (false discovery rate < 0.01; fold-change ≥ 2.0 or ≤ -2.0).”

3. Line 266-268. In contrast to a very detailed and well-performed phenotypic analysis of the ovules in the duodecuple mutant, expression analysis for the *TCPs* under study is very limited. Only for *TCP4* some low resolution GUS staining pictures are provided in the supplements. It is not needed to show the detailed spatial and temporal expression pattern of each individual *TCP* gene during ovule development, but a more elaborate investigation of *TCP4* expression linked to the proposed functions would be essential. For example: What about GFP signal in the complementing *TCP4*pro-*TCP4*-GFP line? Linked to this: I do not see GUS signal in supp Fig. 3C!?

Answer: Thank Reviewer #2 and Reviewer #1 for the suggestion. According to the suggestion, we used different methods to analyze the expression pattern of *TCP4* in ovules. We performed RT-qPCR analysis using the ovule tissues instead of pistils, the results showed that *TCP4* was expressed in ovules. GUS staining of *TCP4*pro-*TCP4*-GUS showed that *TCP4* protein was clearly expressed in ovules. We further performed in situ hybridization of *TCP4*. The results showed that *TCP4* was highly expressed in both nucellus and integuments during ovule development, consistent with the expression patterns shown by GUS staining of *TCP4*pro-*TCP4*-GUS (Supplementary Fig. 7). We also analyzed the expression pattern of some other *TCPs* in ovules.

We included these data in Supplementary Fig. 7 and in the Results of the revised manuscript as follows (Line 300-314).

“We collected mature ovules from pistils of wild type plants treated with 22°C or 28°C to test the expression of Class II *TCPs* and *TCP16* using reverse transcription quantitative PCR (RT-qPCR). The results showed that almost all these *TCPs* except *TCP1* and *TCP17* were expressed in mature ovules (Supplementary Fig. 7a). We further cloned a 3043-bp promoter of *TCP1*, a 2466-bp promoter of *TCP12*, a 2428-bp promoter of *TCP18* and a 2571-bp promoter of *TCP24*. We generate *TCP1*pro-GUS, *TCP12*pro-GUS, *TCP18*pro-GUS and *TCP24*pro-GUS constructs in which these promoters were used to drive *GUS* reporter gene, respectively. GUS staining showed that the expression of *TCP1* was hardly detected (Supplementary Fig. 7b, c), while the expression of *TCP12*, *TCP18* and *TCP24* was clearly observed in the early ovules (Supplementary Fig. 7d-i). Staining experiments with the previously reported *TCP4*pro-*TCP4*-GUS transgenic line confirmed that *TCP4* protein was expressed during ovule development (Supplementary Fig. 7j-m)³⁶. In situ hybridization showed

that *TCP4* was highly expressed in nucellus and integuments, consistent with the GUS staining of TCP4pro-TCP4-GUS (Supplementary Fig. 7o-r).”

4. Line 280/281: ‘These data suggest that TCPs may be important mediators of gene repression in pistils under HT.’. The authors are already careful, but whereas we deal here with the final steady state expression levels, observed effects can be completely indirect and opposite of eventual initial direct effects. I would remove this suggestion.

Answer: We removed the sentence in the revised manuscript as suggested by Reviewer #2. Thanks.

5. Line 365-368, Yeast two hybrid results. The fact that TCP4 and TCP18 bind to each individual domain of SEP3 and AG could also mean that the observed interactions are not specific. I miss here some controls. Could other MADS proteins be included? And what about STK and SHP1/2? Are these also interacting? If so, what does that mean for the proposed model!?!?

Answer: The comments proposed by Reviewer #2 was the concern also proposed by Reviewer #1. We also think that it was unusual that all the truncated SEP3 and AG proteins interacted with TCP4/TCP18 in yeast two hybrid assays. We do not know the reason currently. Because the interactions between the truncated SEP3/AG proteins and TCP4/TCP18 was not necessary for the conclusion of our work, we decided to remove the data in the revised manuscript as suggested by Reviewer #3. We hope that Reviewer #2 could agree with us. Thank you.

6. Line 400/401: ‘....suggesting that TCPs and BEL1 formed a complex with AG-SEP3 to inhibit the activity of AG-SEP3 quartets.’. This could be, but the authors do not provide any evidence that the effect is specifically on quartet formation. Note that the performed experiments do not allow to say anything about the exact stoichiometry of the formed complexes! Please modify.

Answer: We deleted the sentence to avoiding misunderstanding. Thank you.

7. Line 409: ‘.....the fluorescence was significantly increased’. I am not convinced. Please quantify, perform a proper statistical test, and draw the conclusion based on sufficient repetitions.

Answer: We repeated the experiments three times and quantified the fluorescence as suggested by Reviewer #2. We included the results in the revised Fig. 6f and in the newly added Supplementary Fig. 11. Thanks.

8. The model presented in figure 8 and also the concluding sentence in the discussion: ‘We further showed that TCP4 directly interacted with BEL1 and AG-SEP3 to stabilize the BEL1-AG-SEP3 complex and negatively regulated the function of AG-SEP quartets.’, suggest that the TCP4 mode of action is to interfere with AG-SEP3 complex formation. This could indeed be the case, but how to link this to the observed differences in expression of supposed downstream genes and showing that TCP4 can bind to the promoters of these genes??? This suggests that the observed phenotypic effects depend on TCP4-mediated DNA binding! I do not expect the authors to figure out what is exactly happening and causal, but this discrepancy should at least be dealt with in the discussion.

Answer: We deleted the sentence to avoiding misunderstanding. As suggested by Reviewer #2, we added some discussion in the Discussion as follows (Line 634-639). Thank you.

“In addition, because TCP4 directly binds to the promoter region of carpel-related genes such as *CRC*, *SPT* and *HEC1*, TCP4 may also act as transcription factor to inhibit these genes and to prevent the carpelloid conversion during ovule development by recruiting the transcriptional repressor TCP INTERACTOR CONTAINING EAR MOTIF PROTEIN 1 (TIE1) and the corepressor TPL/TPRs”

9. Line 586/587: ‘and both TCP4 and BEL1 repressed the expression of carpel identity genes by interacting with the AG-SEP3 quartet.....’. See my comment above. No evidence is provided that it acts directly on quartet formation! Further more, only circumventional evidence is provided that the proteins are present in one large complex.

Answer: We revised the sentence as follows (Line 667-669) to avoiding misunderstanding. Thank you.

“Our biochemical data showed that TCP4 interacted with BEL1, and both TCP4 and BEL1 repressed the expression of carpel identity genes.”

Minor points

- Line 32/33: ‘Disruption of all Class II *TCPs* in a duodecuple *tcp2/3/4/5/10/13/17/24/1/12/18/16* (*tcpDUO*) mutant....’. Please make clear that *TCP16* is not a class II *TCP*. For example by changing this sentence into: ‘Disruption of all Class II *TCPs* and the Class I *TCP16* gene in a duodecuple *tcp2/3/4/5/10/13/17/24/1/12/18/16* (*tcpDUO*) mutant.....’.

Answer: Thank Reviewer #2 for the suggestion. We fixed it.

- Line 170: ‘member but contains a *TCP* domain more similar to that of Class II *TCP* proteins....’. Please provide the correct reference for this statement:

<https://doi.org/10.1074/jbc.M111.256271>

Answer: We added the proper reference. Thank you.

- Line 271/272: ‘indicating that no complete transcripts for the 12 *TCPs* were generated in the *tcpDUO* null mutant’. This is true for the majority of the *TCPs*, but cannot be seen for the CRISPR mutagenised *TCP24* gene in the current representation. Also for the *TCPs* for which no expression was found it cannot be concluded. Please be more precise and rephrase accordingly.

Answer: Thank Reviewer #2 for the suggestion. We rephrased the sentences as follows in the revised manuscript.

“In contrast to the *TCP* reads from wild-type plants, which mapped to the whole corresponding *TCP* genomic sequences, the 9 T-DNA insertion mutated *tcp* reads from *tcpDUO* did not, indicating that no complete transcripts for the 9 *TCPs* were generated in the *tcpDUO* null mutant (Supplementary Fig. 6b). In addition, no transcripts of *TCP1* or *TCP16* were found in wild type and *tcpDUO*, and the transcripts of *TCP24* carrying one-bp-insertion were decreased in *tcpDUO* (Supplementary Fig. 1b, 3b).”

- Label figure 3H. *tcpUO* should be *tcpDUO*.

Answer: We fixed it. Thank you so much.

- Label Supp fig 4B: *tcpDOU* should be *tcpDUO*.

Answer: We fixed it. Thank you so much.

Reviewer #3 (Remarks to the Author):

This complex manuscript uses a range of methods to examine the roles of TCP genes in ovule development. Much of the work is well-described and supports conclusions of the manuscript, other parts of the work are more difficult to follow, and some appear insufficient to support associated conclusions. I will not separate out the positive and less positive evaluation of the work, but rather will follow the order of the manuscript in my evaluation of the presented work.

They show that combined mutation in 12 TCP genes (*tcpDUO*) has significant effects on pistil development and the fertility of ovules. The fruits are of reduced length, but this appears to result from a reduction in seed set. TCP4 is able to complement these effects. The *tcp4DUO* mutant line is shown to produce carpelloid ovules at high frequency in elevated temperatures, and this effect is also rescued by the TCP4 gene.

RNAseq experiments show that carpelloid ovules express carpel developmental regulators. This is to be expected and so is not a particularly interesting result. They show by ChIP that TCP4 appears to bind to regions flanking carpel development genes CRC, SPT, and HEC1. This is interesting as it suggests possible direct effects on expression of these genes. This binding is further supported by gel shift assays in Fig. 4k. Given that expression of these genes is elevated in *tcpDUO* mutants it is a good hypothesis that the TCP proteins repress expression of these genes, but this is not directly demonstrated. The genes could be indirect targets of such repression. For Fig. 4, only panels e, g, h, j and k seem relevant to the text and perhaps the other panels should be moved to supplementary figure.

Answer: Thank Reviewer #3 for the comments and suggestion. We reorganized Fig. 4 and moved the panel a, b and c to Supplemental Fig. 5.

They show by three methods that TCP4 interacts with AG and SEP2 proteins, strongly supporting these interactions. They do the same to show interaction between TCP4 and BEL1.

They use transient expression studies to show that TCP4 can repress expression from the CRC and SPT promoters, and that this repression is dominant over expression enhancement induced by a combination of AG and SEP3 proteins. That this is not just an interaction with AG-SEP3 complex is shown by the repression of expression occurring even in the absence of AG and SEP3 (this conflicts with what they say on

lines 383 – 385). They also use gel shift assays to show direct interaction of TCP4 with these promoter regions.

Answer: Thank Reviewer #3 for the comments. We rephrased the sentence as follows. “These results suggest that TCP proteins prevent AG-SEP3 activating the transcription of carpel-related genes in ovules.”

Using split luciferase assays they provide evidence that TCP4 can increase the affinity of AG/SEP3 for BEL1. However, these qualitative assays are not completely convincing as they can vary from infiltration to infiltration and that could explain the variability. So, while this is evidence, it does not allow a firm conclusion.

Answer: We repeated the experiments three times and quantified the fluorescence as suggested by Reviewer #3 and Reviewer #2. We included the results in the revised Fig. 6f and in the newly added Supplemental Fig. 11. Thanks.

In Line 420 – 437 they claim instability of BEL1 is shown by Fig. 6 f – h, but this is not convincing. The control band appears to change in concert with the BEL1 containing bands. The 0 time controls in the three panels vary significantly and they do not provide an explanation for this. Based on the description of the experiments it would have been expected that these zero time points would be similar. Also, they appear to measure steady-state protein levels without inhibition of protein synthesis, so protein stability is not actually measured. This experiment does not appear to provide evidence of differential stability of BEL1 in high temperature. Perhaps the experiment is misunderstood, but then it needs to be more clearly explained.

Answer: Thank Reviewer #3 for the comments. We are sorry for that we have not described more clearly, especially about the P2A sequence. P2A is a sequence from porcine teschovirus-1. The 2A peptide mediates self-cleavage, and thus results in one mRNA containing a P2A sequence to be translated into two separate proteins. We introduced a P2A sequence between BEL1-FLAG and GFP to evaluate the stability of BEL1-FLAG under high temperature. The zero time controls in the three panels varied, because the cleavage efficiency mediated by P2A was different in every experiment of the transient expression of BEL1-FLAG-P2A-GFP. We think the different cleavage efficiency did not affect the results of experiments, because the size of the uncleaved BEL1-FLAG-P2A-GFP, the cleaved BEL1-FLAG and GFP is different. The abundance of both the uncleaved BEL1-FLAG-P2A-GFP and the cleaved BEL1-FLAG

could be evaluated to determine the stability of BEL1. The abundance of proteins was compared with the zero time controls in the three different experiments, respectively. So, the difference of the three zero time controls did not affect the evaluation.

It is true that we should inhibit protein biosynthesis before the protein stability is measured. To provide more evidence to determine the stability of BEL1 under high temperature, we generated 35Spro-BEL1-FLAG in which *BEL1* fusion with the sequence encoding FLAG tag was driven by CaMV 35S promoter. We transformed 35Spro-BEL1-FLAG into Arabidopsis. We treated the flowers of 35Spro-BEL1-FLAG transgenic plants with a protein synthesis inhibitor cycloheximide (CHX). The results showed that HT treatment leads to obvious degradation of BEL1, while the proteasome inhibitor MG132 inhibited the degradation of BEL1 protein. These results further confirmed that BEL1 was unstable under HT.

We included the results in the Fig. 6i and the revised manuscript and rewrote the paragraph for more clarity as follows (Line 476-499).

“We generated 35S-BEL1-FLAG-P2A-GFP, in which expression of *BEL1-FLAG* and *GFP* (connected by a sequence encoding P2A) was driven by a CaMV 35S promoter. P2A is a sequence from porcine teschovirus-1. The 2A peptide mediates self-cleavage, and thus results in one mRNA containing a P2A sequence to be translated into two separate proteins^{51,52}. Because P2A self-cleavage caused the expressed BEL1-FLAG-P2A-GFP to be cleaved into equal amounts of BEL1-FLAG and GFP, the abundance of GFP served as the internal control for the evaluation of BEL1-FLAG stability. We transiently expressed 35S-BEL1-FLAG-P2A-GFP in tobacco leaves and treated the leaves with HT. The results showed that the abundance of both the uncleaved BEL1-FLAG-P2A-GFP product and BEL1-FLAG cleaved from it decreased after HT treatment, while the abundance of GFP cleaved from the fusion protein was not altered, indicating that BEL1 was unstable under HT (Fig. 6g). To determine whether the stability of BEL1 under HT was dependent on the 26S proteasome, we treated the samples with MG132 (a proteasome inhibitor) and HT. The degradation of BEL1 under HT was highly inhibited by MG132, suggesting that the 26S proteasome was required for the degradation of BEL1 (Fig. 6h). To provide more evidences to determine the stability of BEL1 under HT, we generated 35Spro-BEL1-FLAG in which *BEL1* fusion with the sequence encoding FLAG tag was driven by a CaMV 35S promoter. We

treated the flowers of 35Spro-BEL1-FLAG transgenic plants with a protein synthesis inhibitor cycloheximide (CHX). The results showed that HT treatment led to the obvious decrease of BEL1, while the proteasome inhibitor MG132 inhibited the degradation (Fig. 6i), further confirming that BEL1 was regulated by the degradation via 26S proteasome under HT. We further co-expressed miR319-resistant *mTCP4* with 35S-BEL1-FLAG-P2A-GFP and found that TCP4 might increase the stability of BEL1 under HT (Fig. 6j), consistent with the results showing that TCP4 promoted the interaction between BEL1 and AG-SEP3.”

They examine combinations of *tcpDUO* mutations with newly generated alleles of *bell* and *stk* and compare the effects to the single mutants. They do not say what allele of *bell* is shown in Fig. 7 e and f. The ovules in these images don't look like any *bell* ovules I have seen (I have seen a lot). The outer layers have surface features like normal outer integuments, and there appear to be separate inner and outer integument structures, which is never the case in strong *bell* mutants. In contrast, the secondary ovules in Fig. 7 j look exactly like *bell* ovules from previous publications. Also, past experience with *bell* indicates that waiting longer (developmentally) leads to more ovules becoming carpelloid (as these authors later note on lines 631 – 632). The ovules in some of the figures showing carpelloid ovules in Fig. 7 appear to be in older pistils than those in panels showing less carpelloid conversion. The time after floral opening should be described for the structures imaged in these figures to allow evaluation of the effects independent of carpel aging. Still, the formation of carpelloid structures appears significantly enhanced in the combination of *tcpDUO* with *bell* over either gene mutant alone. Similarly, they show that *tcpDUO* in combination with *stk* makes carpelloid ovules, which are not seen in the *stk* single mutant. Thus they show that TCP proteins help maintain ovule structure and prevent carpelloid conversion of ovules, especially at elevated temperatures.

Answer: Thank Reviewer #3 for the comments. The allele of *bell* shown in Fig. 7 e and f was *bell-8*. We replaced *bell-8* with a one-bp-insertion line *bell-1*.

As also suggested by Reviewer #1, we observed the ovules from the flower developmental stage 9 to stage 12 of *tcpDUO* and wild type under 28°C. The results showed that the ovule homeotic conversion took place as early as flower developmental

stage 11 in *tcpDUO* under 28°C, indicating that the effects of TCPs on ovule identity may be independent of carpel aging.

We added these data in revised Fig. 3 and in the Results in the revised manuscript as follows (Line 254-261).

“To investigate the developmental stage at which the ovules of *tcpDUO* begin to be converted into carpelloid structures, we observed the ovules from the flower developmental stage 9 to stage 12 of *tcpDUO* and wild type control under 28°C. The results showed that all the ovules from wild type had no homeotic conversion (Fig. 3j-n). The early ovules of flowers at stage 9 and 10 were also rather normal in *tcpDUO* (Fig. 3o-q). However, the ovules at flower developmental stage 11 began to be homeotically converted into carpelloid structures (Fig. 3r). Clear carpelloid structures were observed in the flowers at stage 11 and 12 in *tcpDUO* under 28°C (Fig. 3r, s).”

Their model is reasonable and most of it is supported by their data, but it does not incorporate all of their data. It is consistent with TCP proteins binding to BEL1, SEP and AG, but does not incorporate the apparent ability of TCP to bind to SEP and AG when BEL1 is not present. It also does not incorporate the ability of TCP to directly bind the CRC and SPT promoters in the absence of other proteins. The observations lead to the question: is TCP4 a transcription factor or a protein interaction factor? It is possible it is both and perhaps both activities should be discussed. The model incorporates differential protein instability of BEL1, but unless I misunderstand their assay, this has not been demonstrated in the manuscript. This is not essential to the model, but is part of their explanation of the apparent temperature-dependence of the carpelloid ovule formation.

Answer: We think that TCP4 may act as both protein interaction factor and transcription factor. We discussed this point in the Discussion as suggested by Reviewer #3 as follows (Line 634-639).

“In addition, because TCP4 directly binds to the promoter region of carpel-related genes such as *CRC*, *SPT* and *HEC1*, TCP4 may also act as transcription factor to inhibit these genes and to prevent the carpelloid conversion during ovule development by recruiting the transcriptional repressor TCP INTERACTOR CONTAINING EAR MOTIF PROTEIN 1 (TIE1) and the corepressor TPL/TPRs⁶⁰.”

As mentioned above, we provided more evidences to supported that BEL1 was regulated by degradation by HT. We generated 35Spro-BEL1-FLAG in which *BEL1*

fusion with the sequence encoding FLAG tag was driven by CaMV 35S promoter. We transformed 35Spro-BEL1-FLAG into Arabidopsis. We treated the flowers of 35Spro-BEL1-FLAG transgenic plants with a protein synthesis inhibitor cycloheximide (CHX). The results showed that HT treatment leads to the obvious degradation of BEL1, while the proteasome inhibitor MG132 inhibited the degradation of BEL1 protein. We included the results in the revised manuscript as follows (Line 489-496). Thank you.

“To provide more evidences to determine the stability of BEL1 under HT, we generated 35Spro-BEL1-FLAG in which *BEL1* fusion with the sequence encoding FLAG tag was driven by a CaMV 35S promoter. We treated the flowers of 35Spro-BEL1-FLAG transgenic plants with a protein synthesis inhibitor cycloheximide (CHX). The results showed that HT treatment led to the obvious decrease of BEL1, while the proteasome inhibitor MG132 inhibited the degradation (Fig. 6i), further confirming that BEL1 was regulated by the degradation via 26S proteasome under HT.”

I provide below more detailed comments for suggestions for improvement or correction of the manuscript.

Lines 79 – 89 There is more detail provided than needed on embryo sac development. Development of the embryo sac is not a real subject of this manuscript.

Answer: We rewrote the paragraph more briefly as follows (Line 78-84). Thank you.

“In Arabidopsis, megasporogenesis begins with a subepidermal cell specified to develop into a megaspore mother cell (MMC) in the distal nucellus. The MMC undergoes meiosis to give rise to the functional megaspore (FM) which further develops into the female gametophyte⁹. The funiculus coordinately elongates, and the inner and outer integuments grow asymmetrically to envelop the nucellus, leaving a micropyle for pollen tube entry to complete the double fertilization⁹.”

Line 105-106 For the statement “Overexpression of AG causes the transformation of ovules into carpelloid structures, possibly due to increased activity of AG-SEPs quartets in ovules.” They cite ref. 16, Bowman, Drews and Meyerowitz (1994). But that paper doesn’t address this topic. That ectopic AG expression produces carpelloid ovules in Arabidopsis was shown by Ray et al. (1994) PNAS 91:5761-5766, and in tobacco by Mandel, et al. (1992) Cell 71:133-143. The reference should be corrected. Notably, Ray

et al. also hypothesized that BEL1 inhibited AG carpel promoting activity in ovules, a point of the current manuscript.

Answer: Thank Reviewer #3 for the suggestion. We cited the two references in the revised manuscript.

Line 116 This isn't really true. The cited paper showed that *bell* ovules are aberrant at the normal growth temperature of 22° but that the effects are less severe at the low temperature of 16°. It may be that particular *bell* allele is temperature sensitive. Also, there are many changes for a plant grown at 16°. 22° is not an elevated temperature for *Arabidopsis*.

Answer: We rewrote the paragraph more accurately as follows (Line 111-113). Thanks.

“The *bell* produced ovules that could be converted into carpelloid structures at the normal growth temperature (22°C), but the phenotype is less severe at the low temperature (16°C)²⁵”

Line 126 The TCP family is not “highly conserved”, in fact members are quite divergent. Perhaps they meant to say the TCP family is “present in all plants”.

Answer: We fixed it as suggested by Reviewer #3 as follows (Line 118-119). Thanks.

“TEOSINTE BRANCHED 1/CYCLOIDEA/PCF (TCP) proteins belong to a plant-specific transcription factor family that is present in all plants²⁴”

Line 206 – 208. It actually looks like the main defect is a failure in the outer integument (OI) to grow sufficiently. Look at the cells at the edge of the OI, they are long in WT and very small in *tcpDUO*. The failure of the OI to grow could allow the II to grow because of the absence of a confining OI. This is the case in other ovule mutants where OI growth is deficient. Still, it does result in what they describe.

Answer: Thank Reviewer #3 for the comment.

Line 227 the failure in female gametophyte (FG) formation could be a secondary effect of the aberrant sporophytic parts of the ovules. This is the case for mutants like *bell* and *ant* where all FG form normally in heterozygous plants, where half of FG would be haploid mutants. So the effect on the mutant is on the ovule, not the FG directly.

Answer: We rewrote the sentence as follows (Line 234). Thank you.

“These results suggest that TCPs play pivotal roles during ovule development.”

Line 408 – 412 The signal with TCP4 present does appear stronger, but how much variation is there between individual leaf infiltrations of the with the same input. It is hard to conclude that this is real enhancement without more duplication or quantitation.

Answer: fluorescence as suggested by Reviewer #3 and Reviewer #2. We included the results in the revised Fig. 6f and in the newly added Supplemental Fig. 11. Thanks.

Line 420 – 421 by “expression level” I think they mean “steady state level of mRNA”. Expression could refer to the amount of protein. They should make this clear.

Answer: We revised “the expression level” as “the transcript level of *BEL1*” to avoid misunderstanding in the revised manuscript as follows (Line 473-474). Thanks.

“because the transcript level of *BEL1* was not significantly altered by HT treatment (Supplementary Fig. 10a, c).”

Line 422 What is “P2A”? I can find no reference for this. Two prolines and an alanine? Or is it some peptide sequence? Why is it cleaved? Is it an intein? Clearly it is not cleaved efficiently as the whole protein is visible in every lane of Fig. 6 f – h. More information is needed here.

Answer: As described above, we explained the “P2A” and added the references. It is true that the cleavage efficiency mediated by P2A was different in different experiments, but this did not affect the results. We added more information and further provided more evidences to support that *BEL1* was regulated by protein degradation under HT using the stable transgenic plants expressing *BEL1*-FLAG. Thank you for the comment.

Line 420 – 437. And Fig. 5 f – h. Not convinced of the conclusions here at all. Why is the 0 time point so different between f and g and f an h? Almost no cleaved *BEL1*-FLAG even though GFP levels are nearly constant. In g the control GFP band looks to change as much as the whole protein and the free *BEL1*-FLAG so that an effect of MG132 is not at all clear. Same is true in h for proposed effect of TCP4. These results are not at all convincing.

Answer: Thank Reviewer #3 for the comments. As explained above, the zero time controls in the three panels varied, because the cleavage efficiency mediated by P2A was different in every experiment of transient expression of *BEL1*-FLAG-P2A-GFP.

We think the different cleavage efficiency did not affect the results of experiments, because the size of the uncleaved BEL1-FLAG-P2A-GFP, the cleaved BEL1-FLAG and GFP is different. The abundance of both the uncleaved BEL1-FLAG-P2A-GFP and the cleaved BEL1-FLAG could be evaluated to determine the stability of BEL1. The abundance of proteins was compared with the zero time controls in the three different experiments, respectively. So, the difference of the three zero time controls did not affect the evaluation.

The “Almost no cleaved BEL1-FLAG even though GFP levels are nearly constant.” just proved that BEL1-FLAG was degraded and unstable after being cleaved from BEL1-FLAG-P2A-GFP, while the control GFP protein was not degraded after being cleaved from BEL1-FLAG-P2A-GFP.

Additional evidences was also provided to confirm the degradation of BEL1 under HT. Thank you.

Line 932 – 933 Year is incorrect for reference 40.

Answer: The Year (2015) is right. The 2014 is an online version. Thanks anyway.

> Cell Res. 2015 Jan;25(1):121-34. doi: 10.1038/cr.2014.145. Epub 2014 Nov 7.

The molecular mechanism of sporocyteless/nozzle in controlling Arabidopsis ovule development

Baoye Wei ¹, Jinzhe Zhang ¹, Changxu Pang ¹, Hao Yu ¹, Dongshu Guo ¹, Hao Jiang ¹, Mingxin Ding ¹, Zhuoyao Chen ¹, Qing Tao ¹, Hongya Gu ², Li-Jia Qu ³, Genji Qin ¹

Line 1282 – 1285 Figs. f, g, h are mislabeled e, f, g in the legend.

Answer: We fixed it. Thank you so much.

They often fail to say what was done in an experiment. For example. Supp. Fig. 4. Is this RNA seq data? Is this QRTPCR? There is no information on what experiment was performed to get the reported numbers. I am guessing RNA seq, but there is no way to tell from the information provided. The method could be added to the last sentence of the legend “...of three biological replicates of [whatever this experiment was]. They need to always describe what the experiment is for all of the figures. They should check over the whole manuscript for this.

Answer: Thank Reviewer #3 for the suggestion. We checked all the Figures and briefly described the experiments in the Figure legends.

SUMMARY

This is a very interesting manuscript and most of the conclusions are supported by the data. In particular, they show an important function for TCP genes in maintenance of ovule identity and show that this results in part from interactions with known ovule identity genes. This importance is more apparent at elevated temperature. If the poorly supported conclusions were removed, it would still be an interesting manuscript that makes a significant contribution to the field.

Answer: Thank Reviewer #3 very much for the positive comments.

REVIEWER COMMENTS

Reviewer #2 (Remarks to the Author):

In the revised version of the manuscript 'Arabidopsis TCP4 transcription factor inhibits high temperature induced homeotic conversion of ovules', the authors have carefully addressed the comments and suggestions of the reviewers. A large number of new and additional experiments were done and reported, and various points have been addressed in a proper way and after full satisfaction. Nevertheless, the manuscript still has some flaws. More importantly, issues with a direct effect on the conclusions remain, and I therefore still have strong doubts about part of the conclusions (proposed molecular mode of action of the TCPs) and the model presented at the end of the manuscript.

Points of attention in the revised manuscript (In order of the text and not in order of importance):

- The authors link their research to thermomorphogenesis in the abstract (e.g. lines 69-71) and discussion (lines 644-650). Considering the definition of thermomorphogenesis, which is also used by the authors (Thermally induced developmental and morphological changes in plants, which are frequently represented by accelerated stem elongation, increased leaf hyponasty, and formation of long, thin leaves.), this is correct. However, the authors couple this to global warming and morphological changes of plants and animals (only reproductive traits) upon exposure to ambient high temperatures. In plants, these are supposed to be adaptive traits to deal with heat. In this case, the morphological changes are not occurring in a 'wildtype' plant, but only in the tcpDUO mutant background. As such, I see it as an increased temperature sensitivity due to the lack of numerous functional TCP genes. Therefore, I would rephrase the introduction (first paragraph) and discussion (Lines 644-655) and completely remove the discussion of the underlying mechanism in turtles (H3K27me3 deposition) because it is not relevant and it is out of context.
- In line with above comment, the reasoning provided in the results section, lines 283-285 'Although the effects of HT on the growth of hypocotyls, petioles, roots and male organs have been studied extensively, the mechanisms underlying the effects of HT on female organs, including homeotic conversion of ovule identity, remain largely unknown.'. As the authors show, these morphological effects are not present in wild-type plants! Hence, you can also not study the underlying mechanisms!
- Results, lines 160, 161: 'while the seeds frequently arose consecutively from the same side of the locule in that of tcpSEP (Fig. 1b, c)'. Is this true? At least for me it is not clear from the presented pictures. Could the observed altered arrangement of ovules not simply be because of the lack of elongation of the pistil (carpel wall and septum)? It seems to me that the seeds are still alternating in position).
- Results, lines 181, 182: 'the tcpDUO mutant produced even shorter siliques'. I don't think that the mutant produces even shorter siliques, but that the authors mean that the short siliques don't stretch that much as in wildtype and that consequently the final silique length is even shorter in comparison to wildtype. If so, please rephrase.
- Results, line 383 'Venn diagram analysis showed....'. This should be something like 'Comparative analysis showed.....'. A Venn diagram is only a way to visualize the outcome of the comparison.
- Results, lines 390-391 '....., indicating that TCP4 directly represses CRC and SPT to control ovule identity.'. This cannot be concluded based on these experiments and observations! The EMSA shows indeed binding and there is a significant expression difference between the mutant and wildtype. However, the latter is only a correlation and not proof that the observed expression difference is caused by TCP4!

- Results, line 409 'the expression of STK, SHP1 and SHP2 were not obviously changed'. This is a vague non-scientific expression. Furthermore, the change is actually the same as observed in the qRT-PCR shown in Fig S10b!
- Results, lines 416, 417 'These data suggest that TCPs are important for maintaining the expression level of STK in ovules under HT.'. Indeed! And this could probably explain the observed homeotic conversions at 28 degrees Celsius, considering the known function of STK (see my comments below on the currently proposed molecular and biochemical mode of action of the TCPs).
- Based on the result presented in lines 541, 542 'Nevertheless, no ovule-like structures or secondary carpel-like structures were observed in tcpDUO stk (Fig. 7l-n)', the authors conclude in line 542, 543 'These results unequivocally demonstrate that TCP transcription factors play a critical role in maintaining the identity of ovules.'. This is a weird conclusion based on this result!? It is about the additional effect of a stk mutation and the function of STK!
- Results lines 553-558 'In wild-type ovules under normal temperature (22°C), TCPs form complexes with BEL1 and AG-SEP quartets to inhibit the activity of AG-SEP, and the predominant activity of AG-SEPs-STK/SHPs quartets promotes the expression of ovule-related genes to determine ovule identity (Fig. 8a)'. This is not clear (what is the mechanism?) and according to me, this model is not very plausible. How can the interaction of BEL1 with AG-SEP3 have an effect on the activity of the other complex? This requires an explanation and substantiation by experimental data.
- Lines 571-573 'In the tredecuple mutant tcpDUO stk, the loss of STK function reduces the activity of AG-SEPs-SHPs, and BEL1 is insufficient to repress the activity of AG-SEP quartets without the help of TCPs'. This is a very strange synthesis. How can the loss of STK have an effect on the BEL1-AG-SEP3 complex in which it doesn't take part?
- The discussion in lines 625-634 is very speculative and not supported by data. I propose to remove this part.
- Fig 4 legend: K should be H.
- Legend Fig. S9 'The FPKM from RNA-seq data indicated that WUS was induced in tcpDUO under normal temperature or HT'. Under normal temperature, no significant difference is observed so this cannot be concluded.

Points related to the rebuttal:

- Response to point 4, i), Reviewer #1: 'The staining with aniline blue showed that pollen germinated on the stigma and pollen tubes grew through styles in pistils from both tcpDUO and wild-type control (Supplementary Fig. 2e, f), indicating that the transmitting tract of tcpDUO was rather normal.'. Indeed, pollen germination seems not to be affected, but based on the presented pictures (Fig S2 e and f) ingrowth seems to be strongly reduced in the tcpDUO mutant. So based on what is presented, I do not agree with the conclusion.
- Response to point 4, v), Reviewer #1: Considering the new data on pollen tube staining in the tcpDUO mutant upon pollination with wild-type pollen, the possibility that many ovules are unfertilized is very plausible and should be included as well.
- Point 8, Reviewer #1 and point 5, Reviewer#2: I do not agree! You cannot simply remove data that you do not expect or cannot explain. The authors should perform the suggested additional negative control (interaction tests with other unrelated TCP), and even more important, an investigation of the interaction between TCP4/18 and STK/SHPs. In case interaction is also detected with STK, the presented model which is for a large part based on these Y2H outcomes doesn't hold!
- Response to point 2, Reviewer #2: The authors added the sentence 'The results showed that no TCPs were significantly regulated' and this is not completely correct. Please change to something like 'The results show that none of the TCPs was more than 2 fold changed.'.

- Response to point 3, Reviewer #2: The authors performed extra expression analyses and concluded 'expression of TCP12, TCP18, and TCP24 was clearly observed in the early ovules (Supplementary Fig. 7d-i)'. I see only a very faint signal for TCP12 and 18 and no signal for TCP24 in the presented pictures!?
- Response to point 6, Reviewer #2. Indeed, this sentence is deleted, but in the remainder of the text and in the model the authors still refer to MADS quartets!? Please modify.
- Response to point 8, Reviewer #2: 'We deleted the sentence to avoiding misunderstanding. As suggested by Reviewer #2, we added some discussion in the Discussion as follows (Line 634-639). Thank you.'. Thanks. But please note that depending on the outcome of the required additional yeast two-hybrid assays it could be that there is more evidence for this mode of action than for the model currently presented in the manuscript. This is also noted by Reviewer 3!

Reviewer #3 (Remarks to the Author):

I was reviewer #3. At the request of the editor, I address responses to both my comments, and those of reviewer #1.

Reviewer 1

1. The authors made additional mutant combinations gaining new insight. The added supplemental figure is great. Their manuscript changes completely address comment 1.
2. I think the reviewer asked too much for in situ on the members of this large gene family. It is not necessary to the conclusions of this manuscript, and the additional data they add is more than sufficient.
3. The authors use in situ hybridization to show that TCP4 is expressed in cells where BEL1, AG and SEP3 are known to be expressed in ovules, supporting the possibility of direct protein interaction in vivo.
4. I agree that the results they obtained were little related to their work and do not need to be added to the manuscript.
The added figure adequately addresses the request for additional phenotypic characterization of the mutant.
The addition of frequencies of the phenotypes addresses the request.
The figure change is good.
Other changes adequately address the comments and improve the manuscript.
5. This is a good comment that led to a nice clarification in the text.
6. Supplemental figures 8,9, and 10 represent substantial additional work and effectively address the reviewer's concerns.
7. The concern is effectively addressed.
8. The concern is effectively addressed by the additional data presented.
9. The concern is effectively addressed by the additional text added to the manuscript.

Minor points:

All effectively addressed by changes in the text.

Reviewer 3

The authors have adequately addressed concerns about the experiments on stability of BEL1 and the involvement of the proteasome through the addition of experiments shown in Fig. 6.

The replacement of the *bel1* mutant with the well-characterized *bel1-1* alleviates concerns

about the mutant images. The changes in the text are an appropriate response and improvement to concerns about timing of ovule development. All other comments were also effectively addressed by changes in the manuscript.

Summary:

The authors have included significant additional experimental work in response to comments by reviewers 1 and 3. The text has also been modified to address additional reviewers' concerns. All concerns expressed by these two reviewers have been effectively addressed. The manuscript is substantially improved and I have no additional concerns. Both reviewers expressed enthusiasm for the topic and main findings of the manuscript in the original reviews so I will not add to that here.

Response to Reviewer Comments

Reviewer #2 (Remarks to the Author):

In the revised version of the manuscript ‘Arabidopsis TCP4 transcription factor inhibits high temperature induced homeotic conversion of ovules’, the authors have carefully addressed the comments and suggestions of the reviewers. A large number of new and additional experiments were done and reported, and various points have been addressed in a proper way and after full satisfaction. Nevertheless, the manuscript still has some flaws. More importantly, issues with a direct effect on the conclusions remain, and I therefore still have strong doubts about part of the conclusions (proposed molecular mode of action of the TCPs) and the model presented at the end of the manuscript.

Points of attention in the revised manuscript (In order of the text and not in order of importance):

1. The authors link their research to thermomorphogenesis in the abstract (e.g. lines 69-71) and discussion (lines 644-650). Considering the definition of thermomorphogenesis, which is also used by the authors (Thermally induced developmental and morphological changes in plants, which are frequently represented by accelerated stem elongation, increased leaf hyponasty, and formation of long, thin leaves.), this is correct. However, the authors couple this to global warming and morphological changes of plants and animals (only reproductive traits) upon exposure to ambient high temperatures. In plants, these are supposed to be adaptive traits to deal with heat. In this case, the morphological changes are not occurring in a ‘wildtype’ plant, but only in the tcpDUO mutant background. As such, I see it as an increased temperature sensitivity due to the lack of numerous functional TCP genes. Therefore, I would rephrase the introduction (first paragraph) and discussion (Lines 644-655) and completely remove the discussion of the underlying mechanism in turtles (H3K27me3 deposition) because it is not relevant and it is out of context.

Answer: Thank Reviewer #2 for the comment. In our manuscript, we did not define the homeotic transformation (ovules converted into carpelloids) of *tcpDUO* as plant thermomorphogenesis, but took it a new aspect of plants affected by moderate high temperature (HT). In the past decades, researchers mainly focused on the molecular mechanisms underlying thermomorphogenesis affected by HT, but the effects of HT on ovules have been ignored. Our results showed that the ovules of wild type become unstable and have the potential homeotic change under HT. TCP transcription factors protect the homeotic conversion from ovules to carpelloids under HT in wild type. We think that the introduction from the relative familiar plant thermomorphogenesis under HT to the novel HT effects on homeotic transformation of ovules meets logic. We rephrased the last sentence of the paragraph in the revised manuscript as follows.

The homeotic conversion of reproductive organs has been found in the sessile fertilized eggs of some fish and reptile species, but the homeotic conversion of reproductive organs in sessile plants has been ignored. The comparison of homeotic transformation by HT in animals and plants in the discussions is also logical.

We think that these are relevant, and could make readers more easily understand the contents, the importance and the novelty of our findings. So, we decide to keep the introduction and the discussion in the revised manuscript.

“the effects of HT on the homeotic transformation of reproductive organs in plants remain largely unknown.”

2. In line with above comment, the reasoning provided in the results section, lines 283-285 ‘Although the effects of HT on the growth of hypocotyls, petioles, roots and male organs have been studied extensively, the mechanisms underlying the effects of HT on female organs, including homeotic conversion of ovule identity, remain largely unknown.’. As the authors show, these morphological effects are not present in wild-type plants! Hence, you can also not study the underlying mechanisms!

Answer: We revised “homeotic conversion of ovule identity” into “the stability of ovule identity” in the manuscript as follows.

“the mechanisms underlying the effects of HT on female organs, including the stability of ovule identity, remain largely unknown.”

2. Results, lines 160, 161: ‘while the seeds frequently arose consecutively from the same side of the locule in that of *tcpSEP* (Fig. 1b, c).’. Is this true? At least for me it is not clear from the presented pictures. Could the observed altered arrangement of ovules not simply be because of the lack of elongation of the pistil (carpel wall and septum)? It seems to me that the seeds are still alternating in position).

Answer: We have rephrased the sentence as follows in the revised manuscript. Thank you.

“except that the siliques of *tcpSEP* were shorter (Fig. 1a), and the seeds were more crowded than that in wild-type control (Fig. 1b, c).”

3. Results, lines 181, 182: ‘the *tcpDUO* mutant produced even shorter siliques’. I don't think that the mutant produces even shorter siliques, but that the authors mean that the short siliques don't stretch that much as in wildtype and that consequently the final silique length is even shorter in comparison to wildtype. If so, please rephrase.

Answer: We have rephrased the sentence as suggested by Reviewer #2 as follows in the revised manuscript. Thank you.

“the siliques of *tcpDUO* mutant did not fully elongated and were even shorter than that of wild-type control”

4. Results, line 383 ‘Venn diagram analysis showed....’. This should be something like ‘Comparative analysis showed.....’. A Venn diagram is only a way to visualize the outcome of the comparison.

Answer: We have revised “Venn diagram analysis” into “The comparison of the ChIP-seq data with the DEGs of WT Vs *tcpDUO*”. Thank you.

5. Results, lines 390-391 ‘....., indicating that TCP4 directly represses CRC and SPT to control ovule identity.’. This cannot be concluded based on these experiments and

observations! The EMSA shows indeed binding and there is a significant expression difference between the mutant and wildtype. However, the latter is only a correlation and not proof that the observed expression difference is caused by TCP4!

Answer: We have revised “indicating that TCP4 directly represses *CRC* and *SPT* to control ovule identity.” into “suggesting that TCP4 directly interacted with the promoters of *CRC* and *SPT*.”. Thank you.

6. Results, line 409 ‘the expression of STK, SHP1 and SHP2 were not obviously changed’. This is a vague non-scientific expression. Furthermore, the change is actually the same as observed in the qRT-PCR shown in Fig S10b!

Answer: We described the results more accurately as suggested by Reviewer #2 in the revised manuscript as follows. Thank you.

“The results showed that the expression of *STK* was up-regulated under normal temperature, but was a little down-regulated under HT in pistils (Supplementary Fig. 10a). *SHP1* was significantly induced in the pistils of *tcpDUO*. The expression of *SHP2* and *SEP* genes were not significantly altered in pistils (Supplementary Fig. 10a).”

7. Results, lines 416, 417 ‘These data suggest that TCPs are important for maintaining the expression level of STK in ovules under HT.’. Indeed! And this could probably explain the observed homeotic conversions at 28 degrees Celsius, considering the known function of STK (see my comments below on the currently proposed molecular and biochemical mode of action of the TCPs).

Answer: Thank Reviewer #2 for the good suggestion. We added this point in the revised manuscript as follows.

“These data suggest that TCPs are important for maintaining the expression level of *STK* in ovules under HT, and the decreased expression of *STK* is consistent with the homeotic conversion in *tcpDUO* under HT.”

“BEL1 degradation induced by HT releases the activity of AG-SEPs, but TCPs and the remaining BEL1 (protected by TCPs) are sufficient to inhibit the activity of AG-

SEPs. TCPs also promote the expression of *STK* to prevent the conversion of ovules into carpelloid structures under HT (Fig. 8b).”

“However, in the *tcpDUO* ovules under HT, the loss of TCP function and HT-induced degradation of BEL1 release the activity of AG-SEPs, the expression of *STK* is decreased without TCPs and the predominant activity of AG-SEPs over AG-SEPs-STK/SHPs promotes the expression of carpel-related genes and the generation of carpelloid structures (Fig. 8d).”

8. Based on the result presented in lines 541, 542 ‘Nevertheless, no ovule-like structures or secondary carpel-like structures were observed in *tcpDUO stk* (Fig. 7l-n)’, the authors conclude in line 542, 543 ‘These results unequivocally demonstrate that TCP transcription factors play a critical role in maintaining the identity of ovules.’. This is a weird conclusion based on this result! It is about the additional effect of a *stk* mutation and the function of *STK*!

Answer: The last sentence is the conclusion of all the results of the section, but not the conclusion for the detailed description of *tcpDUO stk*, which was suggested by Reviewer #1 last time. To avoid misunderstanding, we revised the last sentence and took it as an independent paragraph. Thank you.

“The ovule defective phenotypes of *tcpDUO bell* and *tcpDUO stk* unequivocally demonstrate that TCP transcription factors play a critical role in maintaining the identity of ovules.”

9. Results lines 553-558 ‘In wild-type ovules under normal temperature (22°C), TCPs form complexes with BEL1 and AG-SEP quartets to inhibit the activity of AG-SEP, and the predominant activity of AG-SEPs-STK/SHPs quartets promotes the expression of ovule-related genes to determine ovule identity (Fig. 8a).’. This is not clear (what is the mechanism?) and according to me, this model is not very plausible. How can the interaction of BEL1 with AG-SEP3 have an effect on the activity of the other complex? This requires an explanation and substantiation by experimental data.

Answer: This is a seesaw working model. Because AG, SEPs, STK and SHPs co-exist in ovules. The ovules are specified by AG-SEPs-STK/SHPs complexes, while the carpels are specified by AG-SEPs. The ovule identity is determined by the balance between the two kinds of complexes. If the activity of AG-SEPs-STK/SHPs is higher than that of AG-SEPs, the ovule identity is specified. If the activity of AG-SEPs is higher than that of AG-SEPs-STK/SHPs, the carpelloid identity is specified. The interaction of TCPs and BEL1 with AG-SEPs leads to the inhibition of the activity of AG-SEPs. Although the activity of AG-SEPs-STK/SHPs is not changed, but when compared to the TCPs-BEL1 inhibited AG-SEPs, the activity is relatively higher. When the function of BEL1 and TCPs is lost, the activity of AG-SEPs is relatively higher, and the carpelloid structures were formed in *tcpDUO bell* mutants. We revised the paragraph for more clarity in the revised manuscript as follows.

“We propose a seesaw working model for the function of TCPs in determining ovule development and identity. In wild-type ovules, AG, SEPs, STK and SHPs co-exist in ovules. The ovules are specified by AG-SEPs-STK/SHPs, while the carpels are specified by AG-SEPs. The balance between the two kinds of complexes is important for the specification of ovule identity¹³⁻¹⁵. Under normal temperature (22°C), TCPs interacts with BEL1 and AG-SEPs to inhibit the activity of AG-SEPs in wild type. The inhibition of AG-SEPs activity leads to the relatively higher activity of AG-SEPs-STK/SHPs which promotes the expression of ovule-related genes to determine ovule identity (Fig. 8a).”

10. Lines 571-573 ‘In the tredecuple mutant *tcpDUO stk*, the loss of STK function reduces the activity of AG-SEPs-SHPs, and BEL1 is insufficient to repress the activity of AG-SEP quartets without the help of TCPs’. This is a very strange synthesis. How can the loss of STK have an effect on the BEL1-AG-SEP3 complex in which it doesn’t take part?

Answer: As explained above, the balance between the activity of AG-SEPs-STK/SHPs and that of AG-SEPs determines the ovule identity. STK facilitates the activity of AG-SEPs-STK/SHPs, while TCPs inhibit the activity of AG-SEPs. In the *tcpDUO stk*, on

one side, the loss of *STK* function inhibits the activity of AG-SEPs-STK/SHPs, on the other side, the loss of *TCP* function facilitates the activity of AG-SEPs. So, in *tcpDUO stk*, the activity of AG-SEPs is relatively higher, leading to the conversion of ovules into carpelloids in *tcpDUO stk*. We rewrote the sentence for more clarity in the revised manuscript as follows.

“In the tredecuple mutant *tcpDUO stk*, the loss of *STK* function reduces the activity of AG-SEPs-SHPs, while the loss of function of *TCP* released the activity of AG-SEPs; therefore, the relative higher activity of AG-SEPs promotes the formation of carpelloid structures instead of ovules in the carpels of *tcpDUO stk* under normal temperature (Fig. 8f).”

11. The discussion in lines 625-634 is very speculative and not supported by data. I propose to remove this part.

Answer: We have removed this part in the revised manuscript as suggested by Reviewer #2. Thank you.

12. Fig 4 legend: K should be H.

Answer: We fixed the error in the revised manuscript. Thank you.

13. Legend Fig. S9 ‘The FPKM from RNA-seq data indicated that *WUS* was induced in *tcpDUO* under normal temperature or HT’. Under normal temperature, no significant difference is observed so this cannot be concluded.

Answer: We have revised it in the manuscript as follows. Thank you.

“Supplementary Fig. 9 *WUS* was up-regulated in the ovules of *tcpDUO* under normal temperature or HT. a, The FPKM from RNA-seq data indicated that *WUS* was induced in pistils of *tcpDUO* under HT. **b,** The RT-qPCR analysis using mature ovules indicated that *WUS* was up-regulated in *tcpDUO* under normal temperature or HT. ”

Points related to the rebuttal:

14. Response to point 4, i), Reviewer #1: ‘The staining with aniline blue showed that

pollen germinated on the stigma and pollen tubes grew through styles in pistils from both *tcpDUO* and wild-type control (Supplementary Fig. 2e, f), indicating that the transmitting tract of *tcpDUO* was rather normal.'. Indeed, pollen germination seems not to be affected, but based on the presented pictures (Fig S2 e and f) ingrowth seems to be strongly reduced in the *tcpDUO* mutant. So based on what is presented, I do not agree with the conclusion.

Answer: We deleted “indicating that the transmitting tract of *tcpDUO* was rather normal.” in this sentence. Because this manuscript is focused on the roles of TCPs on the ovule development, the difference of pollen tube growth in the *tcpDUO* pistils from that of wild-type control will be investigated in detail in the future study. Thank you.

15. Response to point 4, v), Reviewer #1: Considering the new data on pollen tube staining in the *tcpDUO* mutant upon pollination with wild-type pollen, the possibility that many ovules are unfertilized is very plausible and should be included as well.

Answer: Because our statistical analysis of defective ovules in pistils of *tcpDUO* is 84.3%, and the seed-setting rate is about 18%. The rate of defective ovules matches the seed-setting rate. We do not think that the low seed-setting rate is due to the abnormal fertilization.

16. Point 8, Reviewer #1 and point 5, Reviewer#2: I do not agree! You cannot simply remove data that you do not expect or cannot explain. The authors should perform the suggested additional negative control (interaction tests with other unrelated TCP), and even more important, an investigation of the interaction between TCP4/18 and STK/SHPs. In case interaction is also detected with STK, the presented model which is for a large part based on these Y2H outcomes doesn't hold!

Answer: We added the negative controls using the other unrelated TCP16 and STK, SHP1 and SHP2. We did not detect the interactions in the negative controls. We included the results of TCP16 and STK in the Supplemental Figure 10 and Results in the manuscript.

“To identify the domains in AG and SEP3 responsible for their interactions with TCP4 and TCP18, we separated both AG and SEP3 into three fragments: AG-MI or SEP3-MI (containing the MADS-box and Intervening domain), AG-IK or SEP3-IK (containing the Intervening and Keratin domains), and AG-C or SEP3-C (containing the C-terminus) (Supplementary Fig. 11a, c). Yeast two hybrid assays indicated that these truncated proteins all interacted with TCP4 and TCP18 (Supplementary Fig. 11b, d), while no interactions were detected in the negative controls. That is, no interactions between AG and TCP16, SEP3 and TCP16, STK and TCP4, and STK and TCP18 were found (Supplementary Fig. 11b, d). These data suggested that the various domains of AG and SEP3 play roles in their interactions with TCPs.”

17. Response to point 2, Reviewer #2: The authors added the sentence ‘The results showed that no TCPs were significantly regulated’ and this is not completely correct. Please change to something like ‘The results show that none of the TCPs was more than 2 fold changed.’.

Answer: We have revised it as suggested by Reviewer #2 in the revised manuscript as follows.

“The results showed that the gene expression was not changed more than 2 fold in the *TCPs* under HT (false discovery rate < 0.01; fold-change ≥ 2.0 or ≤ -2.0).”

18. Response to point 3, Reviewer #2: The authors performed extra expression analyses and concluded ‘expression of TCP12, TCP18, and TCP24 was clearly observed in the early ovules (Supplementary Fig. 7d-i)’. I see only a very faint signal for TCP12 and 18 and no signal for TCP24 in the presented pictures!?

Answer: Please compare the staining in the negative control shown in Supplementary Fig. 7j. *TCP12* and *TCP18* were clearly expressed in the ovule at FG2. *TCP24* was clearly expressed in the ovule at FG1. qRT-PCR analysis also indicated that they were expressed in ovules. We have adjust the contrast of pictures to make them clearer.

19. Response to point 6, Reviewer #2. Indeed, this sentence is deleted, but in the remainder of the text and in the model the authors still refer to MADS quartets!? Please modify.

“Last time point 6. Line 400/401: ‘...suggesting that TCPs and BEL1 formed a complex with AG-SEP3 to inhibit the activity of AG-SEP3 quartets.’. This could be, but the authors do not provide any evidence that the effect is specifically on quartet formation. Note that the performed experiments do not allow to say anything about the exact stoichiometry of the formed complexes! Please modify.”

Answer: We have not concluded that TCPs specifically affected quartet formation of AG-SEPs in the other part of our manuscript. We suggest that the interaction of TCPs and AG-SEPs affected the transactivation activity of AG-SEPs quartets. We deleted “quartets” or termed it as “complexes” in the revised manuscript.

20. Response to point 8, Reviewer #2: ‘We deleted the sentence to avoiding misunderstanding. As suggested by Reviewer #2, we added some discussion in the Discussion as follows (Line 634-639). Thank you.’. Thanks. But please note that depending on the outcome of the required additional yeast two-hybrid assays it could be that there is more evidence for this mode of action than for the model currently presented in the manuscript. This is also noted by Reviewer 3!

Answer: We added the required additional yeast two-hybrid assays. No interactions were found in the negative controls. We included the results in the Supplemental Figure 10 and Results in the manuscript.

Reviewer #3 (Remarks to the Author):

I was reviewer #3. At the request of the editor, I address responses to both my comments, and those of reviewer #1.

Reviewer 1

1. The authors made additional mutant combinations gaining new insight. The added supplemental figure is great. Their manuscript changes completely address comment 1.
2. I think the reviewer asked too much for in situ on the members of this large gene family. It is not necessary to the conclusions of this manuscript, and the additional data they add is more than sufficient.
3. The authors use in situ hybridization to show that TCP4 is expressed in cells where BEL1, AG and SEP3 are known to be expressed in ovules, supporting the possibility of direct protein interaction in vivo.
4. I agree that the results they obtained were little related to their work and do not need to be added to the manuscript.

The added figure adequately addresses the request for additional phenotypic characterization of the mutant.

The addition of frequencies of the phenotypes addresses the request.

The figure change is good.

Other changes adequately address the comments and improve the manuscript.

5. This is a good comment that led to a nice clarification in the text.
6. Supplemental figures 8,9, and 10 represent substantial additional work and effectively address the reviewer's concerns.
7. The concern is effectively addressed.
8. The concern is effectively addressed by the additional data presented.
9. The concern is effectively addressed by the additional text added to the manuscript.

Minor points:

All effectively addressed by changes in the text.

Answer: Thank reviewer #3 for the approval.

Reviewer 3

The authors have adequately addressed concerns about the experiments on stability of BEL1 and the involvement of the proteasome through the addition of experiments shown in Fig. 6.

The replacement of the bell mutant with the well-characterized bell-1 alleviates

concerns about the mutant images. The changes in the text are an appropriate response and improvement to concerns about timing of ovule development.

All other comments were also effectively addressed by changes in the manuscript.

Summary:

The authors have included significant additional experimental work in response to comments by reviewers 1 and 3. The text has also been modified to address additional reviewers' concerns. All concerns expressed by these two reviewers have been effectively addressed. The manuscript is substantially improved and I have no additional concerns. Both reviewers expressed enthusiasm for the topic and main findings of the manuscript in the original reviews so I will not add to that here.

Answer: Thank reviewer #3 for the approval.